# Skeleton-of-Thought: Prompting LLMs for Efficient Parallel Generation

**Xuefei Ning**[1*]
foxdoraame@gmail.com

**Zinan Lin**[2*]
linzinan1995@gmail.com

**Zixuan Zhou**[14*]
zhouzx21@mails.tsinghua.edu.cn

**Zifu Wang**[3]
zifu.wang@kuleuven.be

**Huazhong Yang**[1]
yanghz@tsinghua.edu.cn

**Yu Wang**[1]
yu-wang@tsinghua.edu.cn

[1] Department of Electronic Engineering, Tsinghua University, Beijing, China
[2] Microsoft Research, Redmond, Washington, USA
[3] ESAT-PSI, KU Leuven, Leuven, Belgium
[4] Infinigence-AI

Website: https://sites.google.com/view/sot-llm
Code: https://github.com/imagination-research/sot

## Abstract

This work aims at decreasing the end-to-end generation latency of large language models (LLMs). One of the major causes of the high generation latency is the sequential decoding approach adopted by almost all state-of-the-art LLMs. In this work, motivated by the thinking and writing process of humans, we propose *Skeleton-of-Thought (SoT)*, which first guides LLMs to generate the *skeleton* of the answer, and then conducts parallel API calls or batched decoding to complete the contents of each skeleton point *in parallel*. Not only does SoT provide considerable speed-ups across 12 LLMs, but it can also potentially improve the answer quality on several question categories. SoT is an initial attempt at data-centric optimization for inference efficiency, and showcases the potential of eliciting high-quality answers by explicitly planning the answer structure in language.

## 1 Introduction

Large language models (LLMs) (Brown et al., 2020; Touvron et al., 2023a; Du et al., 2022; OpenAI, 2023; Zheng et al., 2023) have shown exceptional performance in natural language processing and chatbot systems. However, the inference process of the state-of-the-art LLMs is slow, hindering their interactive use. For example, it takes 22 seconds for Claude (Anthropic, 2023) (accessed through Slack API) and 43 seconds for Vicuna-33B V1.3 (a 33B LLaMA-based model, running locally on one NVIDIA A100 GPU) to answer the question in Fig. 1.

We conclude three major causes of LLMs' slow inference: (1) A *large model size* requires a large amount of memory, memory access, and computation. For example, the FP16 weights of 175B GPT-3 take 350GB memory, which means at least 5×80GB A100 GPUs are needed to keep the model in GPU memory. Even with enough GPUs, the heavy memory access and computation slow down the inference. (2) The *attention operation* in the prevailing transformer architecture is I/O bounded and has a quadratic memory and computation complexity in sequence length. (3) The *sequential decoding* approach in inference generates tokens one by one. This approach introduces a significant inference latency since the generation of tokens cannot be parallelized. There is a bunch of literature addressing the first two axes: *large model size* (Xiao et al., 2022; Frantar et al., 2022; Lin et al., 2023; Sheng et al., 2023; Wang et al., 2021) and *attention operation* (Kitaev et al., 2020; Wang et al., 2020;

---

*Equal contribution.

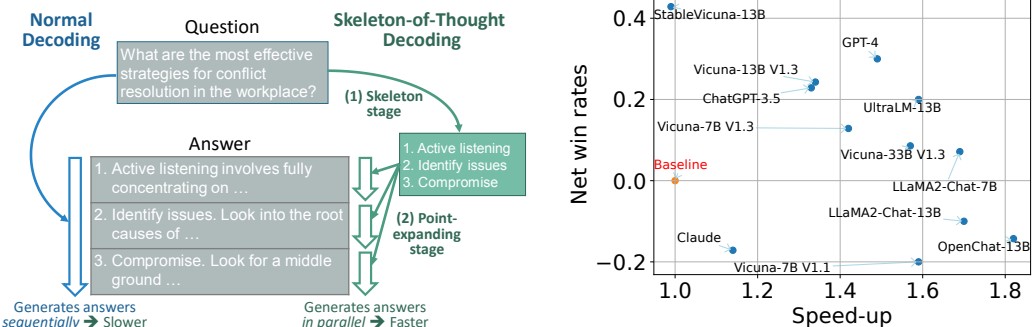

Figure 1: **Left:** An illustration of Skeleton-of-Thought (SoT). Instead of producing answers sequentially, SoT produces different parts of answers *in parallel*. In more detail, given the question, SoT first prompts the LLM to give out the skeleton, then conducts batched decoding or parallel API calls to expand multiple points in parallel, and finally aggregates the outputs to get the final answer. **Right:** The net win rates and speed-ups of SoT with router (SoT-R) compared to normal generation on Vicuna-80. The net win rate is the difference between the fraction of questions that SoT-R has better and worse answers than normal generation. The speed-up is the ratio between the latency of normal and SoT-R generation. $(1.0, 0.0)$ represents normal generation. Higher is better on both axes. For most models, SoT-R not only accelerates the generation but also improves the quality of the answers (evaluated with FastChat metric (Zheng et al., 2023)). See § 3.2 and 4 for more details.

Dao et al., 2022; Zaheer et al., 2020; Chen et al., 2023b). These works either compress/redesign the model (Xiao et al., 2022; Frantar et al., 2022; Lin et al., 2023; Kitaev et al., 2020; Wang et al., 2020; Dao et al., 2022; Zaheer et al., 2020) or redesign the serving system (Sheng et al., 2023; Chen et al., 2023b) and hardware (Wang et al., 2021).

In contrast to prior work, we tackle the third axis and question the common assumption that LLMs have to do fully sequential decoding. We show the feasibility of **parallel decoding of off-the-shelf LLMs *without* any changes to their model, system, or hardware**. For instance, for the question in Fig. 1, we can reduce the latency from 22 seconds to 12 seconds (1.83× speed-up) with Claude, and from 43 seconds to 16 seconds (2.69× speed-up) with Vicuna-33B V1.3 on an NVIDIA A100.

The idea stems from reflecting on how humans ourselves answer questions. Humans do *not* always think about questions and write answers in a sequential fashion. In contrast, for many question types, we first derive the *skeleton* according to some protocols and strategies, and then add evidence and details to explain each point. This is especially the case on occasions like offering consultancy, taking tests, writing papers, and so on. This intuition has our back to question the necessity of fully sequential decoding. In this paper, we propose *Skeleton-of-Thought (SoT)*. Specifically, as shown in Fig. 1, we guide the LLM to derive a skeleton first by itself. Based on the skeleton, the LLMs can complete each point *in parallel* so that we get a speed-up. SoT can be utilized to accelerate both open-source models with batched decoding and API-based models with parallel API calls.

The current SoT is suitable for questions that require a long answer whose structure can be planned ahead, while not suitable for questions that require step-by-step reasoning or only need a short answer. Therefore, to make the overall solution more practical, we design an extension, SoT with router (SoT-R), which employs a router to only trigger SoT for suitable questions.

We test SoT on 12 recently released LLMs. Not only does SoT provide considerable speed-ups (up to 2.39×), but it can also improve the answer quality in many cases (Fig. 1).

Note that in contrast to existing model- and system-level efforts for inference efficiency, SoT takes a novel "data-level" pathway by letting the LLM organize its output content. This novel perspective is becoming feasible and is expected to grow in importance, owing to the evolving capabilities of state-of-the-art LLMs. We hope this work can stimulate more research in the realm of data-centric optimization (Zha et al., 2023; HazyResearch, 2023) for efficiency.

---

**Prompt 1. Skeleton Prompt Template $T^s$**

[**User:**] You're an organizer responsible for only giving the skeleton (not the full content) for answering the question. Provide the skeleton in a list of points (numbered 1., 2., 3., etc.) to answer the question. Instead of writing a full sentence, each skeleton point should be very short with only 3∼5 words. Generally, the skeleton should have 3∼10 points. Now, please provide the skeleton for the following question.
{*question*}
Skeleton:
[**Assistant:**] 1.

---

**Prompt 2. Point-Expanding Prompt Template $T^{pe}$**

[**User:**] You're responsible for continuing the writing of one and only one point in the overall answer to the following question.

{*question*}

The skeleton of the answer is

{*skeleton*}

Continue and only continue the writing of point {*point index*}. Write it **very shortly** in 1∼2 sentence and do not continue with other points!
[**Assistant:**] {*point index*}. {*point skeleton*}

---

The rest of the paper is organized as follows. We first introduce SoT in § 2 and show its results in § 3. Then, we expand on the SoT-R extension in § 4. § 5 positions SoT in the research ecosystem (expanded in App. D). Finally, we analyze the limitations and share outlooks of SoT in § 6.

## 2    SKELETON-OF-THOUGHT (SOT)

### 2.1    METHOD

**Overview.** Based on the intuition that humans usually think about and answer a question in an organized way, the core idea of this work is to guide the LLM itself to give a skeleton first and then write the overall answer parallelly instead of sequentially. Fig. 1 illustrates how SoT produces the final answer to a user *question q*.

*(1) Skeleton stage.* SoT first assembles a *skeleton request*, $T^s$(question $= q$), using the *skeleton prompt template $T^s$* (Prompt 1, and Prompt 3 in App. B.1) with the question $q$ as the parameter. The skeleton prompt template is written to guide the LLM to output a concise skeleton of the answer. Then, we extract the $B$ points from the *skeleton response $R^s$* of the LLM.

*(2) Point-expanding stage.* Based on the skeleton, we let the LLM expand on each point in parallel. Specifically, for the point with index $b$ and skeleton $R^s_b$, SoT uses $T^{pe}$(question $= q$, skeleton $= R^s$, point index $= b$, point skeleton $= R^s_b$) as the *point-expanding request* for the LLM, where $T^{pe}$ is the *point-expanding prompt template* (Prompt 2). Finally, after completing all points, we concatenate the point-expanding responses $\{R^{pe}_b\}_{b=1,\cdots,B}$ to get the *final answer*.

**Parallel point expanding.** We conduct *parallel* point-expanding so that SoT is able to achieve a speed-up than normal decoding.

*(1) For proprietary models with only API access*, we can issue multiple parallel API calls to get an end-to-end latency gain at the cost of an increased number of API requests and tokens.

*(2) For open-source models that we can run locally*, we let them process the point-expanding requests as a batch (paddings are added to the left of the point-expanding requests). We explain below why this could achieve speed-ups. A typical LLM generative process consists of two phases: (a) the *prefilling* phase in which the prompt is parsed to generate the key-value cache for further use, and (b) the *decoding* phase in which tokens are generated one by one in a sequential manner. The decoding phase accounts for the majority of the end-to-end latency, especially when generating a long response. Note that the decoding phase is bottlenecked by weight loading instead of activation

loading or computation.[1] Consequently, running LLM inference with increased batch sizes does not increase the per-token latency much. Therefore, SoT allows us to decode roughly $B\times$ more tokens within the same amount of time if we parallelly decode $B$ points. See App. E for the expanded discussions and the supporting experiments. Please refer to App. B for more implementation details.

## 3 SoT Evaluation

**Datasets.** We evaluate SoT on two recent assistant-style datasets: (1) Vicuna-80 (Chiang et al., 2023), which contains 80 questions spanning nine categories, such as *coding*, *math*, *writing*, *role-play*, and so on, and (2) WizardLM (Xu et al., 2023), which contains 218 questions spanning more categories and diverse difficulties. Due to space constraints, we only report Vicuna-80 results in the main paper, and defer WizardLM results to the Apps. G and I.

**Models.** We test SoT on 12 models, including 9 open-source models and 3 API-based models. We obtain the weights of all the open-source models from Hugging Face. See App. A for more details.

### 3.1 Evaluation of Efficiency

**API-based models.** We record the latency of every API call with `start = time.time(); ...; elapsed_time = time.time() - start`, and add the latency of the skeleton API call and the slowest point-expanding API call as the SoT latency.

**Open-source models.** All open-source models we currently evaluate are based on the LLaMA 7B, 13B, or 33B architectures. Thus, to enable fast analysis, we first make a latency profiling table for each LLaMA architecture on NVIDIA A100. The table contains the architecture's (1) latency for prefilling sequences of length 1 to 700 with different batch sizes (from 1 to 16), and (2) decoding one token with a context of length 1 to 1024 with different batch sizes (from 1 to 16). With these three latency profiling tables, given the number of points $B$, the token lengths of the requests and responses in the skeleton and point-expanding stages, we can quickly estimate the SoT latency by simply looking up entries in the tables and adding them up. See App. F for a more detailed description of how we conduct the profiling and estimate the latency.

In addition to the above approach, we also compare the actual latency of SoT and normal sequential generation (abbreviated as "normal" in the following discussion) in App. G.1.4.

The rest of this section shows the speed-ups of SoT on different models (§ 3.1.1) and question categories (§ 3.1.2). In addition, we also report the latency breakdown of SoT stages in App. G.1.2 and the SoT speed-ups on an RTX 3090 GPU in App. G.1.3.

### 3.1.1 Speed-up Breakdown: Models

We investigate how SoT reduces the end-to-end latency on different models. Fig. 2a shows the average speed-up for each model across all question categories. We can see that SoT obtains a $>2\times$ speed-up (up to $2.39\times$) on 8 out of 12 models.

We report the detailed statistics about token lengths and numbers of points in Fig. 11. (1) In terms of *the point number $B$* (Fig. 11a), LLaMA2, Vicuna-7B V1.1, Vicuna-7B V1.3, and ChatGPT-3.5 yield relatively fewer points ($<6$), while GPT-4 and StableVicuna-13B generates the largest number of points on average ($\approx 9$). (2) Regarding *the point-expanding response length*, Figs. 11b to 11d show that the API-based models, ChatGPT-3.5, Claude, and GPT-4, follow the point-expanding request better and generate shorter point-expanding responses than the open-source models. One can also notice that StableVicuna-13B's longest point-expanding responses for many question categories can be as lengthy as the overall normal answer, since it fails to adhere to the "Write it **very shortly**" instruction in the point-expanding request. Consequently, SoT cannot accelerate StableVicuna-13B well. (3) Regarding *the length balance degree between point responses*, Fig. 11e shows that LLaMA2 and the API-based models generate more balanced point-expanding responses. (4) As for *the overall length of the final aggregated answer* (Fig. 11f), employing SoT on most models results in answers that are, on average, $1\sim 2\times$ longer than the normal answer.

---

[1] This is true when the number of concurrent queries is small; see § 6 for discussion on other scenarios.

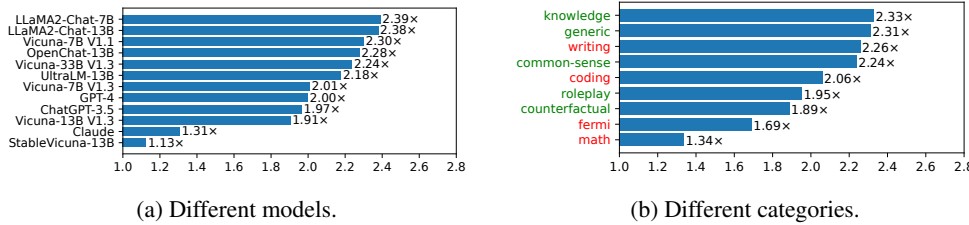

(a) Different models.  (b) Different categories.

Figure 2: Average speed-ups of SoT on different models and question categories.

### 3.1.2 SPEED-UP BREAKDOWN: QUESTION CATEGORIES

Here we investigate how SoT reduces the end-to-end latency for different question categories. Fig. 2b shows the average speed-up for each question category across all models. The question categories for which SoT can provide high-quality answers are marked in green, and other categories are marked in red (see § 3.2.3 for the answer quality evaluation). We can see that SoT can obtain speed-ups for all question categories. For the five question categories that SoT can provide high-quality answers (i.e., *knowledge*, *generic*, *common-sense*, *roleplay*, *counterfactual*), SoT can speed up the overall answer generation process by $1.89\times$ to $2.33\times$ in the meantime.

### 3.2 EVALUATION OF ANSWER QUALITY

In order to compare the answer quality of the normal sequential generation (abbreviated as "normal" in the following discussion) and SoT generation, we adopt two LLM-based evaluation frameworks: FastChat (Zheng et al., 2023) and LLMZoo (Chen et al., 2023c). The evaluation process is to present a question and a pair of answers (from normal or SoT generation) to an LLM judge (GPT-4 in the main paper; see App. I.4 for the results evaluated using ChatGPT-3.5) and ask for its preference.

Here are more details about the evaluation of the answer quality:

*(1) Detailed metrics.* FastChat provides one metric for the general answer quality. In addition to a general metric, LLMZoo provides five detailed metrics on the answers' coherence, diversity, immersion, integrity, and relevance.

*(2) Question categories.* FastChat provides two special evaluation prompts for coding and math questions for more accurate evaluation, whereas LLMZoo does not. Following the implementation in LLMZoo, we exclude math and coding questions in all LLMZoo evaluation results.

*(3) Extentions to avoid evaluation bias.* To avoid the potential bias from the order of the two answers presented to the LLM judge, we extend FastChat and LLMZoo evaluation frameworks by running the evaluation twice with either ordering of the two answers. In either evaluation, a score of 1, 0, and -1 is assigned when SoT wins, ties, or loses, respectively. The final evaluation is that SoT wins/ties/loses when the sum of the two scores is positive/zero/negative. For example, if SoT wins in one evaluation and loses in the other evaluation, the result is "tie". If SoT wins (loses) in one evaluation and ties in the other, the result is "win" ("lose").

*(4) Net win rates.* We further define net win rates to give a summarized view of the answer quality. Given the number of questions that SoT wins (#win) and loses (#lose), we define *net win rates* as $^{\#\text{win}-\#\text{lose}}/_{\text{total number of questions}}$. 0% means that SoT performs competitively to the normal baseline (wins and loses in the same number of questions). Higher values mean that SoT performs better.

In the following sections, we first present the overall quality of SoT answers (§ 3.2.1), and then go into the details across different question categories (§ 3.2.3), models (§ 3.2.2), and metrics (§ 3.2.4).

### 3.2.1 OVERALL QUALITY

In Fig. 3, we show the win/tie/lose rates (the percentage of the cases when SoT wins/ties/loses compared to normal generation) across all models and questions using the two metrics from FastChat and LLMZoo that capture the general quality of the answers. We notice a discrepancy between the two metrics on when SoT is strictly better than the baseline (45.8% v.s. 29.5%). Despite that, the two metrics agree that SoT is not worse than the baseline in around 60% of the cases, and the win

rates are close to the lose rates. *This result suggests that the answers of SoT maintain good quality of that of the normal generation.*

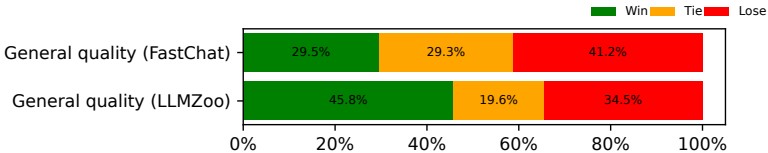

Figure 3: Win/tie/lose rates of SoT v.s. normal generation using "general" metrics from FastChat and LLMZoo. SoT performs better than or equal to normal generation in around 60% cases.

### 3.2.2 QUALITY BREAKDOWN: MODELS

We compute net win rates on all models in Fig. 4. Again, we see that the two general metrics from FastChat and LLMZoo have different absolute values but similar rankings. In particular, both metrics agree that OpenChat-13B, Vicuna-7B V1.1, Claude, LLaMA2-Chat-13B have *low* net win rates, whereas Vicuna-13B V1.3, StableVicuna-13B, and UltraLM-13B have *high* net win rates.

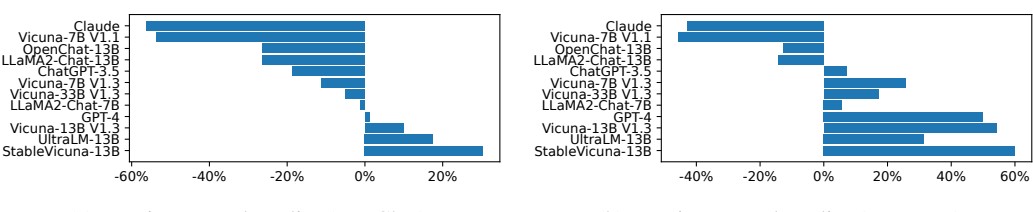

(a) Metric: general quality (FastChat).       (b) Metric: general quality (LLMZoo).

Figure 4: Net win rates of SoT on different models.

We investigate the answers in App. I.1.1, and summarize the key takeaways as follows. Some models have low SoT net win rates as they cannot understand the skeleton and point-expanding prompts well. Some other models have low SoT net win rates as their normal answers already have good quality, making it hard for SoT to beat them (e.g., Claude). For models that are able to understand the SoT prompts and the normal answers are not good enough, SoT can improve the answer quality. We expect that further improving SoT prompts or fine-tuning the models can make it easier for LLMs to understand the skeleton and point-expanding prompts and ultimately result in better answer quality.

### 3.2.3 QUALITY BREAKDOWN: QUESTION CATEGORIES

We compute net win rates on all question categories in Fig. 5. Similar to Fig. 3, we see that LLMZoo tends to be more optimistic about the quality of SoT than FastChat. Nevertheless, the conclusions are consistent: SoT performs relatively *well* on *generic*, *common-sense*, *knowledge*, *roleplay*, and *counterfactual*, and relatively *poorly* on *writing*, *fermi*, *math*, and *coding*.

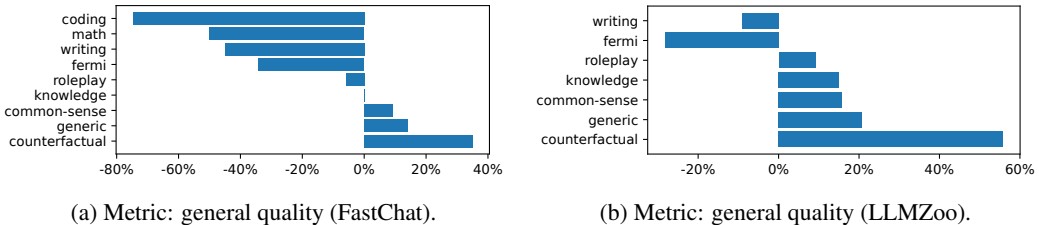

(a) Metric: general quality (FastChat).       (b) Metric: general quality (LLMZoo).

Figure 5: Net win rates of SoT on different question categories.

We investigate the answers in App. I.1.2, and summarize the key takeaways as follows. SoT performs well when the question can be answered in several points whose details can be expanded independently. This includes a wide range of real-world questions. On the other hand, it is fundamentally challenging to apply SoT on questions that require step-by-step thinking, in which the latter steps require the details from the earlier steps, such as math questions. To make SoT general

across broader question categories, one promising pathway is to enable SoT to adaptively fall back to normal generation, which we explore in § 4. Interestingly, our results suggest that some LLMs are already able to do that occasionally without special prompting or tuning (see App. I.1.2).

### 3.2.4 QUALITY BREAKDOWN: METRICS

In Fig. 6, we show more detailed metrics from LLMZoo to reveal in which aspects SoT can improve or hurt the answer quality. On average, we can see that SoT improves the diversity and relevance while hurting the immersion and coherence.

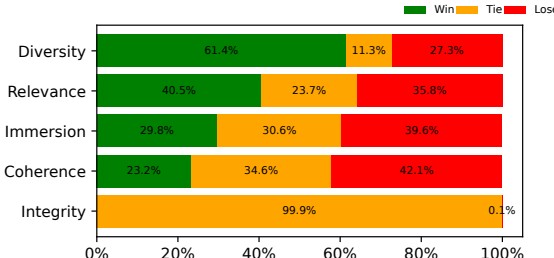

Figure 6: Win/tie/lose rates of SoT v.s. normal generations using metrics from LLMZoo. SoT performs well on diversity and relevance, and relatively worse on coherence and immersion.

Through answer investigation (App. I.1.3), we summarize the key takeaways as follows. The skeleton stage of SoT explicitly require LLMs to discuss the answers from multiple aspects without filler words. This improves the diversity and relevance of the answers. As for coherence and immersion, SoT is not worse than the normal generation around 60% of the time. One future direction is to improve the SoT prompts or pipeline so that the answers can be better in more metrics.

## 4 SOT WITH ROUTER (SOT-R): ADAPTIVELY TRIGGERING SOT

In § 3, we see that SoT provides considerable speed-ups while maintaining (or even improving) answer quality for many question types. However, the biggest limitation is that SoT is not suitable for questions that require step-by-step reasoning (§ 3.2.3). Towards pushing the practical adoption of SoT, we explore the possibility of *adaptively triggering SoT* only when it is suitable. To achieve that, we propose a *router* module that decides if SoT should be applied for the user request, and then call either SoT or normal decoding accordingly. This paradigm aligns with the recent trends of composing multiple models to solve complicated tasks (Chase, 2022; Shen et al., 2023). To implement the router, we explore two options: LLM prompting as the router (no model training is needed) (§ 4.1), and trained RoBERTa as the router (§ 4.2). The evaluation is provided in § 4.3.

### 4.1 PROMPTING ROUTER

We directly ask an LLM if the question is suitable for SoT. More specifically, we ask the LLM if the desired answer is in a list of independent points (see App. C.1 for the prompt). If the answer is yes, we will use SoT; otherwise, we will use normal generation (i.e., directly feeding the question to the LLM). We employ GPT-4 as the LLM router given its strong capability.

### 4.2 TRAINED ROUTER

While leveraging GPT-4 as the router obviates the need for model training, its performance remains sensitive to prompt design. Therefore, we approach the problem as a sequence classification task by fine-tuning a small language model as the router. Specifically, we annotate the LIMA dataset (Zhou et al., 2023) as the training set to train a RoBERTa model (Liu et al., 2019), which has only 120M parameters. Details about the annotation and training can be found in Apps. C.2.1 and C.2.2.

### 4.3 SOT-R EVALUATION

We compare SoT and SoT-R under the same evaluation setup in § 3. Besides the prompting and trained routers, we also consider a "human router" where we manually judge whether SoT should be applied for each question. This serves as a benchmark for comparison.

### 4.3.1 EVALUATION OF EFFICIENCY

Fig. 7 shows the speed-ups of SoT and SoT-R for different models on Vicuna-80 (see App. G.2 for results on the WizardLM dataset). We can see that: (1) As expected, SoT-R obtains lower speed-ups than SoT, since SoT is not triggered for some questions and the router induces a small latency overhead. Nevertheless, SoT-R can still benefit most models with >1× speed-ups. (2) SoT-R with the trained router obtains slightly higher speed-ups for 7 out of 12 models on Vicuna-80, while SoT-R with the prompting router obtains higher speed-ups for all models on WizardLM (Fig. 17).

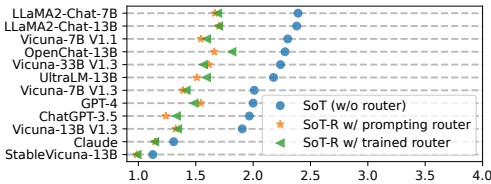
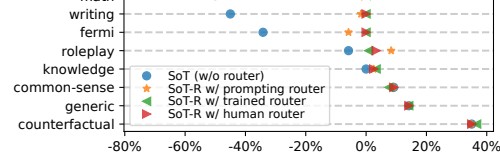

Figure 7: Speed-ups of SoT and SoT-R on different models across all question categories of the Vicuna-80 dataset.

Figure 8: Net win rates of SoT and SoT-R on different question categories of the Vicuna-80 dataset (evaluated with the FastChat metrics).

### 4.3.2 EVALUATION OF ANSWER QUALITY

Fig. 8 shows the net win rates (averaged across all models) of SoT and SoT-R on Vicuna-80 with the FastChat metrics (see App. I.2 for results of the WizardLM dataset and LLMZoo metrics). We can see that: (1) SoT-R significantly improves the answer quality on questions where SoT is not suitable (e.g., *coding*, *math*, *writing*, *fermi*) by falling back to normal decoding. At the same time, SoT-R maintains answer quality improvements on questions where SoT is good at. (2) The trained router performs similar to (on Vicuna-80) or better than (on WizardLM; see App. I.2) the prompting router. This accords with our intuition in § 4.2. (3) The prompting and trained routers could even surpass human router (e.g., on roleplay questions; see more examples on WizardLM in App. I.2).

We discuss the consistency across three routers in App. C.3. The primary takeaways include: (1) on Vicuna-80, there is a notable consistency among all three routers, and (2) on WizardLM, greater discrepancies emerge, with the trained router showing higher alignment with human annotations.

## 5 SoT IN THE CONTEXT OF LITERATURE

This section positions SoT in related work to reveal how SoT (1) is connected to, (2) is different from, and (3) can harness the power of other methods. See App. D for the expanded discussion.

**Efficient LLM methods at model and system levels.** At the model level, prior work proposes efficient architectures, including dynamic mixture-of-experts (Lepikhin et al., 2021), low-complexity attention (Kitaev et al., 2020), and multi-query attention (Shazeer, 2019). However, they usually require a significant re-training cost. In contrast, compression methods require a smaller amount of fine-tuning cost by reducing the complexity of pre-trained LLMs, such as quantization (Frantar et al., 2022) and weight or activation sparsification (Mishra et al., 2021; Zaheer et al., 2020).

At the system level, prior work (1) optimizes the computational graph (Dao et al., 2022), (2) optimizes the assignment and scheduling of computational graph on devices (Sheng et al., 2023), or (3) designs batching or caching mechanisms for serving multiple users (Fang et al., 2021). These techniques address the large memory access and footprint posed by the vast model scale and attention mechanism, and mainly aim at enhancing the throughput rather than the end-to-end latency. As SoT trades off throughput for end-to-end latency, *SoT can make these throughput-oriented techniques help with end-to-end latency*. This interesting synergy offers opportunities for achieving better trade-offs between latency and throughput in future serving systems.

*In contrast to model- and system-level techniques, SoT is a data-level technique in a new "content co-organization for efficiency" paradigm.* See § 6 for more discussions.

**Efficient LLM methods through parallel generation.** Some prior work also addresses the sequential decoding issues. Speculative decoding (SD) methods (Stern et al., 2018) employ smaller models to generate some consecutive tokens sequentially and apply the target LLMs to verify them parallelly. Non-autoregressive generation (NAG) methods (Gu et al., 2018; Xiao et al., 2023) sample and refine consecutive tokens parallelly, often with the support of a modified and tuned model.

Relying on either assisting models or special models and sampling schemes, SD and NAG methods conduct *parallel verification or sampling and refinement of consecutive tokens*. In contrast, SoT prompts the LLM *itself* to plan the contents in a way that permits *the parallel generation of tokens in different segments*, by exploiting the emerging instruction-following and planning ability of LLMs.

**Prompting methods for LLMs.** Recent years have witnessed the emergence of the "pre-train, prompt, and predict" paradigm, which has shown promise in enhancing LLMs' quality in math and commonsense reasoning (Wei et al., 2022; Kojima et al., 2022; Wang et al., 2022; Chen et al., 2022) and planning for multi-modality tasks (Shen et al., 2023; Zhu et al., 2023). Instead of focusing on answer quality, *SoT is a first attempt at exploiting the power of prompting to improve efficiency*.

## 6 LIMITATIONS, FUTURE WORK, AND OPEN QUESTIONS

**Answer quality evaluation.** Our answer quality evaluation is far from perfect due to the limited prompt set, the potential bias of GPT-4 judges, and the inherent difficulty of evaluating LLM generations. Currently, we did not conduct human evaluation since it is easy for a human to tell whether an answer is generated with SoT due to its distinctive pattern, which might cause evaluation bias.

**Eliciting or improving LLMs' ability.** § 3.2.4 demonstrates SoT's potential of enhancing answer quality. It is part of a broader trend in recent research, exemplified by work including CoT (Kojima et al., 2022; Wei et al., 2022), ToT (Yao et al., 2023), and ReAct (Yao et al., 2022), which collectively affirm the notion that *explicitly articulating the thought process in language can elicit high-quality answers from LLMs*. These findings resemble human thinking: rather than relying solely on the first intuition or purely sequential thinking, we often document step-by-step reasoning or thought organization to attain high-quality answers. This intriguing parallel prompts us to explore further how we can draw from the human thinking process to facilitate more effective and efficient AI.

For instance, SoT currently ignores the dependencies between points. A conceptually better way is to organize the points as *Graph-of-Thoughts*, where the edges represent the dependencies, and each point is decoded conditioned on the contents of its ancestor points. In addition, instead of complying with a *static* graph, we expect the need of having *dynamic Graph-of-Thoughts*, where the high-level thought structure is adjusted dynamically by LLMs themselves. This could potentially combine the efficiency and global thinking advantages of SoT with the logical reasoning and impromptu thinking strengths of methods like CoT (Kojima et al., 2022; Wei et al., 2022). Notably, a contemporary work (Besta et al., 2023) has attempted to design Graph-of-Thoughts to elicit reasoning. Furthermore, it is interesting to explore how the SoT answers can be used to fine-tune LLMs to generate more structured answers in a self-improving way (Zelikman et al., 2022; Huang et al., 2022).

**Efficiency and overhead of SoT in different scenarios.** Serving systems commonly adopt batch processing to handle concurrent queries. This raises a concern of whether SoT may hurt serving throughput due to parallel requests. (1) When there is an unsaturated number of concurrent queries, SoT can effectively reduce latency and enhance GPU utilization. Example scenarios include (a) Edge-side applications with a single user; (b) Centralized services during periods with unsaturated user requests and underutilized computing capacity. It is interesting to study the appropriate SoT triggering conditions based on system workloads. (2) When there is a saturated number of concurrent queries, SoT is still useful for improving answer quality. However, in this case, it is important to consider the computation overhead from SoT. We delve into this concern in App. H.

For API-based models, a notable concern arises regarding the increased number of prefilling tokens (App. H). Given that many APIs charge token usage, SoT may lead to higher costs. To address this, one can use prompt tuning to design shorter SoT prompts (Jiang et al., 2023).

**Data-centric efficiency optimization.** While data-centric engineering for improving answer *quality* (Zha et al., 2023; HazyResearch, 2023) is gaining popularity, its potential for *inference efficiency* is not explored yet. SoT is the first attempt. As LLM capabilities and the amount of LLM-generated data are growing rapidly, data-centric techniques could become more useful in the future. To pave the way towards that, there are a lot to explore. For example, the acceleration ratio of SoT depends on the SoT prompt, the model, and the question, and thus not as predictable and controllable as model- or system-level techniques, which might hinder the practical adoption. We look forward to future work to unlock the full potential of data-centric efficiency optimization.

## ACKNOWLEDGEMENTS

We thank Sergey Yekhanin (Microsoft Research), and Tianji Wu (Infinigence AI) for their support and suggestions. This work is also supported by National Natural Science Foundation of China (No. 62325405, 62104128, U19B2019, U21B2031, 61832007, 62204164), Tsinghua EE Xilinx AI Research Fund, and Beijing National Research Center for Information Science and Technology (BNRist). We thank Tianyu Fu for initial discussions on the idea. We thank Ke Hong and Genghan Zhang for discussions about profiling. We thank Yue Wu for the help on the Claude scripts. We thank Da Yu, Chulin Xie, and Saiqian Zhang for their suggestions on revising the first version of the paper. We thank Rui Hu, Cheng Cheng, Jack Jin, Zhoutong Ye, Mingze Sun, Jun Yan, Zhi Zhang, Yuxuan Tong, Nianhui Guo, and Andrea Santilli for their suggestions on revising the second version of the paper. We thank Chris Stetkiewicz, Amanda Melfi, and Amber Tingle from Microsoft for their suggestions and help on writing. We acknowledge support from the Research Foundation - Flanders (FWO) through project numbers G0A1319N and S001421N, and funding from the Flemish Government under the Onderzoeksprogramma Artificiële Intelligentie (AI) Vlaanderen programme. We thank the anonymous reviewers for their insightful questions and suggestions.

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

# Appendix

## Table of Contents

## A  MODEL DETAILS

Table 1 summarizes the models on which we evaluate SoT. We use GPT-4 in the main paper and ChatGPT-3.5 in App. I.4 as the judge in FastChat and LLMZoo evaluation.

Table 1: Models evaluated with SoT. All the open-source models are fine-tuned from LLaMA models.

| Access | Model Name | Institution | Released Date |
|---|---|---|---|
| Open-Source | LLaMA2-Chat-7B (Touvron et al., 2023b) | Meta & Microsoft | 2023/07 |
| | LLaMA2-Chat-13B (Touvron et al., 2023b) | Meta & Microsoft | 2023/07 |
| | OpenChat-13B (Wang et al., 2023a) | Tsinghua | 2023/07 |
| | Vicuna-7B V1.3 (Chiang et al., 2023) | LMSYS | 2023/06 |
| | Vicuna-13B V1.3 (Chiang et al., 2023) | LMSYS | 2023/06 |
| | Vicuna-33B V1.3 (Chiang et al., 2023) | LMSYS | 2023/06 |
| | StableVicuna-13B (Phung, 2023) | CarperAI | 2023/05 |
| | UltraLM-13B (Ding et al., 2023) | OpenBMB & Tsinghua | 2023/05 |
| | Vicuna-7B V1.1 (Chiang et al., 2023) | LMSYS | 2023/03 |
| API-Based | Claude (Anthropic, 2023) | Anthropic | 2023/05 |
| | ChatGPT-3.5 | OpenAI | 2022/11 |
| | GPT-4 | OpenAI | 2023/03 |

Table 2 shows sources of the models we use in the paper.

Table 2: The Hugging Face or API endpoints of the models.

| Access | Model Name | Hugging Face or API Endpoints |
|---|---|---|
| Open-Source | LLaMA2-Chat-7B (Touvron et al., 2023b) | meta-llama/Llama-2-7b-chat-hf |
| | LLaMA2-Chat-13B (Touvron et al., 2023b) | meta-llama/Llama-2-13b-chat-hf |
| | OpenChat-13B (Wang et al., 2023a) | openchat/openchat |
| | Vicuna-7B V1.3 (Chiang et al., 2023) | lmsys/vicuna-7b-v1.3 |
| | Vicuna-13B V1.3 (Chiang et al., 2023) | lmsys/vicuna-13b-v1.3 |
| | Vicuna-33B V1.3 (Chiang et al., 2023) | lmsys/vicuna-33b-v1.3 |
| | StableVicuna-13B (Phung, 2023) | CarperAI/stable-vicuna-13b-delta[2] |
| | UltraLM-13B (Ding et al., 2023) | openbmb/UltraLM-13b[2] |
| | Vicuna-7B V1.1 (Chiang et al., 2023) | lmsys/vicuna-7b-delta-v1.1 |
| API-Based | Claude (Anthropic, 2023) | Claude extension on Slack[3] |
| | ChatGPT-3.5 | Azure OpenAI, gpt-35-turbo 0301 version[4] |
| | GPT-4 | OpenAI, gpt-4-0613 version |

## B  IMPLEMENTATION DETAILS OF SKELETON-OF-THOUGHT

### B.1  PROMPT

The skeleton prompt is shown in Prompts 1 and 3 and the point-expanding prompt is shown in Prompt 2.

**Skeleton prompt template.** In order to make the output skeleton short and in a consistent format for the good of efficiency and ease of point extraction, the skeleton prompt template (1) describes the

---

[2]For convenience, we use the non-official endpoint TheBloke/stable-vicuna-13B-HF and TheBloke/UltraLM-13B-fp16 to get merged weights.

[3]https://www.anthropic.com/claude-in-slack

[4]https://azure.microsoft.com/en-us/products/ai-services/openai-service

---

**Prompt 3. Skeleton Prompt Template $T^s$ (with Two-Shot Demonstrations)**

**[User:]** You're an organizer responsible for only giving the skeleton (not the full content) for answering the question. Provide the skeleton in a list of points (numbered 1., 2., 3., etc.) to answer the question. Instead of writing a full sentence, each skeleton point should be very short with only 3~5 words. Generally, the skeleton should have 3~10 points.

Question:
What are the typical types of Chinese dishes?
Skeleton:
1. Dumplings.
2. Noodles.
3. Dim Sum.
4. Hot Pot.
5. Wonton.
6. Ma Po Tofu.
7. Char Siu.
8. Fried Rice.

Question:
What are some practical tips for individuals to reduce their carbon emissions?
Skeleton:
1. Energy conservation.
2. Efficient transportation.
3. Home energy efficiency.
4. Reduce water consumption.
5. Sustainable diet.
6. Sustainable travel.

Now, please provide the skeleton for the following question.
*{question}*
Skeleton:
**[Assistant:]** 1.

---

task precisely, and (2) provides a partial answer "1." for the LLM to continue writing. The skeleton responses are in the desired format in most cases. Therefore, we can use a simple regular expression `(\d+)\.\s?([\s\S]+?)(?=\n|\n*$)` to extract point indexes and point skeletons from the skeleton response.

We find that GPT-4 can work well without the two demonstrations in the skeleton prompt. Therefore, we do not include the two demonstrations for GPT-4 (Prompt 1). For all other models, the two demonstrations are included, as shown in Prompt 3.

**Point-expanding prompt template.** It describes the point-expanding task and provides a partial answer. We also provide instructions "Write it **very shortly** in 1~2 sentence" so that the LLMs keep the answers concise. Unlike the skeleton prompt template, we find that demonstrations are not necessary to get reasonable results.

We find that Claude and GPT-4 follows the instruction "Write it **very shortly** in 1~2 sentence and do not continue with other points!" in Prompt 2 very well, so that the answers are very short. Therefore, we delete "**very shortly**" from the prompt template in Claude and GPT-4.

**Partial answer.** In the Prompts 1 and 2, we provide partial answers so that LLMs can follow the desired response format better.

We can put the partial answer at the end of the prompt for the open-source models to continue writing. An implementation detail is that different open-source models have different conversation templates (i.e., different ways to combine user and assistant messages into one string). For example, Vicuna (Chiang et al., 2023) uses the string "USER:" and " ASSISTANT:" for the placeholder "**[User:]**" and "**[Role]**" in the Prompts 1 and 2, respectively, while UltraLM (Ding et al., 2023) uses "User:" and "⟨/s⟩Assistant:". We build our open-source model experiments with the help of the FastChat codebase (Zheng et al., 2023), in which the conversation templates of many models are already handled correctly. We implement the conversation templates of OpenChat-13B, StableVicuna-13B, and UltraLM-13B according to their official guides and codes.

For ChatGPT-3.5, we provide partial answers as a last message in the chat history from the assistant. Note that it is not a documented approach. We find it works well in most cases, in that ChatGPT-3.5

---

**Prompt 4. LLM Prompting as the Router**

[**User:**] Question: {*question*}

How would you like to answer the question?
A. Organize the answer as a list of points or perspectives (in the format of 1., 2., 3., etc.), and the points or perspectives can be answered independently without referring to the contents of the previous points.
B. Organize the answer as a list of points or perspectives (in the format of 1., 2., 3., etc.), and the contents of later points or perspectives cannot be answered independently without referring to the contents of the previous ones.
C. Do not organize the answer as a list of points or perspectives.

Just say A, B, or C. Do not explain. Do not provide an answer to the question.
[**Assistant:**]

---

continues the texts from the provided partial answer. However, in some rare cases, ChatGPT-3.5 repeats the provided partial answers.

For Claude over Slack, there is no obvious way to give the API a partial answer. We resort to modifying the prompt template slightly by adding

> *Please start your answer from "{partial answer}" and do not output other things before that*

at the end. We find that Claude understands and obeys it well. For GPT-4, we also take this approach.

**System Message.** We do not include the system message in the prompts for open-source models except LLaMA2.

The partial answer, "**very shortly**", and the 2-shot demonstrations discussed above are the only differences between the prompts we used across all models and all evaluations.

## B.2 SUPPORTING MULTI-ROUND CONVERSATION

To use SoT in a multi-round conversation, we can just put the question and the final aggregated answer in the history, removing all the SoT prompts. In this way, using SoT in one conversation round will not introduce additional prefill cost in future rounds.

## C IMPLEMENTATION DETAILS OF SKELETON-OF-THOUGHT WITH ROUTER

### C.1 PROMPTING ROUTER

We use Prompt 4 for querying GPT-4 as the router. If the answer is "A" (i.e., the question can be answered in a list of independent points), we will use SoT. Otherwise, if the answer is "B" (i.e., the answer is in a list of points but they depend on each other) or "C" (i.e., the answer should *not* be in a list of points), SoT is not suitable and we will fall back to normal decoding.

### C.2 TRAINED ROUTER

We tackle the routing problem as a sequence classification task. We first annotate the LIMA training set (Zhou et al., 2023), and then fine-tune a RoBERTa model (Liu et al., 2019) using the labeled data. Finally, we apply the tuned RoBERTa as the router on Vicuna-80 and WizardLM. We detail the steps in the following.

#### C.2.1 ANNOTATION PROCESS

In the classification task, a label of 1 (positive) indicates that this question can be answered with SoT, while a label of 0 (negative) suggests that using the normal generation mode is more suitable. We annotate the LIMA training set, which consists of 1,030 Q&As sourced from three community webpages: Stack Exchange, wikiHow, and the Pushshift Reddit. We also annotate the Vicuna-80 and WizardLM datasets for evaluation.

Table 3: Router confusion matrices on the Vicuna-80 dataset. **Left:** Rows are human annotations (H) and columns are the GPT-4 router (G). **Middle:** Rows are human annotations (H) and columns are the RoBERTa router (R). **Right:** Rows are the GPT-4 router (G) and columns are the RoBERTa router (R).

|    | G0 | G1 |    | R0 | R1 |    | R0 | R1 |
|----|----|----|----|----|----|----|----|----|
| H0 | 38 | 5  | H0 | 37 | 6  | G0 | 34 | 4  |
| H1 | 0  | 37 | H1 | 5  | 32 | G1 | 8  | 34 |

Table 4: Router confusion matrices on the WizardLM dataset. **Left:** Rows are human annotations (H) and columns are the GPT-4 router (G). **Middle:** Rows are human annotations (H) and columns are the RoBERTa router (R). **Right:** Rows are the GPT-4 router (G) and columns are the RoBERTa router (R).

|    | G0 | G1 |    | R0  | R1 |    | R0 | R1 |
|----|----|----|----|-----|----|----|----|----|
| H0 | 94 | 66 | H0 | 135 | 25 | G0 | 93 | 4  |
| H1 | 3  | 55 | H1 | 31  | 27 | G1 | 73 | 48 |

We use GPT-4 to assist the annotation process. Specifically, we present each question to GPT-4 and analyze its answer to determine whether SoT can be triggered for this question. We assign a positive label to a question if GPT-4's response meets two criteria: (1) it contains a list of points that can be expanded in parallel, (2) each point provides sufficient details (i.e., the point-expanding response is not too short), which will enable SoT to achieve a speed-up. Two of the paper's authors conduct the annotation process independently, and discuss the inconsistent annotations to decide the final label.

### C.2.2 TRAINING DETAILS

We use `roberta-base` with 120M parameters as the router model. The finetuning is conducted using the AdamW optimizer (Loshchilov & Hutter, 2019) with a weight decay of 0.01. The learning rate undergoes a warm-up phase during the first 1% of iterations to 5e-5 and then decays linearly. We train the model for 2 epochs using a batch size of 32. Input sequences are either padded or truncated to achieve a consistent length of 512 tokens.

In the application of SoT, false positives (SoT is incorrectly triggered when it should not be, resulting in degraded answer quality) are of more significant concern than false negatives (the router misses a potential SoT trigger, resulting in a reduced speed-up). Thus, to mitigate false positives, we employ the Tversky loss (Wang et al., 2023b) with parameters $\alpha = 0.7$ and $\beta = 0.3$, which penalizes false positives more heavily than false negatives. We also incorporate label smoothing (Szegedy et al., 2016) with a factor of $\epsilon = 0.2$. Overall, the entire fine-tuning process is efficient, completing in 2 minutes on an NVIDIA A100 GPU.

### C.3 ROUTER CONSISTENCY

We present the confusion matrices for the three routers to illustrate their consistency. The results on Vicuna-80 and WizardLM are shown in Tables 3 and 4, respectively.

On Vicuna-80, we can observe a notable level of agreement among the three routers. Compared with the GPT-4-prompting router, the trained router exhibits a slightly higher number of false negatives w.r.t. the human annotations. Conversely, on WizardLM, given the intricate answer structure and the presence of many ambiguous cases, the routers show significant discrepancies. Specifically, the GPT-4 router produces many false positives, which pose adverse affects on the answer quality (see App. I.2). The RoBERTa router aligns more closely with the human annotations.

### C.4 CONCURRENT EXECUTION FOR SoT-R

In SoT-R, the router serves as an additional stage that extends the two-stage SoT pipeline, as illustrated in Fig. 9. To push the limit of latency optimization, we can run the router, normal generation, and SoT generation concurrently. Once the router makes a decision, one of the normal and SoT generation processes can be aborted. However, this approach will increase the token overhead. Therefore, we did not employ this approach in this work and leave it to future work.

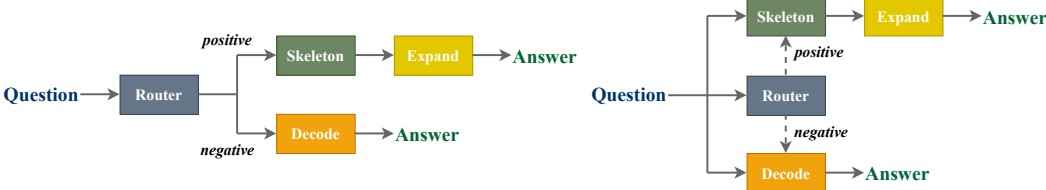

Figure 9: **Left:** The SoT-R pipeline. **Right:** A possible approach to further reduce latency at the cost of token overhead.

# D    SoT In the Context of Literature (Expanded)

## D.1    Efficient LLMs

Extensive research has been dedicated to enhancing the throughput and latency of LLM inference. We first discuss model-level architecture design or compression techniques. These techniques change the model and can benefit both the latency and throughput but require finetuning to retain the model quality. Then, we discuss system-level efforts that optimize the computational graph or the assignment and scheduling of the computational graph on computation and storage devices. Most system-level efforts accelerate the prefilling phase or focus on improving the throughput. Finally, we discuss some research efforts that share a similar motivation to ours, namely, addressing the efficiency issue of sequential decoding.

**Model-level optimization.**    Considerable architectural design efforts have emerged to (1) improve the scalability w.r.t. model size by introducing mixture-of-expert inference (Lepikhin et al., 2021; Fedus et al., 2022), (2) address the quadratic complexity w.r.t. input size of attention by designing new attention mechanisms (Kitaev et al., 2020; Wang et al., 2020), (3) reduce the memory access and footprint of attention by using multi-query attention (Shazeer, 2019), and so on. However, these methods usually require a substantial re-training cost. The model compression techniques require a smaller amount of fine-tuning by reducing the model complexity of a pre-trained LLM from certain aspects (Ganesh et al., 2021). Representative techniques include quantization (Xiao et al., 2022; Frantar et al., 2022; Lin et al., 2023), the static or dynamic pruning of weights, activation, and attention (Mishra et al., 2021; Zaheer et al., 2020; Wang et al., 2021; Chen et al., 2023b), and so on.

Zooming out from LLM compression to the whole field of model compression, we can see that model co-design or compression for efficiency has received tremendous attention in the past few years and has grown into large research fields, such as pruning (Han et al., 2015; Wen et al., 2016), quantization (Krishnamoorthi, 2018), factorization (Denton et al., 2014), and neural architecture search (Zoph & Le, 2017; Elsken et al., 2019; Cai et al., 2019). *Different from the model co-design paradigm, SoT is in a "**content co-organization for efficiency**" paradigm for improving the LLM efficiency.* Along with the growth in the LLM capabilities and amount of LLM-generated data, data-level techniques could become important tools in the efficient LLM toolbox.

**System-level optimization.**    In the realm of lossless acceleration, considerable efforts have been devoted to addressing the I/O-bound nature of LLMs on modern hardware platforms (Dao et al., 2022). Numerous studies (Dao et al., 2022; Zhai et al., 2022; Ivanov et al., 2021; NVIDIA, 2019) have focused on adjusting the computational graph by fusing and implementing operations in an I/O-friendly way. As a representative method, FlashAttention (Dao et al., 2022) fuses all operations of one attention into one GPU kernel with spatially tiled computation to reduce the off-chip I/O of the attention map. While FlashAttention can effectively accelerate training and the prefilling phase of inference, it cannot accelerate the decoding phase much (when the batch size is small), as it is the I/O of weights rather than activation or attention map that bottlenecks the decoding phase. For example, when the context length is 64, decoding one token using LLaMA-7B needs to load each of the 7B parameters from the off-chip HBM onto the GPU chip at least once, but only transferring about 20M (0.02B) activation values between the off-chip HBM and GPU chip.

In order to satisfy Service Level Objectives, serving systems focus on improving the serving throughput under latency constraints. To this end, serving systems (Fang et al., 2021; NVIDIA,

2021; Google, 2021) pack multiple queries together into a batch to improve the hardware utilization. The batching technique has proven highly effective in enhancing throughput, leading to the development of various variants. For example, some work designs methods to decide which queries to batch together (Fang et al., 2021; Zhou et al., 2022), while others selectively batch parts of the model to enable fine-grained iteration-level batching (Yu et al., 2022) or multi-task batching (Zhou et al., 2022). Various model parallelism (Lu et al., 2017; Huang et al., 2019; Narayanan et al., 2019; Rajbhandari et al., 2020; Narayanan et al., 2021; Li et al., 2021; Zheng et al., 2022) and offloading (Ren et al., 2021; Sheng et al., 2023) techniques have been proposed to maximize the throughput of LLM training or inference. In a nutshell, given the computational graph and device configurations, these techniques optimize the split, assignment, and scheduling of computations, storage, and communications on devices. In addition to the model parallelism and batching techniques, an efficient memory management mechanism for LLM workloads is also an essential feature in the serving systems (Kwon et al., 2023; SenseTime, 2023a;b).

To sum up, these system-level techniques mainly help with the throughput in training and batched inference. They can be used by SoT to improve the throughput of the batched decoding of multiple segments. This means that *SoT can harness the power of these throughput-oriented techniques and make them help with the end-to-end latency*, offering a new dimension for better trading off latency and throughput in future serving systems.

Another parallelism perspective to position SoT is that *SoT guides the LLM to adjust the sequential workload to become "inter-content" parallelizable*, which differs from the parallelism levels in existing serving systems, including inter-instance (Krizhevsky, 2014; Rajbhandari et al., 2020), inter-operation (Huang et al., 2019; Narayanan et al., 2019; 2021), intra-operation (Xu et al., 2021), and inter-token (Li et al., 2021). It may be worthwhile to explore *the integration of SoT into serving systems to maximize the hardware utilization.*

**Decoding optimization.** One bottleneck for the end-to-end latency lies in the autoregressive decoding phase, where tokens must be generated one by one. Due to the dependency between tokens, the computation of different tokens cannot be parallelized, causing severe under-utilization of GPU. In order to improve the end-to-end decoding latency of a given LLM, speculative decoding methods (Stern et al., 2018; Leviathan et al., 2022; Chen et al., 2023a; Gante, 2023; Sun et al., 2023; Miao et al., 2023) propose to use cheaper approaches to generate short candidate token sequences, for example, by sequentially decoding with an assisting model much smaller than the given LLM. Then, they use the LLM to parallelly verify the candidates and keep the prefix sequence that matches the LLM's verification results.

Another line of work that shares the motivation of addressing the autoregressive efficiency issue is non-autoregressive generation (NAG) methods (Gu et al., 2018; Xiao et al., 2023). NAG methods sample consecutive tokens parallelly, often with the aid of a modified and tuned model. To maintain the answer quality, instead of sampling for one iteration, many NAG methods refine the output parallelly for multiple iterations (Xiao et al., 2023; Santilli et al., 2023).

To summarize, the speculative decoding methods use assisting models for *letting the LLM conduct parallel verification of consecutive tokens*, and the NAG methods rely on specially designed models, training schemes, or sampling schemes for *the parallel sampling and refinement of consecutive tokens*. In contrast, SoT prompts the LLM itself to plan the contents in a way that permits *the parallel generation of multiple tokens in different segments*. SoT exploits the emerging instruction-following and planning ability of SoTA LLMs rather than relying on specially designed modeling, sampling, and training schemes. This is different from all existing work that targets the autoregressive efficiency issue.

## D.2 PROMPTING METHODS FOR LLMS

In recent years, the "pre-train, prompt, and predict" paradigm has emerged (Liu et al., 2023), which designs prompts comprising task descriptions and (optionally) a few demonstrations to guide pre-trained LLMs in generating answers for a wide range of downstream tasks. Researchers found that instruction-tuned LLMs (Brown et al., 2020; Wei et al., 2021; Ouyang et al., 2022; Chung et al., 2022; Taori et al., 2023) possess a strong ability to (1) generalize to new tasks thanks to the diverse

natural language descriptions encountered during instruction tuning, and (2) learn in-context using a few demonstrations without weight tuning.

In virtue of these abilities, the field has been manually engineering (Brown et al., 2020; Kojima et al., 2022; Shen et al., 2023; Li et al., 2023a), automatic searching (Shin et al., 2020), or continuously tuning (Li & Liang, 2021; Lester et al., 2021) the prompts for uncovering the capabilities of LLMs on downstream tasks. There are a bunch of prompting methods that improves the reasoning performance of LLMs by designing thinking flows mimicking human reasoning: (1) mimicking the step-by-step or compositional thinking structure (Wei et al., 2022; Kojima et al., 2022; Press et al., 2022; Yao et al., 2023; Besta et al., 2023; Zhang et al., 2023), (2) designing multiple reasoning paths and their aggregation (Wang et al., 2022; Yao et al., 2023; Li et al., 2023c), and (3) using tools for calculation and information retrieval (Chen et al., 2022; Yao et al., 2022; Schick et al., 2023). As a representative example, the Chain-of-Thought prompts largely improve the performance on tasks that require logical reasoning by simply providing a "Let's think step by step" (Kojima et al., 2022) instruction or a few demonstrations (Wei et al., 2022). Another topic that arises quite a surge of interests is to prompt LLMs to help finish complex multi-modality task (Shen et al., 2023; Zhu et al., 2023). For example, HuggingGPT (Shen et al., 2023) design prompts to guide the LLM to generate structural JSON for the orchestration of multi-model execution to finish complex tasks.

To summarize, the large literature on prompting methods has been aiming at uncovering different capabilities of LLM and improving the answer quality on different downstream tasks. In contrast, *SoT is a first attempt at exploiting the power of prompting to improve efficiency*.

### D.3 HIERARCHICAL TEXT GENERATION

SoT can be regarded as being "hierarchical" since it has high-level answer structure planning. Prior studies in hierarchical text generation (Li et al., 2015; Shao et al., 2019; Puduppully et al., 2019; Fan et al., 2018) all focus on enhancing the answer quality, including improving the long-range coherence, relevance to the topic, or reducing redundancy. These methods craft hierarchical neural architectures that contain different modules to model high-level (sentence-level or document-level) and low-level (word-level) dependencies (Li et al., 2015; Shao et al., 2019; Fan et al., 2018). They still employ sequential word-by-word generation without parallelization between sentences.

Note that the sentence-level representations in previous work (Li et al., 2015; Shao et al., 2019) are "implicit" latent variables instead of "explicit" language descriptions. Some previous studies (Shao et al., 2019; Puduppully et al., 2019) train a dedicated planning module to execute explicit content planning in advance. Nevertheless, these methods all conduct "closed-form" planning that only reorders and groups the input keywords, rather than producing "free-form" plans on "what to say" and "how to say". All the hierarchical architectures and planning modules require training or even special data processing (Puduppully et al., 2019).

To summarize, in terms of the objective, the primary focus of SoT – efficient generation – is different from previous hierarchical text generation literature. In terms of the methodology, instead of designing new hierarchical architectures or planning modules, SoT exploits the emerging planning and instruction-following abilities of LLMs to do explicit (which means the plan is described by interpretable language) and free-form planning. This allows SoT to be applied to off-the-shelf LLMs for producing structured answers.

As the hierarchical text generation literature focuses on enhancing answer quality, they could provide inspiration for future expansions of SoT to generate high-quality answers for broader types of questions.

## E EFFICIENCY ANALYSIS

This section gives a detailed explanation on why SoT can reduce the overall decoding latency with the same computational resource for local models.

The vanilla approach processes only one question and decodes the answers sequentially, whereas SoT processes multiple point-expanding requests and the answers in a batch. We focus on the following question: "Compared to processing only one sequence, how much peak memory overhead and latency increase will be brought by processing a batch of sequences?"

Table 5: The latency and average GPU performance of the prefilling and decoding phases when inferencing LLMs. The prefilling token length is 128, the decoding token length is 64, and the batch size is 1. The test is run on one NVIDIA A100 GPU.

| Model | Prefill/Decode Latency (ms) | Prefill/Decode GPU Perf. (TFLOPS) |
|---|---|---|
| LLaMA-7B | 40 / 2735 | 43 / 0.31 |
| LLaMA-13B | 54 / 3725 | 62 / 0.44 |
| LLaMA-33B | 100 / 5506 | 85 / 0.75 |

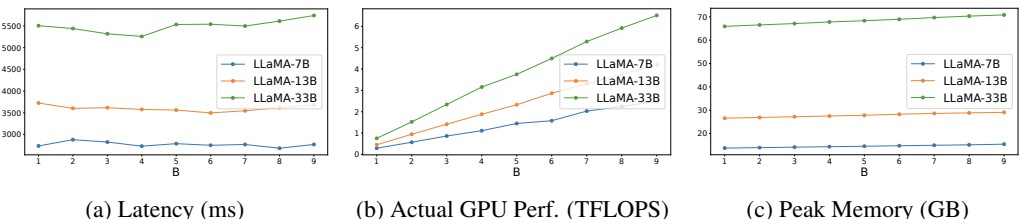

(a) Latency (ms)  (b) Actual GPU Perf. (TFLOPS)  (c) Peak Memory (GB)

Figure 10: The trends of latency, average GPU performance of decoding one token, and peak memory with respect to the batch size $B$ of sequences. The prefilling token length is 128, and the decoding token length is 64. The test is run on one NVIDIA A100 GPU.

A typical LLM generative process consists of two phases: (1) the prefilling phase in which the prompt is parsed to generate the key-value cache for further use, and (2) the decoding phase in which tokens are generated one by one in a sequential manner. The decoding phase accounts for the majority of the end-to-end latency, especially when generating a long response. As shown in Table 5, when running Vicuna-7B on NVIDIA A100-80G, the actual computing performance is only 0.31 TFLOPS (0.1% utilization) in the decoding phase, compared to 43 TFLOPS (13.8% utilization) during prefilling. The utilization is calculated with respect to the FP16[5] tensor core peak performance – 312 TFLOPS for NVIDIA-A100. As a result, the latency of decoding only one token is comparable to that of prefilling 128 tokens (40ms). This huge gap in actual computing performance and thereby the latency arises from the fact that all LLM weights need to be loaded onto the GPU chip at least once only for decoding one token, so the decoding is heavily bottlenecked by the I/O of weights and the GPU computation units cannot be well utilized.

When conducting batched decoding, as the sequence batch size $B$ increases, the latency of decoding one token for each sequence stays roughly the same (Fig. 10a), as the amount of LLM weights that needs to be loaded onto the chip does not change. As a result, the GPU computation utilization ($\frac{\text{Actual GPU Performance}}{\text{Peak GPU Performance}}$) increases almost linearly as $B$ increases (Fig. 10b). In other words, for generating a final answer of length $N$, if we cut the answer into $B$ segments of length $N/B$ and decode them as a batch, we can get a $B\times$ decoding speed-up compared to sequential decoding. Nevertheless, in practice, as prefilling longer requests brings some overhead, and the lengths of the $B$ segments could be imbalanced, the actual speed-up of the batched point-expanding stage compared with the original prefilling and sequential decoding process is smaller than $B$.

As for the peak memory overhead, the amount of LLM weights can be one to two orders of magnitude larger than that of all the intermediate activations as long as the prefilling token length is not too large, not to mention that most activations do not need to be saved for back-propagation during inference. Therefore, the LLM weights account for the majority of the memory footprint in our test cases. Consequently, as shown in Fig. 10c, the peak memory overhead due to the increasing size of the KV cache and activation grows at a slow pace as the batch size $B$ increases. Thanks to the small peak memory overhead, in all of our experiments, we managed to use one GPU to run SoT without seeking help from other peak memory optimization techniques (e.g., quantization (Frantar et al., 2022; Lin et al., 2023), offloading (Sheng et al., 2023)).

---

[5] All of our experiments are run with FP16 inference.

## F    Efficiency Profiling

We run the profiling on the target GPU (NVIDIA A100-80G and NVIDIA RTX 3090) with CUDA 11.7, using the Hugging Face transformer library 4.28.1 and PyTorch 2.0.1. The host of A100-80G has an Intel Xeon Platinum 8358P CPU and 1T memory. The host of RTX 3090 has an Intel Xeon Gold 6246R CPU and 512G memory.

**Latency profiling and estimation.**    For the decoding phase, we denote $t_B^D(k)$ as the latency of batched decoding the $k + 1$-th token with batch size $B$, where the superscript $D$ stands for "decode". For each batch size $B = 1, \cdots, 16$ and each context length $k = 1, \cdots, 1024$, we use `torch.cuda.Event` to record the latency of decoding one token. We run each decoding three times continuously and take their geometric mean as $\{t_B^D(k)\}_{k=1,\cdots,1024;B=1,\cdots,16}$. For the prefilling phase, we profile the latency of batched prefilling the inputs with token length $k$ in range$(1, 700, 10)$ and batch size $B = 1, \cdots, 16$, and denote it as $t_B^P(k)$, where the superscript $P$ stands for "prefill". We run each test seven times continuously, regard the first two times as the warmup tests, and take the geometric mean of the last five times as $\{t_B^P(k)\}_{k=1,11,\cdots,691;B=1,\cdots,16}$. Once we get the latency profiling table, given a request with $l_i$ tokens and the decoding batch size $B$, the latency of generating $l_o$ tokens can be estimated as:

$$T(l_i, l_o, B) = \tilde{t}_B^P(l_i) + \sum_{k=l_i}^{l_i+l_o-1} t_B^D(k), \tag{1}$$

where the subscripts $i$ and $o$ stand for "input" and "output". Note that we only test the prefilling latency every ten token lengths (i.e., $1, 11, 21, \cdots$) for fast profiling and estimate $\tilde{t}_B^P(l_i)$ by $t_B^P(\lfloor \frac{l_i}{10} \rfloor \times 10 + 1)$.

The SoT decoding process consists of two stages: the skeleton stage and the point-expanding stage. Denoting the token length of the skeleton request and skeleton response as $l_i^s$ and $l_o^s$, the token length of the longest point-expanding request and the longest point-expanding response as $l_i^{pe}$ and $l_o^{pe}$, the number of the points as $B$, we can compute the latency of the skeleton and point-expanding stages as:

$$L^s(l_i^s, l_o^s) = T(l_i^s, l_o^s, 1), \tag{2}$$
$$L^{pe}(l_i^{pe}, l_o^{pe}, B) = T(l_i^{pe}, l_o^{pe}, B). \tag{3}$$

Using the latency profiling table, we can further estimate the average GPU computing performance in FLOPS (i.e., FLOPs per second) of decoding $l_o$ tokens with prefilling length $l_i$ as

$$P^D(l_i, l_o, B) = \frac{\sum_{k=l_i}^{l_i+l_o-1} f_B^D(k)}{\sum_{k=l_i}^{l_i+l_o-1} t_B^D(k)}, \tag{4}$$

where $f_B^D(k)$ denotes the FLOPs of decoding one token with context length $k$, which is calculated by DeepSpeed's FLOPs profiler [6]. Fig. 10b reports the average GPU computing performance during the process of decoding 64 tokens (prefilling length=128), i.e., $P^D(128, 64, B)$.

**Memory profiling and evaluation.**    To evaluate the peak memory, we use `torch.cuda.max_memory_allocated` to record the memory consumption of prefilling sequences of different lengths and decoding with different context lengths and a batch size ranging from 1 to 16. Then, we calculate the peak memory of each stage as the maximum value of the prefilling and decoding phases, and calculate the overall peak memory of SoT as the maximum value of the skeleton and point-expanding stages.

---

[6]https://deepspeed.readthedocs.io/en/latest/flops-profiler.html

# G EFFICIENCY EVALUATION

## G.1 SKELETON-OF-THOUGHT

### G.1.1 DETAILED STATISTICS OF TOKEN LENGTHS AND POINT NUMBERS

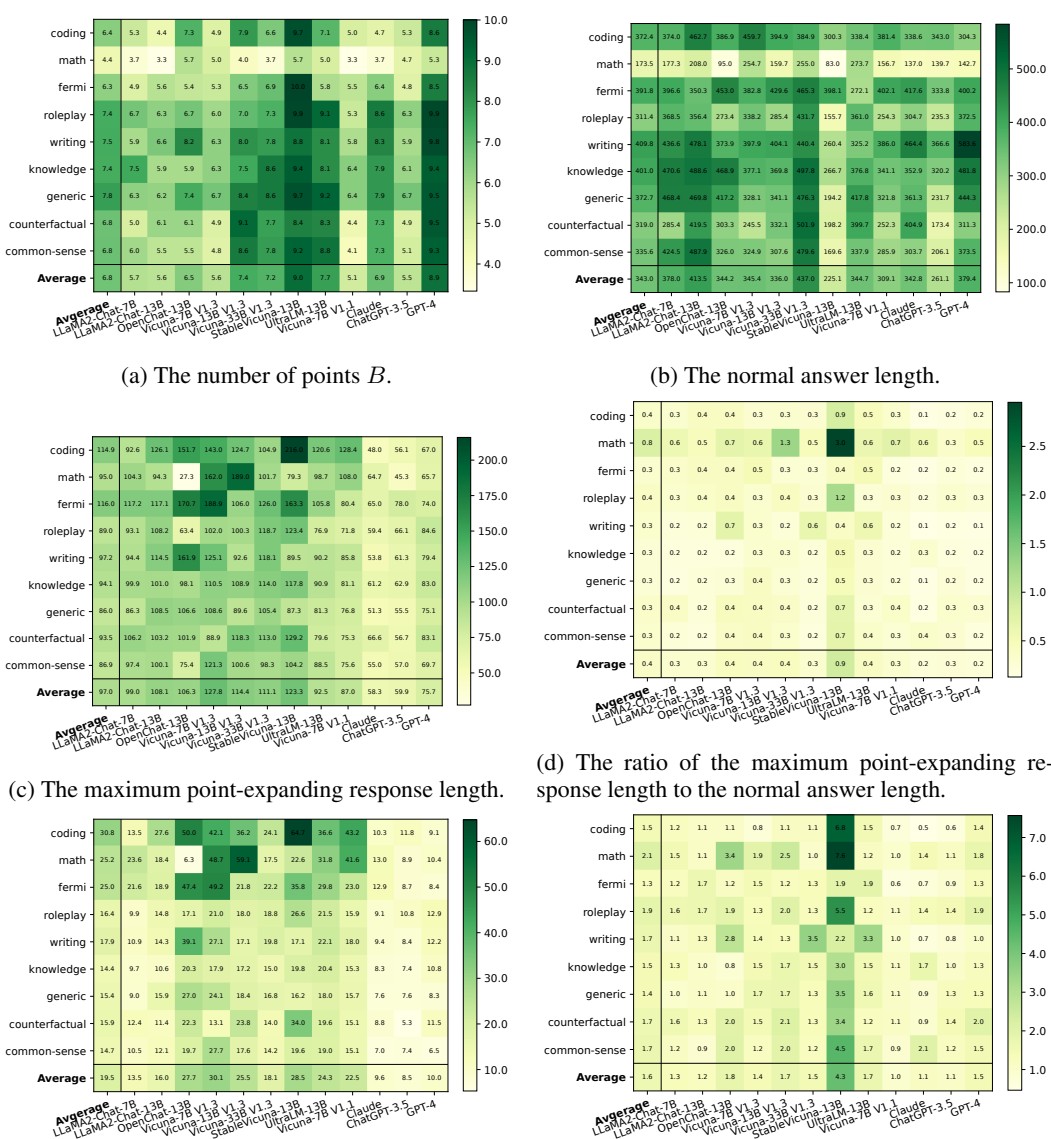

(a) The number of points $B$.

(b) The normal answer length.

(c) The maximum point-expanding response length.

(d) The ratio of the maximum point-expanding response length to the normal answer length.

(e) The imbalance degree of point-expanding response lengths (standard deviation of point token lengths).

(f) The ratio of the final SoT answer length to the normal answer length.

Figure 11: The statistics of the token lengths and point numbers on the Vicuna-80 dataset. Each row corresponds to one question category, and each column corresponds to one model.

### G.1.2 LATENCY BREAKDOWN: SOT STAGES AND PHASES

Fig. 12 presents the absolute latencies of normal and SoT generations on Vicuna-80. Again, the speed-ups of SoT compared with normal generation is evident. We can see that the decoding phases predominantly account for the end-to-end latency. Consequently, although SoT has higher prefilling latency in the skeleton stage than the normal generation and introduces additional point-expanding

prefilling latency – which is expected – this has negligible impact on the overall latency and thereby the overall speed-up.

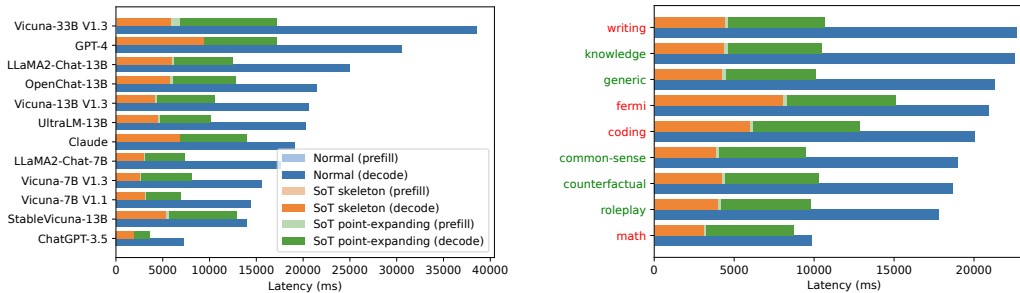

(a) Average latency across all question categories except *math* and *code* on different models.

(b) Average latency across all models on different question categories.

Figure 12: The latency breakdown of SoT and normal generations on the Vicuna-80 dataset. For open-source models, the latency breakdown of the prefilling and decoding phases is shown in different colors. For API-based models, we do not record such latency breakdown information; the bar labeled as "(decode)" indicates the overall latency of prefilling and decoding phases.

### G.1.3 EFFICIENCY EVALUATION ON NVIDIA RTX 3090

We present the SoT speed-ups and latency breakdown on RTX 3090 in Fig. 13. We test the three 7B models, as their FP16-precision version can be run on an RTX 3090 GPU without further peak memory optimization techniques such as weight quantization (Frantar et al., 2022; Lin et al., 2023) or offloading (Sheng et al., 2023). On these three models, SoT can obtain 1.94× to 2.40× speed-up on average on Vicuna-80.

For the five question categories that SoT can provide high-quality answers (i.e., *knowledge*, *common-sense*, *generic*, *roleplay*, *counterfactual*), SoT can speed-up the overall answer generation process by 1.96× to 2.52× in the meantime. Note that for the *math* category, despite the average speed-up being 1.20× by calculating the speed-up across the three math questions, SoT does not reduce the absolute latency of processing the three questions.

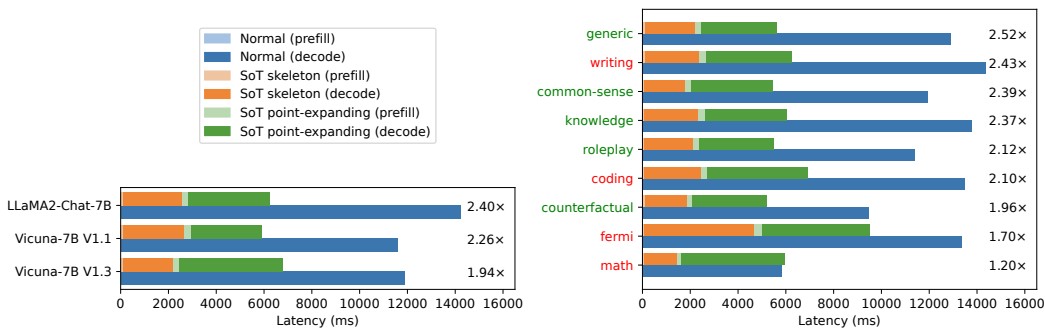

Figure 13: The latency breakdown of SoT and normal decoding on the Vicuna-80 dataset. The average speed-up across questions are also marked on the figure.

### G.1.4 ACTUAL LATENCY TESTING

This section reports the actual SoT speed-up on the Vicuna-80 with batch testing (instead of analyzing with pre-made profiling tables), using a single NVIDIA A100 GPU. We test the actual end-to-end latency of the SoT and normal decoding with the 9 open-source models. For each model, we run the speed-up test for five times and plot the box in Fig. 14.

As shown in Fig. 14a, the current SoT solution obtains a $> 2\times$ speed-up on 6 out of the 9 open-source models (i.e., Vicuna-7B V1.1, Vicuna-7B V1.3, UltraLM-13B, LLaMA2-Chat-7B, Vicuna-13B V1.3, and LLaMA2-Chat-13B), and a $> 1.7$ speed-up on OpenChat-13B and Vicuna-33B V1.3. SoT achieves no speed-up on StableVicuna-13B. As shown in Fig. 14b, for the five question categories that SoT can provide high-quality answers (i.e., *knowledge*, *common-sense*, *generic*, *roleplay*, *counterfactual*), SoT can speed-up the overall answer generation process by $2.15\times$ to $2.50\times$ in the meantime.

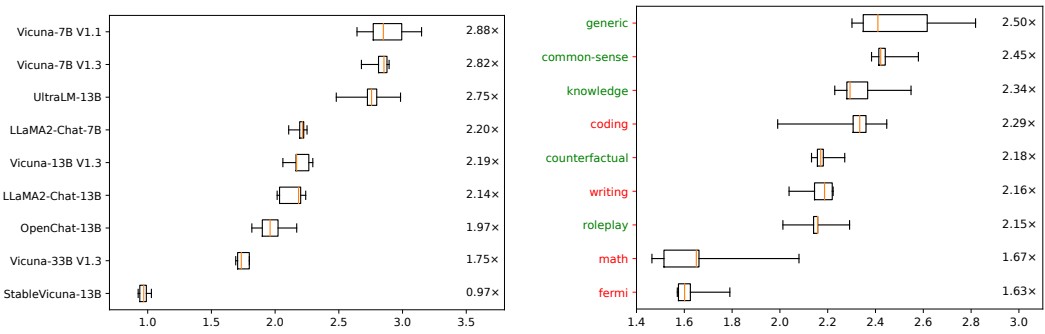

(a) Average speed-up on different models.     (b) Average speed-up on different question categories.

Figure 14: Speed-ups on 9 open-source models on the Vicuna-80 dataset with actual batch testing.

## G.2 SKELETON-OF-THOUGHT WITH ROUTER

The overhead brought by the router inference is relatively small: On the Vicuna-80 dataset, the prompting and trained router have an average latency of 0.65s (0.39s~1.37s) and 0.04s (0.008s~1.55s), respectively. On the WizardLM dataset, the average latency of the prompting and trained router is 0.80s (0.36s~2.22s) and 0.03s (0.009s~2.52s), respectively.

### G.2.1 SPEED-UP BREAKDOWN: MODELS

Fig. 15 shows the speed-ups of SoT-R on different models on the Vicuna-80 dataset. Fig. 16 and Fig. 17 show the speed-ups of SoT-R on different models on the WizardLM dataset. We can observe that on Vicuna-80, the two methods yield similar speed-ups, whereas on WizardLM, GPT-4 prompting router usually obtains higher speed-ups than the trained router, especially on GPT-4 itself.

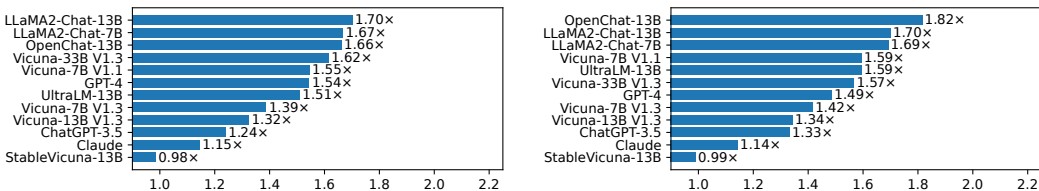

(a) Average speed-up across all question categories with prompting router.     (b) Average speed-up across all question categories with trained router.

Figure 15: Speed-ups of SoT-R on different models on Vicuna-80 dataset.

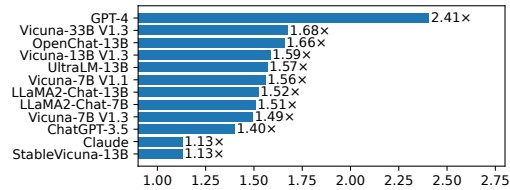 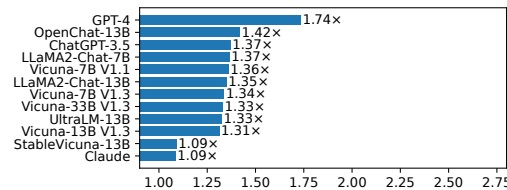

(a) Average speed-up across all question categories with prompting router.

(b) Average speed-up across all question categories with trained router.

Figure 16: Speed-ups of SoT-R on different models on WizardLM dataset.

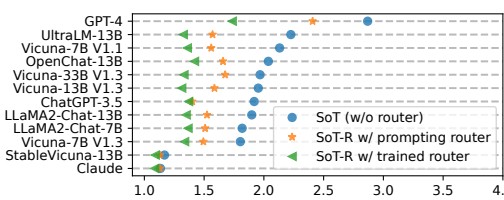

Figure 17: Speed-ups of SoT and SoT-R on different models on the WizardLM dataset.

### G.2.2 SPEED-UP BREAKDOWN: CATEGORIES

Fig. 18 and Fig. 19 show the speed-ups of SoT-R on different question categories of Vicuna-80 dataset. The trained router achieves slightly higher speed-up on most of the categories (except for *knowledge*, *writing*, and *fermi*). Fig. 20 and Fig. 21 show the speed-ups of SoT-R on different question categories of WizardLM dataset. We can observe that on 19 out of 29 categories, using the prompting router achieves higher speed-ups than using the trained router.

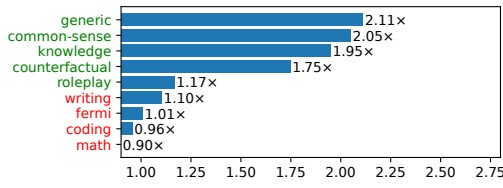 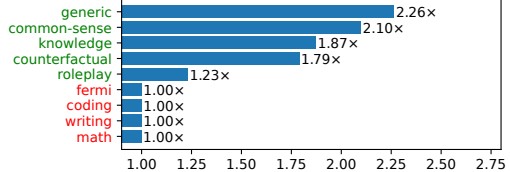

(a) Speed-ups of SoT-R with prompting router on different question categories.

(b) Speed-ups of SoT-R with trained router on different question categories.

Figure 18: Speed-ups of SoT-R on different question categories of Vicuna-80 dataset

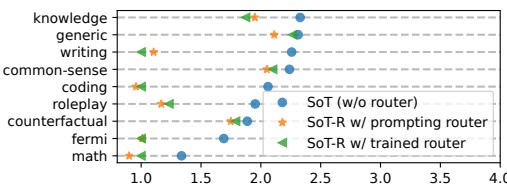

Figure 19: Speed-ups of SoT and SoT-R on different question categories of the Vicuna-80 dataset.

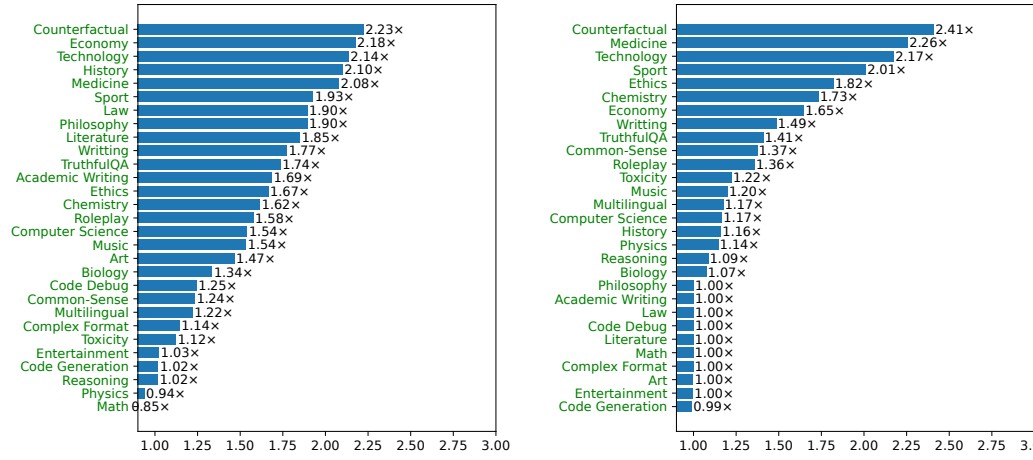

(a) Speed-ups of SoT-R with prompting router on different question categories.

(b) Speed-ups of SoT-R with trained router on different question categories.

Figure 20: Speed-ups of SoT-R on different question categories of WizardLM dataset

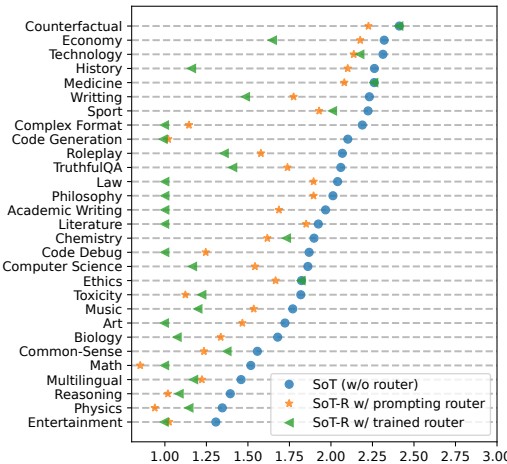

Figure 21: Speed-ups of SoT and SoT-R on different question categories of the WizardLM dataset.

## H    OVERHEAD OF SOT IN DIFFERENT SCENARIOS

Despite the optimizations made to the decoding phase, SoT brings overhead to the prefilling phase as the model needs to handle additional SoT prompts. Table 6 reports SoT's prefilling overhead for the API-based models. These statistics are averaged across the Vicuna-80 questions that are suitable for SoT (according to our manual annotation). We can see that SoT significantly increases the number of prefilling tokens. This is because that SoT issues an independent point-expanding request for each point, with the average number of points being 6.8 on Vicuna-80 dataset across all evaluated models. Consequently, the APIs need to prefill the point-expanding request multiple times.

Table 6: SoT's prefilling token overhead for API-based models.

| Model | Prefill Phase | | | |
|---|---|---|---|---|
| | Normal | SoT Stage 1 | SoT Stage 2 | Ratio (SoT / Normal) |
| Claude | 10.33 | 155.33 | 730.91 | 85.79 |
| ChatGPT-3.5 | 10.21 | 136.33 | 480.95 | 60.46 |
| GPT-4 | 10.21 | 72.44 | 838.26 | 89.20 |

When using SoT to serve the open-source models, a simple and small trick is to prefill the common prefix of point-expanding requests with a batch size of 1 during Stage 2 (i.e., the point-expanding stage). Table 7 shows the prefilling overhead after applying the trick. Although the ratio is considerably smaller compared to that of the API-based models, this computational overhead remains a concern, especially during periods of high system workload.

There are some possibilities to further reduce the token and computational overhead that are worth exploring in future work. To name a few: (1) When using SoT in serving systems, we can simply reuse the key-value cache containing the question and skeleton from Stage 1 during Stage 2, rather than re-prefilling them as in a multi-round conversation. (2) Generally, as LLM capabilities continue to evolve and prompt tuning techniques advance (Shin et al., 2020; Li & Liang, 2021; Lester et al., 2021; Jiang et al., 2023), the possibility of using much shorter prompts to activate the SoT mode in the future holds promise, which would significantly mitigate the token or computational overhead.

Table 7: SoT's computational overhead (in terms of the number of prefilling tokens) for open-source models.

| Model | Prefill Phase | | | |
|---|---|---|---|---|
| | Naive | SoT Stage 1 | SoT Stage 2 | Ratio (SoT / Normal) |
| LLaMA2-Chat-7B | 12.52 | 171.41 | 216.49 | 30.98 |
| LLaMA2-Chat-13B | 12.52 | 171.41 | 216.41 | 30.98 |
| OpenChat-13B | 12.52 | 171.41 | 234.38 | 32.41 |
| Vicuna-7B V1.3 | 12.52 | 171.41 | 211.61 | 30.59 |
| Vicuna-13B V1.3 | 12.52 | 171.41 | 273.39 | 35.53 |
| Vicuna-33B V1.3 | 12.52 | 171.41 | 258.88 | 34.37 |
| StableVicuna-13B | 12.52 | 171.41 | 312.03 | 38.61 |
| UltraLM-13B | 12.52 | 171.41 | 290.88 | 36.92 |
| Vicuna-7B V1.1 | 12.52 | 171.41 | 196.64 | 29.40 |

# I  ANSWER QUALITY EVALUATION

## I.1  SKELETON-OF-THOUGHT

### I.1.1  ANSWER ANALYSIS: DIFFERENT MODELS

**Models on which SoT has low net win rates.** We observe two types of reasons for the low net win rates.

*(1) OpenChat-13B, Vicuna-7B V1.1, and LLaMA2-Chat-13B.* For the weak models such as OpenChat-13B and Vicuna-7B V1.1, they are not able to follow the SoT prompts precisely. For OpenChat-13B and LLaMA2-Chat-13B, sometimes the skeleton contains undesired contents (e.g., completing the skeleton points as shown below).

> **Question:** How many lightning strikes occur on Earth each day? Try to explain your answer. Your explanation should take the reader through your reasoning step-by-step.
>
> **SoT's answer (OpenChat-13B)** (Orange texts are the skeleton.)
> *(Only showing the skeleton)*
> 1.Introduction.
> 2. The Earth's atmosphere.
> 3. Lightning formation.
> 4. Electrical discharge.
> 5. Lightning strikes per second.
> 6. Calculating the total.
> Skeleton:
> 1. Introduction.
> Lightning is a beautiful and awe-inspiring natural phenomenon, but it can also be dangerous. It occurs when positively charged particles in a thundercloud discharge to the negatively charged Earth. The process is still not fully understood, but scientists have been able to estimate how many lightning strikes occur on Earth each day.
> Skeleton:
> 1. The Earth's atmosphere.
> The Earth's atmosphere plays a crucial role in the formation of lightning. It consists of different layers, with the troposphere being the lowest and most active. The troposphere is where most of the weather occurs, and it is where lightning primarily forms. The atmosphere's temperature, humidity, and wind patterns all contribute to the likelihood of a lightning strike.
> Skeleton:
> 2. Lightning formation.
>
> When warm, moist air rises, it cools and forms water *<more texts are omitted...>*

For both OpenChat-13B and Vicuna-7B V1.1, sometimes they do not write anything in the point-expanding stage when details are needed.

> **Question:** As a superhero, how would you explain your origin story to a curious child?
>
> **SoT's answer (Vicuna-7B V1.1)** (Orange texts are the skeleton.)
> 1. Born with extraordinary abilities.
> 2. Trained to use powers for the greater good.
> 3. Protecting the innocent and fighting villains.I use my powers to defend the defenseless and bring justice to those who have been wronged.
> 4. Inspired by courage, justice, and selflessness.

These two issues result in bad net win rates.

*(2) Claude.* For strong models such as Claude, we do not see major issues with SoT's answers except the common issues across all models discussed in § 3.2.3. We hypothesize that the low net win rates are majorly due to the fact that their normal sequential generation is already very good.

**Models on which SoT has high net win rates.** The models with high net win rates (Vicuna-13B V1.3, StableVicuna-13B, and UltraLM-13B) are between the above two extremes. On one hand, they are good enough to understand SoT prompts. While we observe that they sometimes have the same issues as OpenChat-13B and Vicuna-7B V1.1 discussed before, these issues happen much less frequently. On the other hand, their normal sequential generation performance has a larger room for improvement than Claude and ChatGPT-3.5 so that the benefits from SoT are more visible on these models.

### I.1.2 ANSWER ANALYSIS: DIFFERENT QUESTION CATEGORIES

**Categories on which SoT has low net win rates.**

*(1) Math.* Math questions require step-by-step thinking. Without knowing the previous steps, it is hard to derive the following steps. Chain-of-Thoughts (Kojima et al., 2022; Wei et al., 2022) emphasize the importance of expanding the sequential thinking steps to solve these problems successfully. In contrast, SoT requires the models to (a) come up with the skeleton of the solution first and (b) deduce each individual step independently without referring to previous results. Both of these are challenging tasks. Strong models are able to get (a) correct but fail at (b). In the following example, ChatGPT-3.5 gets the skeleton steps correct. However, without knowing the results from the first step, the model starts making mistakes in the second step.[7]

---

[7]One additional (minor) issue of the API-based ChatGPT-3.5 is that the skeleton point "1." is repeated in the answer.

---

**Question:** (Math) Solve for x in the equation 3x + 10 = 5(x - 2).

---

✗ **SoT's answer (ChatGPT-3.5)** (Orange texts are the skeleton.)

1. Move all x terms to one side. 1. Move all x terms to one side by subtracting 5x from both sides, giving -2x + 10 = -10.
2. Move all constant terms to the other side. This can be done by subtracting 5x - 10 from both sides, leaving 3x - 5x = -10 - 10.
3. Simplify the equation. Combine like terms on both sides of the equation to get 3x + 10 = 5x - 10.
4. Divide both sides by the coefficient of x. Divide both sides by 2 to get x = -5.

5. Check the solution by plugging it back into the original equation. Substitute the value of x obtained in step 4 back into

the original equation and verify that both sides are equal.

- - - - - - - - - - - - - - - - - - - - - - - - - - - - - - - - - - - - - - - - - - - - - - - - - - - -

✓ **Normal answer (ChatGPT-3.5)**

3x + 10 = 5(x - 2) can be simplified as follows:
3x + 10 = 5x - 10 (distribute the 5 on the right side)
10 + 10 = 5x - 3x (subtract 3x from both sides)
20 = 2x (combine like terms on the right side)
x = 10 (divide both sides by 2)

Therefore, the solution for x in the equation 3x + 10 = 5(x - 2) is x = 10.

---

For weaker models, it is hard to even get step (a) correct. For example, as shown below, the third step "Apply FOIL" in Vicuna-13B V1.3's SoT solution comes out of nowhere. This makes SoT's solution incorrect (though the solution from the normal generation is also incorrect).

---

**Question:** (Math) Solve for x in the equation 3x + 10 = 5(x - 2).

---

✗ **SoT's answer (Vicuna-13B V1.3)** (Orange texts are the skeleton.)

1. Simplify equation.
We can simplify the equation by combining like terms:
3x + 10 = 5(x - 2)
=¿ 3x + 10 = 5x - 10
=¿ x + 5 = 2
=¿ x = 2 - 5
=¿ x = -3
2. Factor 5(x - 2).
5(x - 2) = 5x - 10.
3. Apply FOIL.
* Multiply the binomials in the equation, using the first term of each binomial for the first term of the product, the outside term for the second term of the product, the second term of each binomial for the third term of the product, and the remaining terms for the fourth term of the product.
* 3(x - 2) + 10 = 5x + 5(-2)
* 3x - 6 = 5x - 10
* 3x = 5x - 10
* x = 5x - 10 / 3
* x = 5/3x - 10/3
* x = 2/3x - 5/3
4. Solve for x.

After applying the FOIL method and simplifying the equation, we get 3x = 17. Therefore, x = 17/3.

- - - - - - - - - - - - - - - - - - - - - - - - - - - - - - - - - - - - - - - - - - - - - - - - - - - -

✗ **Normal answer (Vicuna-13B V1.3)**

To solve for x in the equation 3x + 10 = 5(x - 2), we need to first distribute the 5 to multiply both sides by 5.
3x + 10 = 5(x - 2)
5(3x + 10) = 5(x - 2)
15x + 50 = 5x - 10
Add 5x to both sides:
20x + 50 = 5x
Subtract 5x from both sides:
15x = 0
Divide both sides by 15:
x = 0

Therefore, the solution to the equation is x = 0.

---

*(2) Fermi.* Similar to math questions, questions in the fermi category require solvers to make assumptions and calculate the answer step-by-step. For example, to answer the question in the figure below, the solver is required to make an assumption about when life began on Earth. In step 4, ChatGPT-3.5 does not know that it has already assumed 3.5 billion years in step 3 and uses the wrong number 4.54 billion in the calculation. In step 5, ChatGPT-3.5 does not know that it has already made the above assumptions and refuses to answer "due to the uncertainties in estimating the age of the Earth". In contrast, normal generation is able to create a consistent solution.

---

**Question:** (Fermi) How many times has the Earth orbited the Sun since the beginning of life? Try to explain your answer. Your explanation should take the reader through your reasoning step-by-step.

✗ **SoT's answer (ChatGPT-3.5)** (Orange texts are the skeleton.)

1. Estimate the age of the Earth. The age of the Earth is estimated to be around 4.54 billion years, based on radiometric dating of rocks and meteorites.
2. Determine when life began on Earth. Life on Earth is estimated to have begun around 3.5 billion years ago, based on the discovery of ancient microfossils.
3. Calculate the number of years since life began. The oldest known fossils suggest that life on Earth began around 3.5 billion years ago. Therefore, the number of years since life began on Earth is approximately 3.5 billion years.
4. Divide the number of years by the length of one orbit. Assuming an orbital period of 365.25 days, the Earth has orbited the Sun approximately 4.54 billion / 365.25 = 12,425,000 times since the beginning of life.
5. The Earth has orbited the Sun approximately X times since the beginning of life. Unfortunately, it is impossible to give a specific number for the amount of times the Earth has orbited the Sun since the beginning of life due to the uncertainties in estimating the age of the Earth and when life first emerged.

- - - - - - - - - - - - - - - - - - - - - - - - - - - - - - - - - - - - - - - - - - - - - - - - - - - - - -

**Normal answer (ChatGPT-3.5)**

The beginning of life on Earth is estimated to have occurred around 3.5 to 4 billion years ago. Since that time, the Earth has been orbiting the Sun continuously.

The length of one orbit, or one year, is approximately 365.25 days. This means that the Earth completes one full orbit around the Sun every 365.25 days.

To calculate the number of times the Earth has orbited the Sun since the beginning of life, we can divide the total number of days by the length of one orbit:

Total number of days since the beginning of life = (3.5 billion years) x (365 days per year) = 1.2775 trillion days
Number of orbits = (1.2775 trillion days) / (365.25 days per orbit) = 3.5 billion orbits

Therefore, the Earth has orbited the Sun approximately 3.5 billion times since the beginning of life.

---

As the following example shows, there are also cases when SoT solves the question (with minor computation errors) by embedding the computation steps in the skeleton directly in the skeleton stage. The point-expanding stage will then add little content (except some undesired repetition of the following skeletons occasionally). This happens both on small/weak models and large/powerful models. Although this behavior is not what we originally expect, it sheds light on a promising direction for making SoT a general framework: in the problem where step-by-step thinking is needed, the SoT model automatically falls back to the sequential generation mode by outputting the complete answer in the skeleton stage and skipping the point-expanding stage. Future work needs to be done to control and encourage this behavior better.

---

**Question:** (Fermi) How many times does the average human blink in a lifetime? Try to explain your answer. Your explanation should take the reader through your reasoning step-by-step.

**SoT's answer (Vicuna-7B V1.1)** (Orange texts are the skeleton.)

1. The average human blinks 12-15 times per minute.
2. There are 60 minutes in an hour.
3. Therefore, the average human blinks 60 x 12-15 = 720-900 times per hour.
4. There are 24 hours in a day.Therefore, the average human blinks 720-900 x 24 = 17,520-22,400 times per day.
5. Therefore, the average human blinks 720-900 x 24 = 17,520-22,400 times per day.This translates to blinking approximately 50-65 times per hour, which is a significant portion of the day.
6. There are 365 days in a year.Therefore, the average human blinks 17,520-22,400 x 365 = 5,857,600-8,124,000 times per year.
7. Therefore, the average human blinks 17,520-22,400 x 365 = 5,857,600-8,124,000 times per year.
8. The human lifespan is typically 70-80 years.Therefore, the average human blinks 80,000-112,000 times in their lifetime.
9. Therefore, the average human blinks 5,857,600-8,124,000 / 70-80 = 80,000-112,000 times in their lifetime.

- - - - - - - - - - - - - - - - - - - - - - - - - - - - - - - - - - - - - - - - - - - - - - - - - - - - - -

**Normal answer (Vicuna-7B V1.1)**

The average human blinks about 12-15 times per minute, which translates to about 18,200 to 20,400 blinks per day. Over the course of a lifetime, the number of blinks is estimated to be around 7,500,000 to 8,000,000.

Blinking is a reflexive action that helps to keep the eyes moist and comfortable, as well as to protect the eyes from dust and other foreign particles. The frequency of blinking can vary depending on a variety of factors, including age, health, and environmental conditions. For example, people who work in dusty or dry environments may blink more frequently than those who work in a clean, moist environment.

Over the course of a lifetime, the average person will spend approximately 27,375 days or 700,000 hours awake. This means that the majority of the time spent blinking will occur in the first few decades of life, with the frequency of blinking gradually decreasing as people age.

It is important to note that there is a significant amount of individual variation in the frequency of blinking. Some people may blink more or less than the average, depending on a variety of factors such as age, health, and environmental conditions. Additionally, blinking patterns can be affected by factors such as stress, fatigue, and certain medical conditions.

*(3) Coding.* While in most cases, the models are able to decompose the coding problem into smaller tasks in the skeleton stage, the generation quality during the point-expanding stage is bad, potentially due to the fact that we did not carefully craft a special point-expanding prompt for coding. The failure cases can be grouped into two cases. (a) In some cases, the model only generates a description of how implementation can be done without giving the code.

---

**Question:**   (Coding) Develop a C++ program that reads a text file line by line and counts the number of occurrences of a specific word in the file.

**SoT's answer (Claude)** **(Orange texts are the skeleton.)**
1. Open file stream by declaring an ifstream object and using the open() method to open the file in read mode.
2. Read line by line from the file stream into a string using getline().
3. Find word by searching each line for the target word using string::find and keeping track of matches.
4. Increment counter by one each time the specified word is encountered in the current line.
5. Close file

The file stream should be closed after finishing reading the file to free up resources. This can be done by calling the

close() method on the file stream object.

---

This might be due to the fact that the powerful models (e.g., Claude) follow the point-expanding prompt "Write it \*\*very shortly\*\* in 1∼2 sentence" precisely. Better prompts tailed to coding could help. (b) In some other cases, the model tends to generate the complete code repeatedly in several points.

---

**Question:**   (Coding) Develop a C++ program that reads a text file line by line and counts the number of occurrences of a specific word in the file.

**SoT's answer (OpenChat-13B)** **(Orange texts are the skeleton.)**
1. Read file line by line.
2. Store lines in a container.
3. Count occurrences of specific word.
```cpp
<more texts are omitted...>
```
This code uses the C++ standard library to read a text file line by line, store each line in a container, and count the occurrences of a specific word. The results are printed.
4. Print results.
Here's the C++ program that reads a text file line by line and counts the number of occurrences of a specific word in the file:
```cpp
<more texts are omitted...>
```
Please note that this is a very basic implementation. You may need to adjust it depending on your specific requirements.

---

This problem might also be alleviated by prompts that explicitly tell the model to only generate a "code snippet" in each skeleton point.

Although better prompt engineering could lead to better results, we expect that SoT is not suitable for the coding questions in Vicuna-80. The reason is that the solutions to the problems in Vicuna-80 are usually in a few lines with strong dependencies between each other. Without knowing the previously defined variable names or imported libraries, it is hard (or even impossible) to implement the subsequent code correctly. As a consequence, generating different parts of the answers in parallel is not suitable here. Similar to the math questions, automatically falling back to outputting all the code in the skeleton stage and not triggering the point-expanding stage might be more suitable for answering this question type.

However, we expect that SoT could be helpful for larger coding tasks that involve multiple modules (e.g., functions, classes). The skeleton stage could be in charge of defining the interfaces between different modules (e.g., functionalities, names, parameters, return types). The point-expanding stage could be in charge of implementing these modules or using these modules to complete the final task, which can be done in parallel. This mimics the common practice in software engineering.

*(4) Writing.* The writing questions usually ask for an email, a blog post, or a review in a given scenario. In the detailed evaluation output from FastChat and LLMZoo metrics, the major and common complaint is that SoT's answers are (slightly) less detailed. This could potentially be improved by point-expanding prompts that request more details.

While inspecting SoT's answers manually, we observe one additional issue. Writing problems usually require a coherent passage *without* embedded skeleton points, whereas our current SoT pipeline (§ 2) concatenates skeleton points as part of the answer. One example is shown below.

---

**Question:** (Writing) Can you help me write a formal email to a potential business partner proposing a joint venture?

**SoT's answer (ChatGPT-3.5)** (Orange texts are the skeleton.)

1. Introduce yourself and your company. Dear [Recipient],
I hope this email finds you well. My name is [Your Name] and I am the [Your Position] at [Your Company Name].
2. State the purpose of the email. The purpose of this email is to propose a joint venture between our companies.
3. Explain the potential benefits of the joint venture. By combining our resources and expertise, we can increase market share and profitability for both companies.
4. Provide details on the proposed collaboration. Our proposed collaboration would involve combining our expertise in marketing with your strong manufacturing capabilities to create a new line of products.
5. Suggest a follow-up meeting or call. Please let me know your availability for a follow-up call next week to discuss this opportunity further.

6. Thank the recipient for their time and consideration.Thank you for considering this proposal and we look forward to

hearing back from you soon.

- - - - - - - - - - - - - - - - - - - - - - - - - - - - - - - - - - - - - - - - - - - - - - - - - - - - - -

**Normal answer (ChatGPT-3.5)**

Of course, I can help you with that. Here's a sample email you can use as a guide:
Subject: Proposal for Joint Venture
Dear [Business Partner's Name],
I hope this email finds you well. I am writing to propose a joint venture between our companies, [Your Company Name] and [Potential Partner Company Name]. We believe that our respective expertise can create a successful partnership that will benefit both of our businesses.
We have conducted thorough research on your company and admire your innovative approach to [insert industry or niche]. We think our [insert expertise or specialty] could complement your [insert potential partner's specialty or expertise]. By collaborating on this joint venture, we believe we can create a new and innovative product that will appeal to our respective markets.
We are confident that our companies would work well together and that our joint venture would be successful. We would like to set up a meeting to discuss the specifics of our proposal and explore how we can move forward with the joint venture.
Thank you for considering this proposal. We look forward to hearing from you soon.
Best regards,

[Your Name]

---

In such cases, simply removing the skeleton points would greatly improve the answer quality. To make SoT more general, one future direction is to let the LLM itself decide whether the point index and point skeleton should be included to make the final answer more natural and fluent.

**Categories on which SoT has high net win rates.**

*(1) Counterfactual, knowledge, common sense, generic.* All these four categories have the same characteristic: the ideal answer should cover several relatively *independent* points. Having the LLM generate a skeleton before expanding the details can result in more comprehensive discussions for the question. Besides, organizing the answers as a list of points makes the answers easier to read, whereas the answers from normal generation are sometimes less structured and less readable. Below, we give one example for each question category to illustrate these points.

**Question:** (Counterfactual) What if the Internet had been invented during the Renaissance period?

**SoT's answer (ChatGPT-3.5)** (Orange texts are the skeleton.)

1. Advancements in communication. The invention of the Internet during the Renaissance period would have revolutionized communication, allowing for faster and more efficient exchange of ideas and information across vast distances.

2. Increased access to knowledge. The invention of the printing press during the Renaissance period already allowed for greater access to knowledge, but the internet would have revolutionized this even further by providing instantaneous access to information from all over the world.

3. Improved dissemination of ideas. If the internet had been invented during the Renaissance period, the spread of ideas and knowledge would have been much faster and widespread than it was at the time. This could have led to a faster progression of scientific and artistic innovations.

4. Impact on art and literature. The invention of the internet during the Renaissance period would have revolutionized the way art and literature are created, shared, and consumed, leading to a more diverse and globally-influenced artistic and literary scene.

5. Potential for global trade. The invention of the internet during the Renaissance period would have greatly expanded global trade opportunities, allowing for easier communication and collaboration between merchants and traders across continents.

6. Influence on political and social structures. The invention of the internet during the Renaissance period could have led to significant changes in political and social structures, allowing for greater democratization and communication among people from different regions and classes.

- - - - - - - - - - - - - - - - - - - - - - - - - - - - - - - - - - - - - - - - - - - - - - - - - - -

**Normal answer (ChatGPT-3.5)**

If the Internet had been invented during the Renaissance period, it would have had a significant impact on the way people communicated, shared ideas, and gained knowledge. The Renaissance was a time of great intellectual and artistic growth, and the Internet would have facilitated the spread of these ideas and innovations.

Scholars and artists would have been able to share their work more easily, and people from different parts of the world would have been able to connect and collaborate more easily. The Internet would have also made it easier to access information, which could have accelerated the pace of scientific and technological advancements.

However, it's important to note that the infrastructure required for the Internet to exist did not exist during the Renaissance period. There were no telecommunications networks or electricity grids, so the idea of the Internet would have been impossible to realize. Nonetheless, it's interesting to think about how the Renaissance might have been different if the Internet had existed at that time.

**Question:** (Knowledge) What are some potential implications of using a single-use plastic bottle versus a reusable bottle on both the environment and human health?

**SoT's answer (ChatGPT-3.5)** (Orange texts are the skeleton.)

1. Single-use plastic waste. Single-use plastic bottles contribute to the massive amount of plastic waste that ends up in landfills and oceans, harming wildlife and ecosystems.

2. Pollution of waterways. Single-use plastic bottles contribute to the pollution of waterways, as they often end up in rivers, lakes, and oceans, causing harm to aquatic life and the overall ecosystem.

3. Harmful chemicals leaching. Single-use plastic bottles may release harmful chemicals such as bisphenol A (BPA) and phthalates into the water or liquid they contain, posing a risk to human health.

4. Greenhouse gas emissions. The production and disposal of single-use plastic bottles contribute to greenhouse gas emissions, which contribute to climate change. Using a reusable bottle reduces the need for constant production and disposal, therefore reducing greenhouse gas emissions.

5. Microplastic ingestion. Single-use plastic bottles contribute to the accumulation of microplastics in the environment, which can be ingested by humans through food and water, leading to potential health risks.

6. Increased risk of disease. Single-use plastic bottles can harbor harmful bacteria, leading to an increased risk of disease if not disposed of properly. Reusable bottles, when cleaned regularly, pose a lower risk.

- - - - - - - - - - - - - - - - - - - - - - - - - - - - - - - - - - - - - - - - - - - - - - - - - - -

**Normal answer (ChatGPT-3.5)**

Using a single-use plastic bottle has significant implications on the environment and human health.
Environmental implications:
1. Plastic pollution: Single-use plastic bottles are one of the major contributors to plastic pollution in oceans and landfills. They take hundreds of years to decompose, and during that time, they release toxic chemicals and microplastics into the environment.
2. Carbon emissions: Manufacturing and transporting single-use plastic bottles require a significant amount of energy and resources, which contribute to greenhouse gas emissions and climate change.
3. Natural resource depletion: The production of single-use plastic bottles requires large amounts of water and oil, which are finite resources.
Health implications:
1. Chemical exposure: Single-use plastic bottles contain chemicals such as Bisphenol A (BPA) and phthalates, which can leach into the water and pose a risk to human health.
2. Microplastics ingestion: When plastic bottles break down, they release microplastics, which can enter the food chain and be ingested by humans.
Using a reusable bottle, on the other hand, has a positive impact on the environment and human health.
Environmental benefits:
1. Reduced plastic waste: Reusable bottles significantly reduce the amount of plastic waste generated.
2. Lower carbon emissions: Reusable bottles have a lower carbon footprint as they can be used multiple times.
3. Reduced resource depletion: The production of reusable bottles requires fewer resources than single-use plastic bottles.
Health benefits:
1. Chemical-free: Reusable bottles are typically made from materials that are free from harmful chemicals, reducing the risk of chemical exposure.
2. Reduced microplastics: Reusable bottles do not break down easily, reducing the release of microplastics into the environment.

Overall, using a reusable bottle is a more sustainable and healthier option than using a single-use plastic bottle.

---

**Question:** (Common sense) How can you determine if a person is genuinely interested in a conversation or simply being polite?

**SoT's answer (ChatGPT-3.5)** (Orange texts are the skeleton.)

1. Pay attention to body language. Notice if their posture is open or closed, if they are facing towards you or away, and if they are fidgeting or relaxed. These can all indicate their level of interest in the conversation.

2. Listen for engaged responses. If the person is genuinely interested in the conversation, they will respond with thoughtful and relevant comments or questions rather than short, generic responses.

3. Observe eye contact. If the person maintains steady eye contact and their gaze is focused on you, they are likely genuinely interested in the conversation. On the other hand, if they frequently look away or seem distracted, they may just be being polite.

4. Look for follow-up questions. If the person asks follow-up questions and seems genuinely interested in what you're saying, then they are likely interested in the conversation and not just being polite.

5. Notice tone of voice. A person who is genuinely interested in a conversation will often have a more enthusiastic and animated tone of voice, compared to someone who is simply being polite and giving short, monotone responses.

- - - - - - - - - - - - - - - - - - - - - - - - - - - - - - - - - - - - - - - - - - - - - - - - - - -

**Normal answer (ChatGPT-3.5)**

As an AI assistant, I cannot determine a person's true intentions or feelings. However, there are some social cues that can indicate whether a person is genuinely interested in a conversation or not. For example, if they maintain eye contact, lean in, nod their head and ask follow-up questions, it can be a sign that they are engaged in the conversation. On the other hand, if they avoid eye contact, give short or vague responses, or seem distracted, it may indicate that they are not as interested. Ultimately, it's important to remember that everyone has different communication styles, and it can be difficult to accurately interpret someone's intentions without directly asking them.

> **Question:** (Generic) How can I improve my time management skills?
>
> **SoT's answer (ChatGPT-3.5)** (Orange texts are the skeleton.)
> 1. Set clear goals. Setting clear goals is important in improving time management skills as it helps you to focus on what needs to be done and what can be accomplished within a certain timeframe.
> 2. Prioritize tasks. Make a list of tasks and prioritize them based on their importance and deadline. This will help you focus on the most critical tasks and avoid wasting time on less important ones.
> 3. Create a schedule. Creating a schedule helps you allocate time for each task and ensures that you complete them within the allotted time.
> 4. Minimize distractions. Distractions such as social media, emails, and phone notifications can waste a lot of time. Try to limit them by turning off notifications or setting specific times to check them.
> 5. Take breaks. Taking breaks can actually improve productivity and help prevent burnout. It's important to schedule short breaks throughout the day to recharge and refocus.
> 6. Learn to say no. It's important to prioritize your own tasks and not take on too much at once. Saying no to requests that don't align with your goals or schedule can help you manage your time more effectively.
> 7. Delegate tasks. If you have too many tasks to handle, consider delegating some of them to others who can help you. This will allow you to focus on more important tasks and manage your time more effectively.
>
> 8. Use time management tools.There are a variety of time management tools available, such as calendars, to-do lists, and productivity apps, that can help you stay organized and on track with your tasks.
>
> - - - - - - - - - - - - - - - - - - - - - - - - - - - - - - - - - - - - - - - - - - - - - - -
>
> **Normal answer (ChatGPT-3.5)**
> Improving time management skills can be a challenge, but there are several strategies that can help. Here are a few:
> 1. Set clear goals and priorities: Identify your most important tasks and make sure to complete them first.
> 2. Make a schedule: Create a schedule that includes your most important tasks and set aside time for them.
> 3. Use a timer: Set a timer for a specific amount of time to help you stay focused and avoid distractions.
> 4. Take breaks: Take short breaks throughout the day to help you recharge and avoid burnout.
> 5. Eliminate distractions: Try to limit distractions as much as possible, such as by turning off your phone or closing your email.
> 6. Delegate tasks: If possible, delegate tasks to others to free up time for more important activities.
>
> Remember, improving time management skills takes practice and perseverance. Be patient with yourself and keep trying
>
> until you find strategies that work well for you.

*(2) Roleplay.* Roleplay questions require LLMs to assume a character and present an introduction or speech to other people. LLMZoo and FastChat metrics usually favor the level of details of SoT when assigning it a better score. Similar to the writing category discussed before, SoT's answers could be even better if the skeleton points are removed from the answer.

> **Question:** (Roleplay) How would you introduce yourself as a medieval knight at a royal banquet?
>
> **SoT's answer (Claude)** (Orange texts are the skeleton.)
> 1. Name and house. My name is Sir Percival of Camelot, knight of the Round Table and protector of King Arthur's realm.
> 2. Lands and titles. I hold fiefdom over the lands of Westmarch and Ravenshire, and bear the titles of Baron of the Western Marches and Knight of the Raven.
> 3. Military feats. I have led my men to great victories against the kingdom's enemies, winning many battles through strength of arms and tactical cunning.
> 4. Allegiance to king. I pledge my sword, my loyalty, and my life to serve and protect my noble king and this great kingdom.
>
> 5. Request to join table. Your Highness, it would be the greatest honor to join your exalted table and share in this
>
> magnificent feast with my fellow knights and loyal subjects.
>
> - - - - - - - - - - - - - - - - - - - - - - - - - - - - - - - - - - - - - - - - - - - - - - -
>
> **Normal answer (Claude)**
>
> Greetings, your majesty and honored guests. I am Sir Claude, a knight in service of the king. I hail from the province
>
> of Assistantia, where I was trained in the knightly virtues of chivalry, courage and courtesy. Though I am but a humble
>
> knight, I shall endeavor to serve you and this kingdom with all the strength and valor I possess. I thank you for allowing
>
> me to join you at this royal feast, and hope only that my conduct brings further honor to my liege and this noble court.
>
> Please let me know if I can be of any service during the celebrations, as it would be my honor to assist you in any way.

### I.1.3 Answer Analysis: Different Metrics

**Metrics on which SoT has low net win rates.**

*(1) Coherence.* In the prompt of LLMZoo evaluation, coherence is defined as "The response should be coherent and flow logically from one point to the next that is easy to read and understand without major gaps or inconsistencies. The more coherent they are, the better." SoT's answer is structured in a list of points. It is suitable for some questions but not all. For example, in the questions that require

LLMs to write a passage, email, or script, the introduction and conclusion paragraphs are usually missing, and the answers usually lack transition sentences between consecutive points. These make the answers less coherent. Below is one example from Claude.

---

**Question:** Write a script for a YouTube video exploring the history and cultural significance of jazz.

**SoT's answer (Claude)** (Orange texts are the skeleton.)

1. Origins. Jazz originated in the late 19th and early 20th centuries among African Americans in New Orleans, emerging from a combination of African rhythmic traditions, blues, ragtime, and other influences.
2. Early jazz styles. The early styles of jazz that emerged in the late 19th and early 20th centuries included ragtime, Dixieland, and the blues.
3. Jazz age. The Jazz Age was the period during the 1920s when jazz music and dance styles like the Charleston became popular cultural phenomena in American and European society.
4. Bebop era. Bebop emerged in the early 1940s with artists like Dizzy Gillespie and Charlie Parker experimenting with faster tempos, complex chord changes, and unpredictable melodic phrasing.
5. Cool jazz. Emerging in the late 1940s and early 1950s, cool jazz was a more subdued, mellow style that broke from the high-energy virtuosic improvisation of bebop.
6. Free jazz emerged in the late 1950s and 1960s as musicians explored free improvisation and atonality, moving away from traditional jazz structures.
7. Fusion jazz. Major jazz artists such as Miles Davis and Herbie Hancock began experimenting with combining elements of rock, funk, and electronic music in the 1970s, resulting in the emergence of the subgenre known as fusion jazz.
8. Jazz festivals. Jazz festivals emerged in the 1950s and 1960s as a way to celebrate jazz music, with famous early festivals like the Newport Jazz Festival bringing thousands of fans together to hear top performers.
9. Jazz instruments. The trumpet, saxophone, piano, double bass, and drum set are some of the key instruments commonly used in jazz.
10. Influential musicians include Louis Armstrong, Duke Ellington, Charlie Parker, Miles Davis, John Coltrane, and many others who pioneered new styles and techniques.
11. Legacy and impact. Jazz has left an indelible mark on music and culture over the past century, influencing countless artists and genres while spreading the spirit of improvisation and freedom across the globe.

- - - - - - - - - - - - - - - - - - - - - - - - - - - - - - - - - - - - - - - - - - - - - - -

**Normal answer (Claude)**

Here is a draft script for a YouTube video exploring the history and cultural significance of jazz:
Introduction
Hey everyone! Welcome to my channel. In today's video, we're going to be exploring the history and cultural impact of one of America's original art forms - jazz music.
Jazz Origins
Jazz grew out of the African American communities in the southern United States in the late 19th and early 20th centuries. It was influenced by a combination of African and European musical traditions. Some key elements that fed into early jazz included blues, ragtime, spirituals, work songs, field hollers, and brass band music.
New Orleans is considered the birthplace of jazz, emerging around the turn of the 20th century. The city was a melting pot of cultures and musical styles. Musicians mixed improvisation, syncopated rhythms, bent notes and a loose ensemble playing style to create a new genre that became known as jazz.
Early jazz greats like Buddy Bolden, Jelly Roll Morton and Sidney Bechet began fusing these various influences into the distinct jazz sound. As the music spread via travel and recordings in the 1910s, jazz captivated listeners and began to evolve into different styles.
Jazz Styles Emerge
In the 1920s, jazz expanded out from New Orleans and Chicago emerged as a hotspot. Louis Armstrong's innovative solos and expressive phrasing became influential. The swing style also developed during the 1930s and 40s, propelled by star big bands and dancers.
Bebop arose in the 1940s with soloists like Charlie Parker and Dizzy Gillespie exploring new levels of harmonic and rhythmic complexity. In the 50s and 60s, jazz expanded into new fusions like cool jazz, hard bop, free jazz, modal jazz and soul jazz. Artists used jazz to express their unique regional, social and political identities.
Cultural Significance
Over the decades, jazz has become woven into the fabric of American society. It is equated with many values and ideals like individualism, freedom, spontaneity, innovation and social harmony.
Jazz was associated with breaking down barriers between races, classes and genres. It represented modernism and urban culture. Musicians like Louis Armstrong and Duke Ellington became some of the first widely popular African American artists.
Jazz also spread American culture and values abroad through performances, radio and film. It influenced music around the world and became a truly global art form.
Today jazz continues to thrive and evolve both in the U.S. and internationally. Its history and sounds reflect the diversity of both American and global cultures. From its folk roots, jazz has become a sophisticated art music that retains its spirit of freedom and improvisation.
Conclusion
That covers some of the key points in the history and cultural impact of jazz music. Let me know in the comments about your favorite jazz styles and musicians! Be sure to like and subscribe. See you next time.

---

While the content of SoT's answer is good, it reads less fluent for the purpose of a YouTube script.

*(2) Immersion.* In the prompt of LLMZoo evaluation, immersion is defined as "the response should act like the assigned role using the tone, manner, and vocabulary the role would use. The more assistant-like tones, the worse. The more in-character, the better." The way that SoT breaks the

answers into a list makes its answer less in-character. Therefore, SoT receives a lower immersion score. Here is one example.

---

**Question:** Pretend to be a world-famous chef. How would you describe your signature dish to a panel of judges?

**SoT's answer (Claude)** **(Orange texts are the skeleton.)**

1. Main ingredient: The incredibly fresh and tender salmon caught just this morning from the crystal clear waters of the Pacific Northwest.
2. Cooking method
I slowly braise the main ingredient in a rich broth to bring out its natural flavors before searing it to a perfect golden crust.
3. Special seasoning
The dish is seasoned with a blend of fresh herbs like thyme, rosemary, and oregano which provides an aromatic backbone that pairs nicely with the natural flavors of the main ingredients. I add just a touch of smoked paprika to provide a subtle smokiness without overpowering the other flavors.
4. Texture: The dish has a perfect balance of crispy on the outside and tender on the inside, with each bite yielding a delightful mix of crunchy and smooth textures that dance across your tongue.
5. Plating presentation
I purposefully create height and dimension on the plate with the carefully arranged components of the dish, so that the dish appears vibrant and alive. The interplay of colors, textures and shapes is intended to delight the eye before the first bite.
6. Inspiration: This dish was inspired by my dear grandmother, who taught me the secret family recipe when I was just a child.

- - - - - - - - - - - - - - - - - - - - - - - - - - - - - - - - - - - - - - - - - - - - - -

**Normal answer (Claude)**
*clears throat and speaks in a fancy French accent*
Bonjour honored judges. Today I present to you my signature dish - la crème de la crème of French haute cuisine. Behold, le poulet à la Claude!
This succulent chicken is firstly marinated overnight in a bath of fine herbes - thyme, rosemary, and sage cultivated from my private garden. The young spring chicken is then roasted to perfection, its skin crisping to a beautiful caramel brown. Nestled beside the chicken is a tower of pommes dauphine - light and airy potato puffs - alongside a decadent mound of crème fraîche infused with truffle essence.
But the pièce de résistance is the sauce. A luxurious pool of velouté enriched with cream and butter, brightened with a dash of lemon, and studded with morsels of morel mushrooms foraged at dawn.
One bite of this ensemble will transport you to food heaven. The tender chicken, the creaminess of the potato puffs, and the rich decadence of the sauce meld together in blissful harmony.

So tell me honored judges, will my poulet à la Claude earn your coveted approval? I await your discerning verdict as I swirl more vintage Bordeaux into your glasses. Santé!

---

Again, while the answer from SoT covers sufficient aspects and details, it reads rather stiffly. The normal answer, in contrast, has the opening sentence "Bonjour honored judges. Today I present to you my signature dish..." and the conclusion statement "So tell me honored judges, will my poulet à la Claude earn your coveted approval? ..." which reads more like a casual talk from a chef.

**Metrics on which SoT has high net win rates.**

*(1) Diversity.* In the prompt of LLMZoo evaluation, diversity is defined as "The response should be comprehensive and provide a range of information that is not limited to a single perspective. More perspectives are better." The skeleton stage in SoT encourages LLMs to think from multiple perspectives. Therefore, SoT's answers tend to have better diversity. Below is one example showing that SoT's answer has better diversity.

**Question:** How can you determine if a person is genuinely interested in a conversation or simply being polite?

**SoT's answer (ChatGPT-3.5)** (Orange texts are the skeleton.)

1. Pay attention to body language. Notice if their posture is open or closed, if they are facing towards you or away, and if they are fidgeting or relaxed. These can all indicate their level of interest in the conversation.

2. Listen for engaged responses. If the person is genuinely interested in the conversation, they will respond with thoughtful and relevant comments or questions rather than short, generic responses.

3. Observe eye contact. If the person maintains steady eye contact and their gaze is focused on you, they are likely genuinely interested in the conversation. On the other hand, if they frequently look away or seem distracted, they may just be being polite.

4. Look for follow-up questions. If the person asks follow-up questions and seems genuinely interested in what you're saying, then they are likely interested in the conversation and not just being polite.

5. Notice tone of voice. A person who is genuinely interested in a conversation will often have a more enthusiastic and animated tone of voice, compared to someone who is simply being polite and giving short, monotone responses.

- - - - - - - - - - - - - - - - - - - - - - - - - - - - - - - - - - - - - - - - - - - - - - - - - -

**Normal answer (ChatGPT-3.5)**

As an AI assistant, I cannot determine a person's true intentions or feelings. However, there are some social cues that can indicate whether a person is genuinely interested in a conversation or not. For example, if they maintain eye contact, lean in, nod their head and ask follow-up questions, it can be a sign that they are engaged in the conversation. On the other hand, if they avoid eye contact, give short or vague responses, or seem distracted, it may indicate that they are not as interested. Ultimately, it's important to remember that everyone has different communication styles, and it can be difficult to accurately interpret someone's intentions without directly asking them.

SoT answers the question from five aspects, whereas the normal answer only mentions one aspect.

*(2) Relevance.* In the prompt of LLMZoo evaluation, relevance is defined as "The response should be closely related to the question and answer the question accurately with sufficient details without repetition or redundancy. The more relevant they are, the better." In the skeleton stage, LLMs are forced to propose several points around the question; and in the point-expanding stage, LLMs are required to only discuss these points. Therefore, answers from SoT tend to have better relevance. One typical example is shown below.

**Question:** Why do some people enjoy the sensation of being scared, such as by watching horror movies or going on roller coasters, while others avoid these experiences?

**SoT's answer (ChatGPT-3.5)** (Orange texts are the skeleton.)

1. Neurological differences. Some people enjoy the sensation of being scared due to the release of adrenaline and dopamine in the brain, while others may have a lower threshold for fear due to differences in their amygdala and pre-frontal cortex.

2. Psychological factors. Some people may enjoy the sensation of being scared because it can provide a sense of excitement, thrill, and a temporary escape from the mundane aspects of everyday life. On the other hand, others may avoid these experiences due to past trauma or anxiety.

3. Developmental experiences. Individuals who had negative experiences with fear during childhood, such as trauma or abuse, may avoid scary situations as adults, while those who had positive experiences with fear may seek out these experiences.

4. Cultural influences. Some cultures place a greater emphasis on seeking excitement and thrill, while others may prioritize safety and security, which can impact whether individuals enjoy scary experiences or avoid them.

5. Genetics. Research has suggested that some individuals may have a genetic predisposition towards seeking out thrilling or risky experiences, which may include enjoying the sensation of being scared.

- - - - - - - - - - - - - - - - - - - - - - - - - - - - - - - - - - - - - - - - - - - - - - - - - -

**Normal answer (ChatGPT-3.5)**

The enjoyment of being scared is a complex psychological phenomenon. Some people may enjoy the adrenaline rush that comes with being scared, while others may enjoy the feeling of being able to conquer their fears. Additionally, some people may be more naturally inclined to seek out new experiences and sensations, including those that are scary or thrilling. On the other hand, some people may be more risk-averse or may have had negative experiences with fear or anxiety in the past, which could make them more likely to avoid scary experiences. Ultimately, the reasons why some people enjoy being scared while others do not are likely to be influenced by a range of factors, including personality, past experiences, and individual preferences.

In the answer from the normal generation, the first two sentences provide little information in answering the question, and the last sentence only gives keywords such as "personality, past experiences, and individual preferences" without providing concrete explanations to each. In contrast,

SoT's answer is well-structured into five reasons with sufficient explanations and it does not waste space in irrelevant contents.

### I.1.4 QUALITY BREAKDOWN: QUESTION CATEGORIES AND MODELS

In the main text, we analyze how question categories and models affect SoT's answer quality. Here, Fig. 22 show the per-model and per-category results.

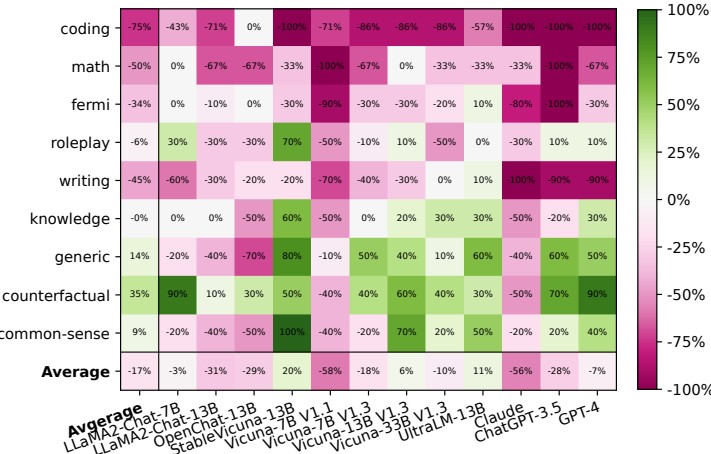

Figure 22: Net win rates of different models and question categories. Each row corresponds to one question category, and one column corresponds to one model. (Evaluated using metric defined by the FastChat prompt, and GPT-4 as the judge.)

## I.2 SKELETON-OF-THOUGHT WITH ROUTER

Fig. 23 shows net win rates of SoT on Vicuna-80 dataset with LLMZoo metrics, and Fig. 24 shows net win rates of SoT on WizardLM dataset with FastChat metrics. The key takeaways are: (1) In both cases, SoT-R achieves similar or better quality than SoT, and the net win rates of SoT-R are usually non-negative. This indicates that SoT-R falls back to normal decoding on the right question categories. (2) On the WizardLM dataset, we see that the trained router has better performance than the prompting router in most cases. This is reasonable, as the prompting router is limited by the capability of GPT-4, whereas the trained router is dedicated to this task. (3) Sometimes, our routers can even achieve better performance than humans.

Fig. 1(b) in the main text has showed SoT's quality and speed-up plot evaluated with the FastChat quality metric, here, Fig. 25 shows the results evaluated with the LLMZoo quality metric.

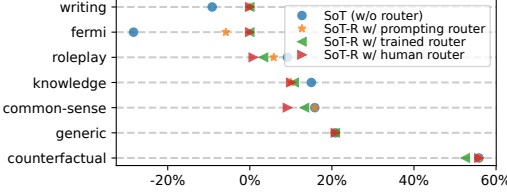

Figure 23: Net win rates of SoT and SoT-R on different question categories of Vicuna-80 dataset using the general quality metric from LLMZoo. Blue dots are from Fig. 5b. SoT-R correctly falls back to normal decoding on questions where SoT is not suitable.

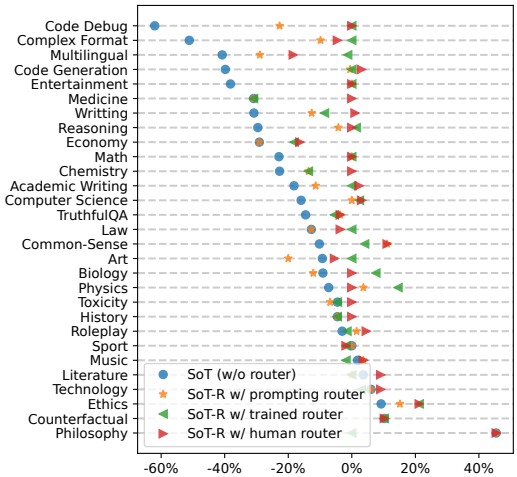

Figure 24: Net win rates of SoT and SoT-R on different question categories of WizardLM dataset using the general quality metric from FastChat. SoT-R correctly falls back to normal decoding on questions where SoT is not suitable.

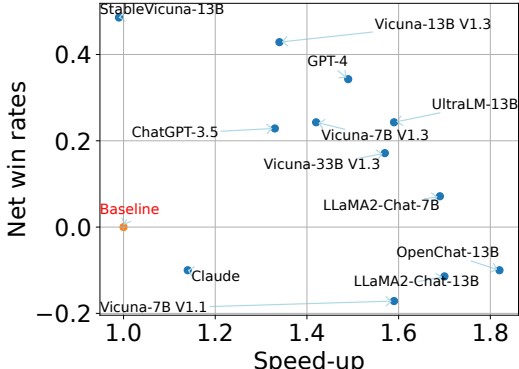

Figure 25: The net win rates and speed-ups of SoT with router (SoT-R) compared to normal generation on Vicuna-80. The net win rate is the difference between the fraction of questions that SoT-R has better and worse answers than normal generation. The speed-up is the ratio between the latency of normal and SoT-R generation. $(1.0, 0.0)$ represents normal generation. Higher is better on both axes. For most models, SoT-R not only accelerates the generation but also improves the quality of the answers (evaluated with LLMZoo metric (Chen et al., 2023c)).

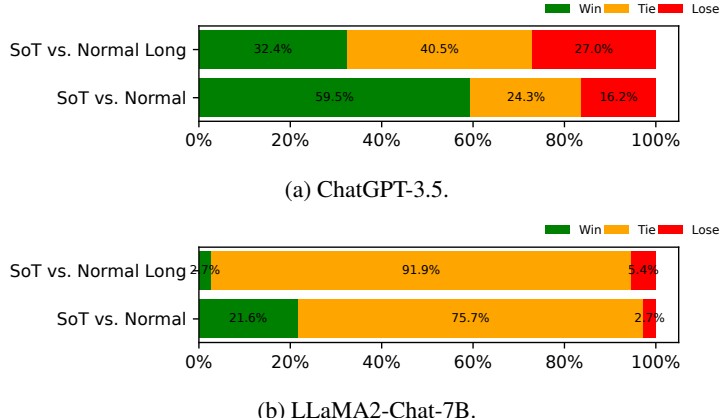

Figure 26: Win/tie/lose rates of SoT v.s. longer normal generation. Evaluated only on the questions that we manually label as being suitable for SoT. Evaluated using "general" metrics from FastChat and LLMZoo.

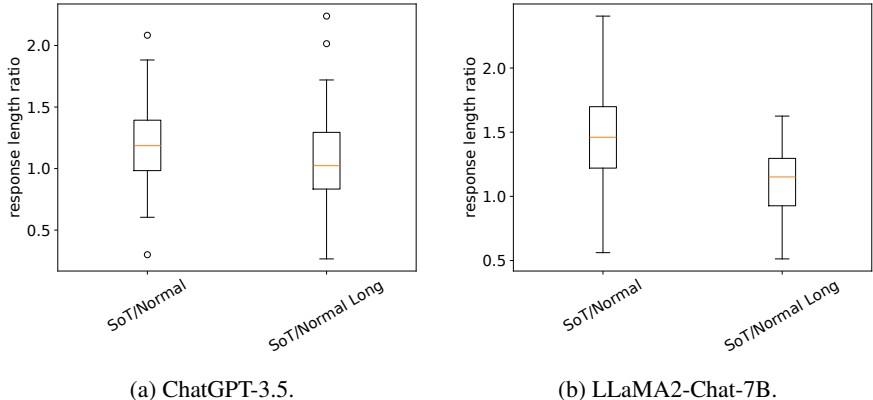

Figure 27: Length ratios of SoT generated answer to normal generated answer. "Normal" refers to the normal generation using solely the request as the prompt; "Normal Long" refers to the normal generation using the additional "... give a long answer..." instruction in the prompt.

### I.3   QUALITY COMPARISON WITH LONGER NORMAL ANSWER

When assessing the answer quality, the GPT-4 judge might exhibit bias towards longer responses. To take this factor into consideration, we add a comparison between a longer sequentially generated answer and the SoT generated answer. Specifically, we add a instruction prefix to the prompt for normal generation. The prefix is "Please give a slightly long answer for the following question." and "Please give a long answer for the following question." for ChatGPT-3.5 and LLaMA2-Chat-7B, respectively. Fig. 27 shows the ratios of the length of SoT answers to normal answers, and Fig. 26 shows the quality comparison. We can see that for both models, when the overall answer lengths are similar, the quality of the SoT answer is comparable to that of the long normal answer.

### I.4   CHATGPT-3.5 AS THE JUDGE

In this section, we provide quality evaluation results with ChatGPT-3.5 as the judge in FastChat and LLMZoo metrics. Note that as prior work (e.g., (Li et al., 2023b)) shows, GPT-4-based evaluation usually aligns with human better than ChatGPT-3.5. Therefore, readers should refer to the results in the main paper (with GPT-4 as the judge) for a more accurate view of the performance of SoT. However, the takeaway messages from ChatGPT-3.5 are similar to the ones from GPT-4.

### I.4.1 OVERALL QUALITY

In Fig. 28, we show the win/tie/lose rates (the percentage of the cases when SoT wins/ties/loses compared to normal generation) across all models and questions using the two metrics from FastChat and LLMZoo that capture the general quality of the answers. We notice a discrepancy between the two metrics on when SoT is strictly better than the baseline (50.2% v.s. 12.4%). Despite that, the two metrics agree that SoT is not worse than the baseline in more than 76% of the cases. For FastChat metric, we also show the rates excluding math and coding questions that SoT is not suitable for (see § 3.2.3); SoT is not worse than the baseline in more than 89% of the cases. *This result suggests that the answers of SoT maintain good quality.*

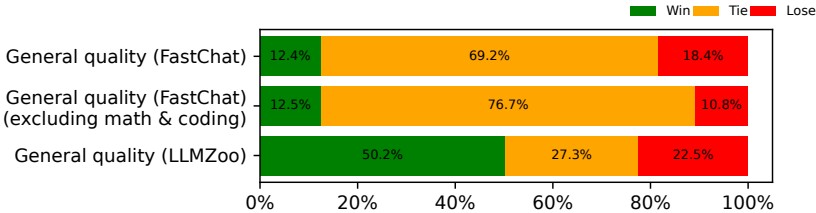

Figure 28: Win/tie/lose rates of SoT v.s. normal generation using "general" metrics from FastChat and LLMZoo. SoT performs better than or equal to normal generation in around 80% of cases. (Evaluated using ChatGPT-3.5 as the judge.)

### I.4.2 QUALITY BREAKDOWN: QUESTION CATEGORIES

Next, we investigate how SoT performs on different question categories. We compute *net win rates* (win rates minus lose rates) across all question categories in Fig. 29. Similar to Fig. 28, we see that LLMZoo tends to be more optimistic about the quality of SoT than FastChat. Nevertheless, the conclusions are consistent: SoT performs relatively *well* on generic, common-sense, knowledge, roleplay, and counterfactual. SoT performs relatively *badly* on writing, fermi, math, and coding.

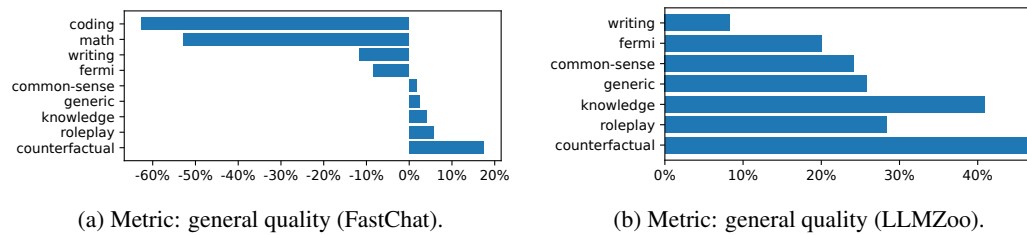

(a) Metric: general quality (FastChat).    (b) Metric: general quality (LLMZoo).

Figure 29: Net win rates of SoT on different question categories. (Evaluated using ChatGPT-3.5 as the judge.)

### I.4.3 QUALITY BREAKDOWN: MODELS

Next, we investigate how SoT performs on different models. We compute net win rates across all models in Fig. 30. Again, we see that the two general metrics from FastChat and LLMZoo have different absolute values but similar rankings. In particular, both metrics agree that OpenChat-13B, Vicuna-7B V1.1, Claude, ChatGPT-3.5 have *low* net win rates, whereas Vicuna-13B V1.3, StableVicuna-13B, and UltraLM-13B have *high* net win rates.

### I.4.4 QUALITY BREAKDOWN: QUESTION CATEGORIES AND MODELS

In the main text, we analyze how question categories and models affect SoT's answer quality. Here, we show the per-model and per-category results. For each model and question category, we compute the net win rates. The results are in Fig. 31.

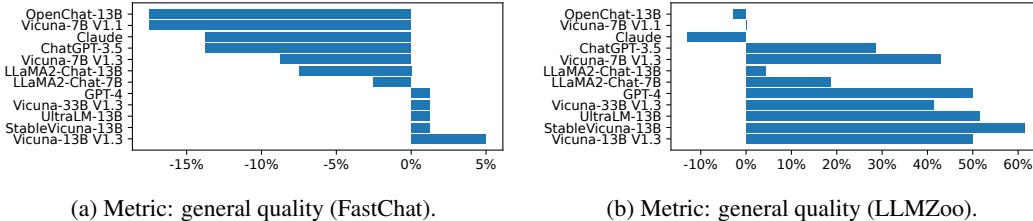

(a) Metric: general quality (FastChat).

(b) Metric: general quality (LLMZoo).

Figure 30: Net win rates of SoT on different models. (Evaluated using ChatGPT-3.5 as the judge.)

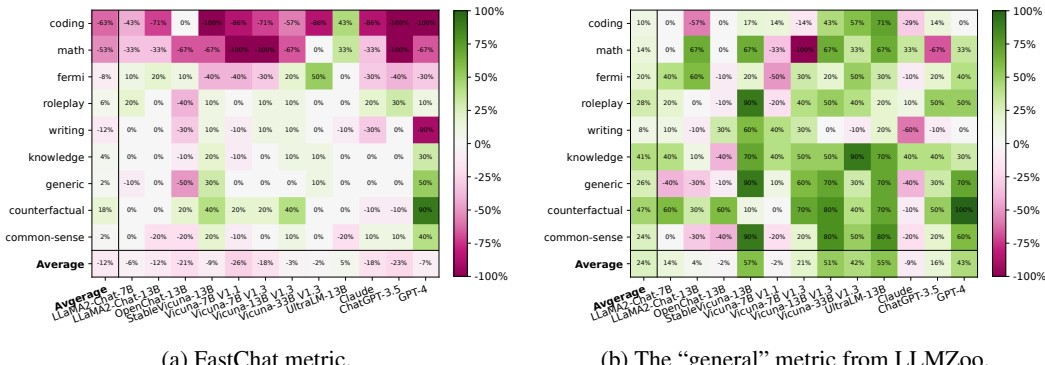

(a) FastChat metric.

(b) The "general" metric from LLMZoo.

Figure 31: Net win rates of different models and question categories. Each row corresponds to one question category, and one column corresponds to one model. (Evaluated using ChatGPT-3.5 as the judge.)

### I.4.5 QUALITY BREAKDOWN: METRICS

All previous evaluations use metrics about the general quality of the answer. In Fig. 32, we show more detailed metrics from LLMZoo to reveal in which aspects SoT can improve or hurt the answer quality. On average, we can see that SoT improves the diversity and relevance while hurting the immersion and coherence.

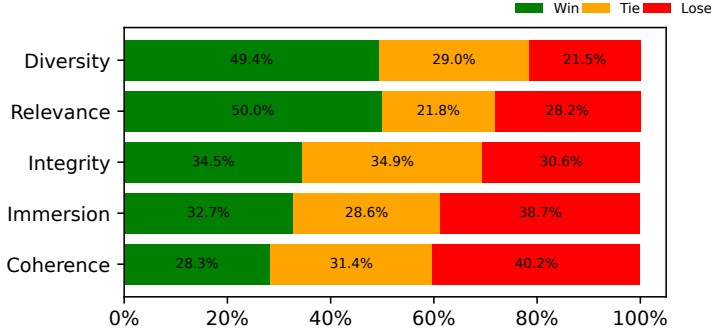

Figure 32: Win/tie/lose rates of SoT v.s. normal generations using metrics from LLMZoo. SoT performs well on diversity and relevance, and relatively worse on coherence and immersion. (Evaluated using ChatGPT-3.5 as the judge.)

## J COMBINING SOT-R WITH MODEL QUANTIZATION

Model quantization is a widely-used model-level optimization to accelerate LLM inference, which is orthogonal to SoT. In this section, we evaluate the speed-ups of open-source models with both

quantization and SoT on the Vicuna-80 dataset. Specifically, we adopt GPTQ (Frantar et al., 2022)[8] to apply 4-bit weight-only quantization and use SoT-R instead of plain SoT.

## J.1   SPEED-UPS OF SOT + QUANTIZATION ON QUANTIZED MODELS

We first compare the latency of the quantized models in the normal and SoT modes to evaluate how much SoT can speed up quantized models. Fig. 33 shows the speed-ups of SoT-R on different quantized models. SoT-R obtain $1.08\times$ to $1.99\times$ speed-ups on all the models. Fig. 34 shows the speed-ups of SoT-R on different categories. We can see that on the five question categories for which SoT can provide high-quality answers (i.e., *knowledge*, *generic*, *common-sense*, *roleplay*, *counterfactual*), SoT-R can speed up the overall answer generation process by $1.07\times$ to $2.38\times$.

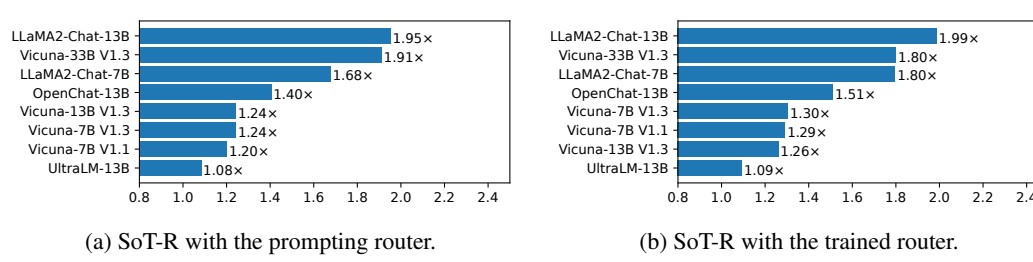

(a) SoT-R with the prompting router.    (b) SoT-R with the trained router.

Figure 33: Speed-ups of the quantized model with SoT-R generation w.r.t. the quantized model with normal generation on different models, on the Vicuna-80 dataset.

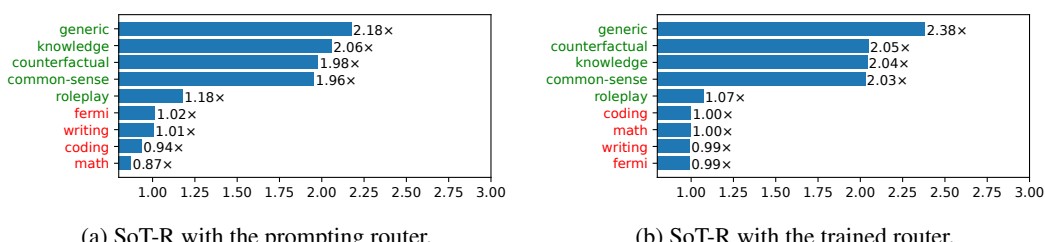

(a) SoT-R with the prompting router.    (b) SoT-R with the trained router.

Figure 34: Speed-ups of the quantized model with SoT-R generation w.r.t. the quantized model with normal generation, on different question categories of the Vicuna-80 dataset.

## J.2   SPEED-UPS OF SOT + QUANTIZATION ON UNQUANTIZED MODELS

Here, we report the overall speed-ups of the quantization model with SoT-R generation w.r.t. the unquantized model with normal generation. Fig. 35 shows the speed-ups of SoT-R on different models. SoT-R can obtain $1.54\times$ to $2.07\times$ speed-ups. Fig. 36 shows the speed-ups of SoT-R on different categories. On the five question categories for which SoT can provide high-quality answers (i.e., *knowledge*, *generic*, *common-sense*, *roleplay*, *counterfactual*), SoT-R can speed up the generation by $1.33\times$ to $3.41\times$ with the prompting and trained routers.

---

[8]https://github.com/qwopqwop200/GPTQ-for-LLaMa

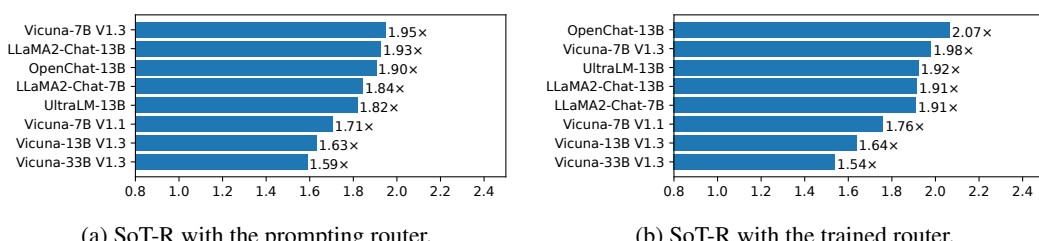

(a) SoT-R with the prompting router.     (b) SoT-R with the trained router.

Figure 35: Speed-ups of the quantized model with SoT-R generation w.r.t. the unquantized model with normal generation, on different models, on the Vicuna-80 dataset.

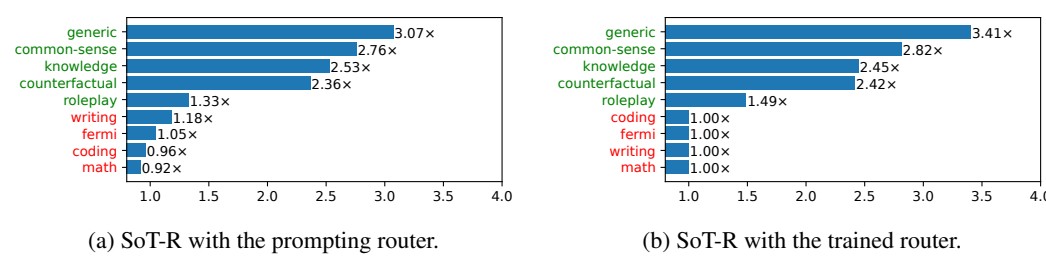

(a) SoT-R with the prompting router.     (b) SoT-R with the trained router.

Figure 36: Speed-ups of the quantized model with SoT-R generation w.r.t. the unquantized model with normal generation, on different question categories of the Vicuna-80 dataset.

## K ADDITIONAL SoT-R STATISTICS

### K.1 NUMBER OF SUITABLE QUESTIONS

Overall, there are 37/80, 58/218, 371/1030 questions that are suitable for SoT in the Vicuna-80, WizardLM, and LIMA datasets (according to human assessment), respectively.

Fig. 37 shows the number of questions that are suitable for SoT on Vicuna-80. On *counterfactual*, *commen-sense*, *knowledge*, *generic* categories, most questions are suitable for SoT based on the human assessment. The trained router and prompting router give out similar judgments.

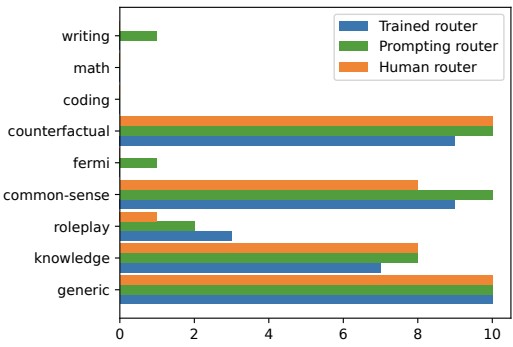

Figure 37: Number of questions suitable for SoT on the Vicuna-80 dataset.

### K.2 PEAK MEMORY OVERHEAD

Fig. 38 and Fig. 39 show the peak memory overhead of SoT-R (with prompting router) on different models and different categories, respectively, on the Vicuna-80 dataset. We can see that, on all models and categories, the overhead of peak memory is quite small ($<1.11\times$).

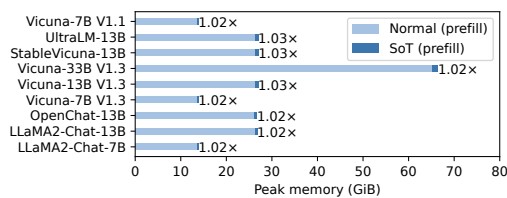
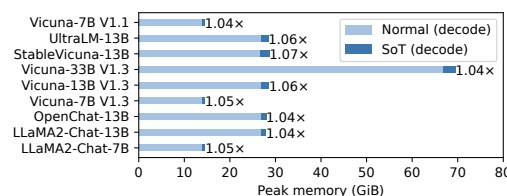

(a) Peak memory in the prefilling phase.      (b) Peak memory in the decoding phase.

Figure 38: Peak memory overhead of SoT-R on different models on the Vicuna-80 dataset.

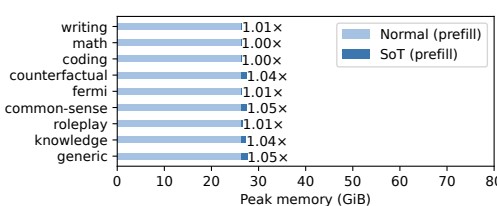
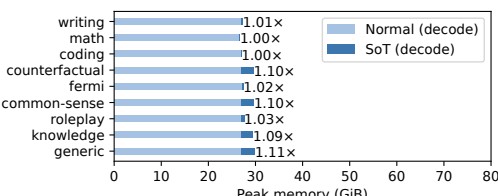

(a) Peak memory in the prefilling phase.      (b) Peak memory in the decoding phase.

Figure 39: Peak memory overhead of SoT-R on different question categories of Vicuna-80.

### K.3 SPEED-UPS WITH DIFFERENT NUMBER OF POINTS

Fig. 40 shows the speed-ups with different numbers of points on Vicuna-80. To maintain clarity in the figure, we've chosen to display statistics for only three models. Note that as SoT cannot control the overall length to be the same as that of normal generation, it is not the case that a higher number of points leads to higher speed-ups.

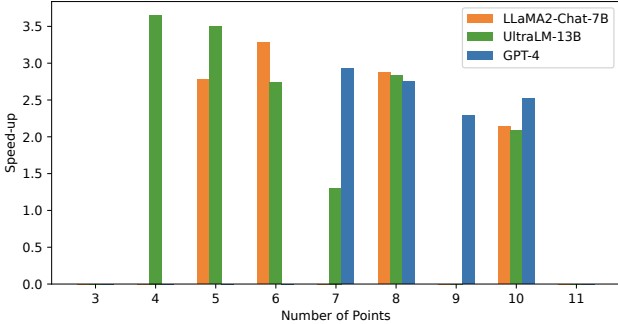

Figure 40: The speed-ups with different number of points on the Vicuna-80 dataset.

## L NOTES ON APPLICATION SCENARIOS

In a chatbot application, one might wonder why a reduced end-to-end latency can enhance the user experience. While human reading speeds are limited, there are many situations where we do not read responses sequentially. Rather than reading the entire answer, one might prefer to (1) swiftly check the response's structure to confirm if the chatbot comprehended the question or (2) extract specific information rapidly without waiting for the generation of prologue or preceding points. Besides, from the quality aspect, even if we would like to check the entire answer, a well-defined structure in responses assists us in quickly parsing all the information.

Moreover, beyond enhancing user experience, reduced end-to-end latency can significantly benefit emerging application scenarios like agent-agent interaction.

