# OpenReview forum: "Skeleton-of-Thought: Prompting LLMs for Efficient Parallel Generation"
_ICLR.cc/2024/Conference — ICLR 2024 poster_

### Official Review · Reviewer_N4tW · 2023-10-30

**Soundness:** 2 fair
**Presentation:** 3 good
**Contribution:** 2 fair
**Rating:** 6
**Confidence:** 4

**Summary:**

This paper proposes a method for speeding up LLM decoding that is typically performed sequentially using a two-step procedure: a) generating the skeleton of the reply (e.g. bullet points) b) expanding the bullet points in parallel through batching or parallel API calls using separate prompts.  Experiments show that this approach can speed up decoding by up to 2.39x and improve the quality of answers in about half the prompt categories & models that they considered based on Fastchat and LLM zoo evaluation. In addition, they propose a backoff strategy that allows to combine the best of sequential decoding and the proposed procedure which leads to better quality overall albeit at the expense of lower decoding speed of up to about 1.8x.

**Strengths:**

- The problem of decoding in parallel is important and worth pursuing with the widespread use of LLMs. In terms of originality, the idea of sketching the reply before generating continuations in parallel is simple and intuitive and a good implementation could have a significant positive impact in the community.
- In terms of exposition and execution, the method is described clearly and the experimentation is thorough as it covers a large number of models & settings.
- Results show that the proposed hybrid method with the backoff strategy to sequential decoding can increase decoding speed by up to 1.80x and improve the quality of the continuations for certain categories (common-sense, generic, counterfactual).

**Weaknesses:**

1. One key weakness of the proposed decoding method is that the generations happen for different bullet points of a reply are independent from each other, limiting its use to answers that require chain-of-thoughts. Hence, it is applicable to specific type of replies and is sensitive to the instruction following abilities of the model that is being applied to, which limits its generality.
2. The proposed method requires prompting the model with a large number of additional tokens to generate continuations for each different bullet point per reply; the number of tokens increases linearly with the number of bullet points. This has both memory/inference cost and latency implications that are not clearly exposed when discussing the main results.
3. When the method is applied without the backoff strategy its quality benefit is not very clear as the improvement is observed in about half the models and categories according to FastChat and LLMZoo evaluations combined.
4. There is no direct comparison to alternative methods that have been proposed for speeding up decoding in autoregressive models such as quantization, batch inference, and speculative decoding. For instance, a simple baseline would be to perform quantization and then apply multi-stage prompting to improve quality for certain categories.

**Questions:**

- From 3.1.1, it is not entirely clear what is the overhead in terms of memory and or latency introduced by the additional prompting required especially for replies that require expansion of many bullet points with and without preflls. Can the authors quantify this? It would show a more clear picture of the tradeoffs involved.
- What is the speed up of the proposed method for different prompt sizes, especially the large ones? It would be useful to show the  improvement for replies that require an increasing number of bullet points to answer.
- In hybrid model case, do you take into account the time required to query the LLM on whether to employ SoT or not? It would be useful to show if it introduces overhead and to what extent.
- Have you tried prompting the model to generate arbitrary partial answer segments other than bullet points to perform continuations on them in parallel? That could potentially make the method applicable beyond answers that require specific bullet point structure.
-  For the quality results in section 3.2.1, what is the average absolute scores achieved per category and across all categories overall when showing quality per model?

---

> ### Author Response · Authors · 2023-11-19
> **Response to Reviewer N4tW (Part 1)**
>
> **W1, about general applicability** One key weakness of the proposed decoding method is that the generations happen for different bullet points of a reply are independent from each other, limiting its use to answers that require chain-of-thoughts.
>
> **A**: Yes, the current SoT method is suitable for only a range of questions. Therefore, we designed a simple extension SoT-R to trigger SoT only for suitable questions by a router module. If we'd like to produce the answer for a broader range of questions in some "structured way" for acceleration and quality, we need to extend the method, for example, by combining the prompting-for-efficiency idea and other structuring prompting techniques to get a better pipeline. Besides prompting, the LLMs themselves might need some adaptation to enable this too. See more discussions of these directions in the global response and Section 6.
>
> **W2, about the overhead**: The proposed method requires prompting the model with a large number of additional tokens to generate continuations for each different bullet point per reply; the number of tokens increases linearly with the number of bullet points. This has both memory/inference cost and latency implications that are not clearly exposed when discussing the main results.
>
> **Q5**: From 3.1.1, it is not entirely clear what is the overhead in terms of memory and or latency introduced by the additional prompting required especially for replies that require expansion of many bullet points with and without preflls. Can the authors quantify this? It would show a more clear picture of the tradeoffs involved.
>
> **A to W2&Q5**: Yes, due to the long prompt, the current SoT introduces additional computation in the prefilling stage.
>
> **In terms of inference latency**, our reported results have already taken the additional prefilling overhead into account. Figure 12 and Figure 13 have shown the latency breakdown of the two phases (prefill and decode). One can see that the prefilling latency is very small compared with the decoding latency.
>
> **In terms of the additional memory footprint brought by batched inference**, Figure 10(c) showed a profiling result. It shows that the peak memory overhead due to the increasing size of the KV cache and activation grows at a slow pace as batch size B increases. Besides the profiling results, following your suggestion, we report the actual peak memory overhead in Appendix K.2 in the revision. The results show that in all models and categories, the peak memory overhead of both the prefilling and decoding phases is quite small (<1.1x).
>
> In Sec. 6 and Appendix H, we discussed the influences caused by the additional computation in different scenarios.
> * When the original concurrent query number is low (i.e., the workload is memory-access-bounded), using SoT can help improve latency without sacrificing throughput, even though it introduces additional computation, as it improves computation utilization.
> * When the original query concurrency is already high (i.e., the workload is already computation-bounded), using SoT trades off the throughput for latency.
>
> Such overhead can be mitigated by the following approaches. (1) Only prefilling the common prefix of the multiple point expansion prompts once, and using the KV cache for batch decoding. (2) Triggering SoT based on the current system workloads. (3) Using shorter prompts to activate the SoT mode. Actually, in our experiments, we found that one can correctly trigger SoT with a much shorter skeleton prompt (without the demonstrations) for strong models such as GPT-4, GPT-3.5, and LLaMA2. (In our experiment, we used one unified prompt for all open-source models without applying this trick.) In addition, applying prompt compression techniques to stronger models could also largely reduce the SoT prompt length, thus lowering the computation overhead.
>
> **W3**: When the method is applied without the backoff strategy its quality benefit is not very clear as the improvement is observed in about half the models and categories according to FastChat and LLMZoo evaluations combined.
>
> **A**: We are not sure if we understood this question correctly. Did you mean that "the quality improvement is *only* observed in about half the models and categories, and *not* the other half"? If this is the question, SoT was originally designed for efficiency improvements. So, as long as the performance does not degrade much, we are satisfied. We regard quality improvements on half the models and categories as a bonus. If we misunderstood this question, please do not hesitate to correct us.

---

> ### Author Response · Authors · 2023-11-19
> **Response to Reviewer N4tW (Part 2)**
>
> **W4**: There is no direct comparison to alternative methods that have been proposed for speeding up decoding in autoregressive models such as quantization, batch inference, and speculative decoding. For instance, a simple baseline would be to perform quantization and then apply multi-stage prompting to improve quality for certain categories.
>
> **A**: Thanks for this valuable suggestion. We follow the suggestion to experiment with model quantization. Model quantization is a model-level optimization approach to accelerate the LLM inference process, which is orthogonal to our SoT and thus can be combined with SoT to achieve a higher speed-up. The following table shows the speed-ups of combining SoT-R and quantization w.r.t. normal generation without quantization. The numbers between brackets are the SoT speed-ups without quantization. More detailed results (e.g., speed-up on different categories, speed-up w.r.t. the quantized model) can be found in Appendix J in our revision.
> | Router | OpenChat-13B | Vicuna-7B V1.3 | UltraLM-13B | LLaMA2-Chat-13B | LLaMA2-Chat-7B | Vicuna-7B V1.1 | Vicuna-13B V1.3 | Vicuna-33B V1.3 |
> | --- | --- | --- | --- | --- | --- | --- | --- | --- |
> | Prompting router | 1.90x (1.66x) | 1.95x (1.39x) | 1.82x (1.51x) | 1.93x (1.70x) | 1.84x (1.67x) | 1.71x (1.55x) | 1.63x (1.32x) | 1.59x (1.62x) |
> | Trained router | 2.07x (1.82x) | 1.98x (1.42x) | 1.92x (1.59x) | 1.91x (1.70x) | 1.92x (1.69x) | 1.76x | 1.64x (1.59x) | 1.54x (1.57x) |
>
>
> **Q6**: What is the speed up of the proposed method for different prompt sizes, especially the large ones? It would be useful to show the improvement for replies that require an increasing number of bullet points to answer.
>
> **A**: We show the speed-ups under different numbers of bullet points in Appendix K.3 in our revision. The speed-up does not increase with the number of bullet points. The reason is that there exist other variables that would affect the speed-up. For example, for questions that yield more bullet points, SoT might give out a much longer response compared with the normal generation process. This will lower the speed-up.
>
> **Q7**: In hybrid model case, do you take into account the time required to query the LLM on whether to employ SoT or not? It would be useful to show if it introduces overhead and to what extent.
>
> **A**: Yes, our results have already taken into consideration the additional latency of the router. The latency number on Vicuna-80 dataset is as follows: 0.65s (0.39s∼1.37s) on average with prompting router and 0.04s (0.008s∼1.55s) on average with trained router. These numbers can be found in Appendix G.2 in our original paper.
>
> **Q8**: Have you tried prompting the model to generate arbitrary partial answer segments other than bullet points to perform continuations on them in parallel? That could potentially make the method applicable beyond answers that require a specific bullet point structure.
>
> **A**: Thanks for proposing this interesting idea. We did not try this. We think it is a valid pathway to improve general applicability. This could also help with the coherence, if the LLM can understand the requirement that the output of one segment will be followed by the given partial segment in the skeleton.
>
> **Q9**: For the quality results in section 3.2.1, what is the average absolute scores achieved per category and across all categories overall when showing quality per model?
>
> **A**: Our original submission gave a per-category-model heatmap on answer quality in Figure 27, which is evaluated by the GPT-3.5 judge. To further answer your question, we added an answer quality heatmap evaluated by the GPT-4 judge to the revision at Figure 22.

---

> ### Author Response · Authors · 2023-11-21
> **Follow-up**
>
> Dear Reviewer N4tW,
>
> Thanks again for your valuable time and insightful questions. Are our answers to your questions satisfactory? If you have any further comments, please do not hesitate to inform us. We are more than willing to provide further discussion.
>
> Many thanks!
>
> Authors

---

> > ### Comment · Reviewer_N4tW · 2023-11-23
> > **Response to rebuttal**
> >
> > Dear authors,
> >
> > Thank you for the detailed responses; would be great if they are reflected in the final version.
> >
> > - Regarding W3 my point is that on half of them there is improvement but on the other half it leads to degradation; it would be nice if improvement doesn't come at the expense of other tasks. The level of degradation is substantial in my view.
> > - The additional results with quantization show that the proposed method is competitive but performs worse than quantization. I am still not convinced in which case one would prefer SoT in place of quantization or other efficient methods, especially if some of the evaluation categories are going to get a hit in performance.
> >
> > For the reasons above, I am maintaining my score.

---

> ### Author Response · Authors · 2023-11-23
> **Thanks to the reviewer**
>
> Dear Reviewer N4tW,
>
> Thanks for your responses and for keeping a positive judgment! Your questions and suggestions are very helpful. We've added the new results and discussions to the revision.
>
> > Regarding W3 my point is that on half of them there is improvement but on the other half it leads to degradation; it would be nice if improvement doesn't come at the expense of other tasks. The level of degradation is substantial in my view.
>
> Oh, we got your point. Yes, the current SoT is not suitable for some question types. Currently, we went with a "router" solution, SoT-R, that simply doesn't use SoT for these questions. Making answer planning work for broader question types indeed needs future improvements.
>
> > The additional results with quantization show that the proposed method is competitive but performs worse than quantization. I am still not convinced in which case one would prefer SoT in place of quantization or other efficient methods, especially if some of the evaluation categories are going to get a hit in performance.
>
> Maybe a point we didn't make clear enough is that we didn't compare SoT with quantization. As quantization and SoT are orthogonal methods, we apply SoT generation to quantized models and report (1) the overall speed-up w.r.t. *the FP model with normal generation* (newly added Appendix J.2 and the table in the response); and (2) the speed-up w.r.t. *the quantized model with normal generation* (newly added Appendix J.1).
>
> Another thing is that we already use SoT-R in all these experiments, i.e., not triggering SoT for unsuitable questions, to avoid hurting the answer quality.
>
> That is to say, the table in the response reports the overall speed-ups of SoT + quantization, where the SoT-alone speed-ups are reported in parentheses. For example, the "1.84x (1.67x)" for LLaMA2-Chat-7B in the table means that SoT-R achieves a 1.67x speed-up , and quantization brings an additional 1.1x speed-up, which multiplies up to 1.84x.
>
> Thanks again!
>
> Authors

---

### Official Review · Reviewer_4Trv · 2023-10-30

**Soundness:** 4 excellent
**Presentation:** 3 good
**Contribution:** 3 good
**Rating:** 6
**Confidence:** 3

**Summary:**

In this work, the authors propose a two-stage generation framework, utilizing an abstraction-generation paradigm for parallel generation, aiming to enhance the generation speed and efficiency of large-scale models. The SoT method demonstrates significant speed-ups across 12 Large Language Models, however, it appears to underperform in tasks such as code generation.

**Strengths:**

1. This paper is well-written and easy to follow.
2. The idea is simple and neat, and it is effective at some specific tasks, e.g., knowledge and role play.
3. The authors conducted extensive and detailed comparative experiments on different models, datasets, and tasks. Moreover, they did not conceal tasks where the performance was poor, such as coding. I think this is highly commendable.

**Weaknesses:**

1. The main contribution of this paper is on the experiments part rather than the idea. In other words, the idea of this article is not insightful because it is very straightforward. Although I believe that skeleton can be generated in different ways, such as through a tree, rather than directly prompting the LLM.
2. If the purpose of this paper is acceleration, then SoT is not universally applicable. The experiments in the article demonstrate it can lead to a decline in the quality of tasks such as code generation. More importantly, because the length (autoregressive steps) of each point the LLM needs to generate varies, the actual acceleration ratio in online-serving may differ from the test scenarios presented in the article, e.g., batch size=1 or batch size > 1
3. I  personally doubt how much parallel generation of n points (i.e., SoT) actually enhances the user experience.
Considering that people read from left to right, generating n points in parallel may not significantly improve the reading experience for users. Instead, we should consider how to accelerate the generation speed of each point. As long as this speed is slightly faster than human reading speed, users might not perceive generation speed as a significant issue, because they also need time to understand the information from the LLM.

**Questions:**

see weakness 3

---

> ### Author Response · Authors · 2023-11-19
> **Response to Reviewer 4Trv**
>
> **W1**: The main contribution of this paper is on the experiments part rather than the idea. In other words, the idea of this article is not insightful because it is very straightforward. Although I believe that the skeleton can be generated in different ways, such as through a tree, rather than directly prompting the LLM.
>
> **A**: Thanks for your recognition of our evaluation! We'd like to share our opinion on the idea: Although the current method design is straightforward, the idea behind SoT -- **prompting an explicit answer planning for efficiency** --  is novel and worth future study, which is enabled by the fact that LLM has increasingly stronger capabilities including instruction following, planning, etc. The idea is to do explicit (by language) answer planning to help make better use of hardware resources (i.e., improving computation utilization) and thus improve efficiency. We think explicitly making the generation workload parallelizable might become a more relevant efficiency-oriented methodology as the LLM gets more widely adopted and stronger.
>
> **W2-1**: SoT is not universally applicable. The experiments in the article demonstrate it can lead to a decline in the quality of tasks such as code generation.
>
> **A**: Yes, the current SoT method is not suitable for all questions. Therefore, we propose a simple extension SoT-R to only trigger SoT on suitable questions. If we'd like to produce the answer for a broader range of questions in some "structured way" for acceleration and quality, we need to extend the method, for example, by combining the prompting-for-efficiency idea and other prompting techniques (e.g., CoT) to get better quality. Besides prompting, the LLMs themselves might need some adaptation to enable this too. See more discussions of these directions in the global response and Section 6.
>
> **W2-2**: Because the length (autoregressive steps) of each point the LLM needs to generate varies, the actual acceleration ratio in online-serving may differ from the test scenarios presented in the article, e.g., batch size=1 or batch size > 1.
>
> **A**: Thanks for pointing out this valuable concern. Speed-up controllability of this data-level technique is a valid concern that we did not cover in our original paper. Unlike most model-level (quantization, static sparse, etc.) and system-level acceleration (fusing, offloading, etc.) techniques, SoT's acceleration ratio depends more on the SoT prompt, the model, and the question. We add a limitation discussion to the "Data-centric efficiency optimization" paragraph in Sec. 6.
>
> As for the case with different batch sizes, Section 6 has discussed the overhead of SoT in two scenarios: memory-access bounded (with a moderate batch size) and computation-bounded (with a large batch size>>1).
>
> **Q3, about the user experience and more application scenario**: How much parallel generation of n points (i.e., SoT) actually enhances the user experience? Considering that people read from left to right, generating n points in parallel may not significantly improve the reading experience for users. Instead, we should consider how to accelerate the generation speed of each point. As long as this speed is slightly faster than human reading speed, users might not perceive generation speed as a significant issue, because they also need time to understand the information from the LLM.
>
> **A**: Thanks for this great question. Our considerations are as follows:
> - **About user experience**: When we are using Chatbots, there are many cases where we only need to check the structure or a certain point of the overall answer. Actually, the unnecessarily long wait for some certain point in the interactive chatbot application is what motivates us to investigate this topic in the first place. We wonder (1) why can't we get the overall structure of the answer first, so that we know whether the chatbot understands the question. If not, we can terminate the generation quickly and rephrase the question, (2) why can't we get the information for a specific point as quickly as possible, when it does not depend on the previous prologue and points. In addition, the user experience might be enhanced with an appropriate interface design. For example, the skeleton can be first outputted to the user, so that the user can check whether the answer structure is as desired, and quickly identify the skeleton points that they want. The user can then click the skeleton points that are most relevant to their interest, which will then expand the detailed answer for that point from the point-expanding stage of SoT.
> - **Other application scenarios**: Besides user experience, application scenarios such as agent-agent interaction can benefit from the reduced latency.
>
> We added this discussion to Appendix L in the revision.

---

> ### Author Response · Authors · 2023-11-21
> **Follow-up**
>
> Dear Reviewer 4Trv,
>
> Thanks again for your valuable time and insightful questions. Are our discussions satisfactory enough? If you have any further comments, please do not hesitate to inform us. We are more than willing to provide further discussion.
>
> Many thanks!
>
> Authors

---

### Official Review · Reviewer_JQY3 · 2023-10-31

**Soundness:** 3 good
**Presentation:** 3 good
**Contribution:** 2 fair
**Rating:** 6
**Confidence:** 3

**Summary:**

This paper is about a prompt-based approach to speeding up inference of LLMs.
The work tries to address the problem by generating "skeleton," which is a set of simple answer sketches, and then based on the skeleton, a model generates answers in parallel.
The main motivation of this generation procedure is that, even if the resulting answer is the same (i.e. the same answer with the naive sequential generation), since applying attention over all previously generated tokens takes a lot of time and accelerator memory (quadratic to the context length), it may be beneficial to split answers into multiple pieces each of which can be generated in parallel.
Each skeleton point is generated via an LLM, and the actual generation process from each skeleton point is done by calling generation queries in parallel (in the case of models that can only be accessed via API) or by batching multiple prompts.
From evaluation, it is empirically proven that the suggested algorithm speeds up generation speed of various models, from 1.13x to 2.39x, across multiple question domains.
To address the issue that the suggested SoT algorithm cannot perform step-by-step generation e.g. chain-of-thought, the authors add LLM or trained model called router to determine whether or not to use SoT, namely SoT-R. Nonetheless of its simple architecture, the router module reduced failure cases where SoT underperforms normal sequential generation, in exchange of reduced speedup.

**Strengths:**

- Good empirical motivation: sentences of generated text may be independent to each other and thus the generation can be done in parallel.
- Good use of LLM prompts: generating skeletons, (some) routing models, and actually generate text from skeleton with small disruption

**Weaknesses:**

- The content relies too much on supplementary materials. The manuscript is preferred to be self-contained.
- Overall, throughput given a fixed amount of compute resource may decrease, due to additional overheads due to skeleton generation and repeated prompts in each parallel generation prompt, since each of which will consume resource even if they are done in parallel.

**Questions:**

- As mentioned in above, as far as I understood this algorithm may use more resource than normal sequential generation. Do you have numbers about overall resource time used for a single generation? i.e. for API-based experiments it would be (skeleton generation time + sum of all parallel execution times) and for batch-based experiments it would be (skeleton generation time + (batch generation time * sequences per minibtach * number of minibatches) ).
- For router experiments, do you have numbers about the ratio of each instance being classified as "capable of being generated via SoT"? For example in Figure 7, StableVicuna-13B with SoT-R experiments achieved ~1.0x speedup, which means that most of them would be generated via normal sequential generation.
- Is naturalness of generated text included in metrics of automatic quality evaluation? The prompt used feels not very natural for me; since as far as I understood each sentence should be always start the "skeleton point," according to Prompt 2 ("... do not continue with other points! [Assistant:] {point index}. {point skeleton}")

---

> ### Author Response · Authors · 2023-11-19
> **Response to Reviewer JQY3 (Part 1)**
>
> **Q1**: The content relies too much on supplementary materials. The manuscript is preferred to be self-contained.
>
> **A**: Many thanks for the suggestion. Given the extensive results on hardware profiling, efficiency analysis, answer examples, failure case analysis, and more, it is difficult to include everything in 9 pages. We tried hard to make sure that the discussion and experiments in the main text were sufficient for readers to understand the key points. Although the main text indeed refers to the appendix in many places (where the appendix provides more supporting experiments and detailed discussions), these references mainly serve as a convenient pointer to the corresponding appendix section.
>
> That being said, if the reviewer thinks any of the contents in the appendix should be elevated to the main text, please let us know and we will make the changes.
>
> **Q2&Q3, regarding the overhead**: Overall, throughput given a fixed amount of compute resource may decrease, due to additional overheads due to skeleton generation and repeated prompts in each parallel generation prompt, since each of which will consume resources even if they are done in parallel. Do you have numbers about the overall resource time used for a single generation?
>
> **A**: You are right that the SoT prompts introduce computation overhead. Will this overhead actually cause throughput degradation? We gave the computation overhead results in Appendix H, and gave a short analysis in Section 6 by looking at two types of scenarios (with or without a large number of concurrent queries).
>
> The discussion said that when the original concurrent query number is low (i.e., the workload is memory-access-bounded), using SoT can help improve latency without sacrificing throughput, even though it introduces additional computation, as it improves the computation utilization. When the original query concurrency is already high (i.e., the workload is already computation-bounded), using SoT might hurt the throughput, and does not help so much with the latency either.
>
> Such overhead can be mitigated by the following approaches.
> 1. Only prefilling the common prefix of the multiple point expansion prompts once, and using the KV cache for batch decoding.
> 2. Triggering SoT based on the current system workloads.
> 3. Using shorter prompts to activate the SoT mode. Actually, in our experiments, we found that one can correctly trigger SoT with a much shorter skeleton prompt (without the demonstrations) for strong models such as GPT-4, GPT-3.5, and LLaMA2. (In our experiment, we used one unified prompt for all open-source models without applying this trick.) In addition, applying prompt compression techniques to stronger models could also largely reduce the SoT prompt length, thus lowering the computation overhead.

---

> > ### Comment · Reviewer_JQY3 · 2023-11-21
> >
> > Dear authors,
> > Thank you for the response.
> >
> > The main reason I mentioned about throughput is in the case where there is limitation in resource and are multiple concurrent queries, i.e. the number of queries that a model can process given a fixed amount of time and machine; in my understanding due to increased number of total tokens, and assuming that the number of tokens that a single machine can process per second doesn't change, it will decrease the throughput.
> > However as you mentioned in the response, the use of KV caching for common prefixes will make the overhead introduced by appending prefixes as small as possible (i.e. minimizing overheads introduced by the proposed algorithm), and for some cases (where the target text to be generated is longer enough), applying parallel decoding can possibly decrease the length thus lead to reduced computation (i.e. actually increasing throughput).
> > I can't judge whether it will result in increased or decreased throughput considering all positive and negative sides onto throughput, however thanks to the authors' explanation I think my understanding made more clear.

---

> > > ### Author Response · Authors · 2023-11-21
> > >
> > > Dear Reviewer JQY3,
> > >
> > > Thanks for your fast reply and explanation of your concern! Indeed, the decreased throughput in computation-bounded scenarios is an important concern.
> > >
> > > To optimize the throughput and latency in the meantime, how to integrate the intra-user parallelism introduced by SoT into the existing serving frameworks needs careful engineering considerations: (1) What is the best way for the newly introduced intra-user parallelism to leverage the multi-user batching mechanism in serving frameworks; (2) How can we choose to trigger SoT based on the current serving workload.
> > >
> > > If there are any further discussions you'd like to have, contact us at any time!

---

> ### Author Response · Authors · 2023-11-19
> **Response to Reviewer JQY3 (Part 2)**
>
> **Q4**: For router experiments, do you have numbers about the ratio of each instance being classified as "capable of being generated via SoT"? For example in Figure 7, StableVicuna-13B with SoT-R experiments achieved ~1.0x speedup, which means that most of them would be generated via normal sequential generation.
>
> **A**: Thanks for this question. On Vicuna-80, on the counterfactual, common-sense, knowledge, and generic categories, the number of problems that are suitable for SoT mode is very high (>7/10).  While on the other categories, the number of suitable examples for SoT is quite small, especially on the math and coding categories. More detailed statistics can be found in Appendix K.1 in the revision.
>
> The reason that StableVicuna cannot achieve good speed-ups is not relevant to the number of questions suitable for SoT, but lies in the shortage of StableVicuna in understanding the point expansion task (it generates long answers in the point-expanding stage regardless of the short answer requirement in the prompt). We can see from Figure 2(a) that plain SoT can only achieve an on-average 1.1x speed up on StableVicuna.
>
> **Q5**: Is naturalness of generated text included in metrics of automatic quality evaluation? The prompt used feels not very natural for me; since as far as I understood each sentence should be always start the "skeleton point," according to Prompt 2 ("... do not continue with other points! [Assistant:] {point index}. {point skeleton}")
>
> **A**: Thanks for this great question. You are right that each point starts with a partial answer. We used the partial answer to avoid useless outputs from LLMs such as "Sure, I'll help you to expand the 2nd point", so that we can directly concatenate the point responses as the final response. For some types of questions, this might result in unnatural responses. We have included a discussion on this pattern in Appendix I.1.2 in the original manuscript.
>
> > Writing problems usually require a coherent passage without embedded skeleton points, whereas our current SoT pipeline (§ 2) concatenates skeleton points as part of the answer In such cases, simply removing the skeleton points would greatly improve the answer quality. To make SoT more general, one future direction is to let the LLM itself decide whether the point index and point skeleton should be included to make the final answer more natural and fluent.
>
> Regarding the metric definitions: Although the two sets of metrics (FastChat, LLMZoo) don't have exactly the word "naturalness" in their evaluation prompts, the definition of the "Coherence" and "Immersion" metrics are actually related. And indeed, our evaluation in Figure 6 found that the current SoT solution hurt immersion and coherence.
>
> In our early experiment, we made some attempts to mitigate this issue: we asked the model itself to judge if the point skeleton should exist in the final response, and exclude the partial answer part in the final response if the judgment is no. We found that strong models such as GPT-3.5 can give judgments that match manual justifications, but unfortunately, the open-source models cannot give good judgments on this aggregation scheme. As the attempt is not thorough and quantitive, we did not include it in the manuscript and deferred it to future work.

---

> ### Author Response · Authors · 2023-11-21
> **Follow-up**
>
> Dear Reviewer JQY3,
>
> Thanks again for your valuable time and insightful questions. Are our discussions satisfactory enough? If you have any further comments, please do not hesitate to inform us. We are more than willing to provide further discussion.
>
> Many thanks!
>
> Authors

---

### Official Review · Reviewer_ieuZ · 2023-10-31

**Soundness:** 2 fair
**Presentation:** 3 good
**Contribution:** 3 good
**Rating:** 8
**Confidence:** 4

**Summary:**

This paper proposes a parallelization for answering questions with large language models. The core idea is to formulate the answer as a list of points (the skeleton), and then elaborate each of them independently with parallel model calls. This is limited to answering a type of questions for which such independent elaborations of a list of points are suitable. The decision whether a question is suitable can be passed to a router, which is an extension that the paper proposes. The parallelization brings about a speedup compared to sequential decoding, and the authors argue that this also mimics the way that humans think.
The method is evaluated for 12 different LLMs on FastChat and LLMZoo, and the results are discussed under various perspectives.

Score was raised after author response.

**Strengths:**

- It’s a good idea and well executed. It will probably inspire future works, as there are a few ideas that could easily extend the proposed method, e.g. for improving coherence of the elaborated points.
- For some models there are efficiency AND quality improvements, which is very attractive for practical adoption in the cases where questions fall into the categories that are suitable for the method.
- The proposed methods are widely evaluated, covering multiple LLMs and benchmarks. Especially having a large number of LLMs tested is helpful for gaining insights into variability and sensitivity of the models and setting expectations for applying the method to other models. There’s also a large number of analyses and more results in the appendices, which overall give the impression that the authors have profoundly studied the effectiveness of their method.

**Weaknesses:**

- The comparison with human thinking is merely intuitive and not founded by any references, hence speculation. I do not think this makes the method any more attractive, and would prefer to see a discussion with scientific evidence, if this should be a benefit of this method. It is clear that the inspiration comes from how humans draft a text or the like - but this does not result in humans and LLMs functioning in the same way. The phrasing that LLMs are made “thinking in the same way” just adds fuel to the pseudoscientific discussions around whether LLMs are “thinking” or not (especially the phrase “pushing LLMs to think more like a human”). In addition, there is no connection between the argument of efficiency and human thinking - is it for efficiency that humans derive a skeleton first?
Missing comparison to previous orthogonal methods proposed for speedups: Where on Figure 1b would they be located? This is necessary to understand the impact of this work in context of the previously proposed and adopted solutions.
- The title is misleading and alludes to a new ability of LLMs (“LLMs can do”): The actual LLM decoding process is technically still sequential and there is no modification to the LLMs decoding loop (not to be confused with non-autoregressive methods), it’s just that the model is prompted to provide a decomposable answer to be elaborated in parallel. A more concise (but probably not great for selling the paper) reformulation of the title would be “LLMs can be prompted to provide decomposable answers”.
- The limitations of the method are not clearly articulated in the introduction: until Section 3 it is not clear that this method is limited to specific types of questions and that there is prompt-specific sensitivity. It might be obvious to the authors, but for positioning the paper in the right scope, this should be mentioned in the introduction.
- An analysis of the effect of added length is missing: Since skeleton-of-thought prompting produces longer answers than the usual prompting (3.1.1), there might be a bias of certain evaluations to prefer more wordy answers even if they have the same contents. Similarly, it remains an open question how the length part (“3-5 words”/”3-10 points”/… see prompt examples) of the prompt should be optimally configured.

**Questions:**

- How would Figure 1b look like for the LLMZoo evaluation? This kind of plot is answering the question for the speed-quality trade-off at one glance, so it would be great to have as well for LLMZoo. It looks like the model with the largest wins (StableVicuna-13B) also has the lowest speed-up for LLMZoo (3.2.2), so it would be nice to see if this is a trend more than for FastChat.
- I do not think that Figure 3 is very expressive, as it is an accumulated result across all models, which in turn (3.2.2) vary largely. Perhaps this could be replaced with a deeper analysis of the loss cases? This would allow the reader to get a better idea of the potential risks.
- The method reminds me of the map-reduce paradigm, but the reduce step is a simple concatenation rather than a joint processing of the whole list of answers. How much would a final “summarize” decoding step add to the quality and remove from the efficiency? I could imagine this increasing the coherence of the answers.
- For 4.3.1 the most relevant question seems to be which ratio of questions the router decides to process with skeleton-of-thought. This directly affects efficiency, since speedups can only possibly be obtained if the skeleton-of-thought route is chosen. Quality and efficiency both seem to be a predictable combination of normal decoding and skeleton-of-thought decoding quality and efficiency once the ratio is known. It would be interesting to report this ratio by category in order to evaluate if it agrees with the categories that were identified to work well for skeleton-of-thought in 3.1.2.

---

> ### Author Response · Authors · 2023-11-19
> **Response to Reviewer ieuZ (Part 1)**
>
> **Q1**: The comparison with human thinking is merely intuitive. It is clear that the inspiration comes from how humans draft a text or the like - but this does not result in humans and LLMs functioning in the same way. The phrasing that LLMs are made “thinking in the same way” just adds fuel to the pseudoscientific discussions around whether LLMs are “thinking” or not (especially the phrase “pushing LLMs to think more like a human”). In addition, there is no connection between the argument of efficiency and human thinking - is it for efficiency that humans derive a skeleton first?
>
> **A**: Thanks for this valuable suggestion! We used these descriptions to give readers an intuitive understanding of the idea, but we agree with the reviewer that they are not well-defined and are not suitable for papers. Upon your suggestion, we also realize that these sentences could mislead the readers to think if we are claiming there is a connection between the argument of efficiency and human thinking (which is not). We revised the paper according to your suggestion.
>
> Except for these two sentences, we think the statements and logic in our introduction and discussion section are solid. The main logic we want to express in the introduction is: (1) The fully sequential decoding is one of the key reasons that cause slow generation. Is this fully sequential way necessary? (2) Humans usually write the answer in a structured, non-sequential way. For example, we usually have a skeleton of the idea in our minds or on paper first. This intuition has our back to question the necessity of fully sequential decoding. (3) Inspired by this, we propose SoT as an attempt towards "data-level organization for efficiency".
>
> Another place we relate SoT to human thinking is in the discussion in Sec. 6: We first relate SoT’s potential of enhancing answer quality to other recent research like CoT, ToT, and ReAct: These work shows that "explicitly articulating the thought process (e.g., CoT) or structure (e.g., SoT) in language can elicit high-quality answers". Then, we claim that "these findings resemble human thinking: rather than relying solely on first intuition or purely sequential thinking, we often document step-by-step reasoning or thought organization to attain high-quality answers".
>
> **Q2**: The title is misleading and alludes to a new ability of LLMs (“LLMs can do”): The actual LLM decoding process is technically still sequential and there is no modification to the LLMs decoding loop (not to be confused with non-autoregressive methods).
>
> **A**: We understand and carefully discussed your concern. As we want to convey the key messages including "prompting", "answer planning", "parallel generation", and "efficiency", we are considering a new title "Skeleton-of-Thought: Prompting LLMs for Efficient Parallel Generation", which contains the messages and avoids the confusion you brought up. At the current review stage, we are not allowed to modify the title. We will make the change in the final revision.  If the reviewer has more comments on this new title, please let us know!
>
> **Q3**: The limitations of the method are not clearly articulated in the introduction: until Section 3 it is not clear that this method is limited to specific types of questions and that there is prompt-specific sensitivity. It might be obvious to the authors, but for positioning the paper in the right scope, this should be mentioned in the introduction.
>
> **A**: Thanks for this suggestion. We add an emphasis to the introduction that SoT is not suitable for some types of questions, which motivates us to develop the SoT-R extension. About prompt-specific sensitivity, we mention it in the "Data-centric efficiency optimization" paragraph in Sec. 6 in the revision.

---

> ### Author Response · Authors · 2023-11-19
> **Response to Reviewer ieuZ (Part 2)**
>
> **Q4**: Missing comparison to previous orthogonal methods proposed for speedups: This is necessary to understand the impact of this work in context of the previously proposed and adopted solutions.
>
> **A**: Thanks for this valuable suggestion. Here, we add an experiment using the widely used quantization optimization. Model quantization is a model-level optimization approach, which is orthogonal to SoT and thus can be combined with SoT to achieve a higher speed-up. The following table shows the speed-ups of combining SoT-R and quantization w.r.t. normal generation without quantization for different models. The numbers between brackets are the SoT-R speed-ups without quantization. More detailed results (e.g., speed-up on different categories, speed-up w.r.t. the quantized model) can be found in Appendix J in our revision.
> | Router | OpenChat-13B | Vicuna-7B V1.3 | UltraLM-13B | LLaMA2-Chat-13B | LLaMA2-Chat-7B | Vicuna-7B V1.1 | Vicuna-13B V1.3 | Vicuna-33B V1.3 |
> | --- | --- | --- | --- | --- | --- | --- | --- | --- |
> | Prompting router | 1.90x (1.66x) | 1.95x (1.39x) | 1.82x (1.51x) | 1.93x (1.70x) | 1.84x (1.67x) | 1.71x (1.55x) | 1.63x (1.32x) | 1.59x (1.62x) |
> | Trained router | 2.07x (1.82x) | 1.98x (1.42x) | 1.92x (1.59x) | 1.91x (1.70x) | 1.92x (1.69x) | 1.76x | 1.64x (1.59x) | 1.54x (1.57x) |
>
>
>
> **Q5**: An analysis of the effect of added length is missing: Since skeleton-of-thought prompting produces longer answers than the usual prompting (3.1.1), there might be a bias of certain evaluations to prefer more wordy answers even if they have the same contents. Similarly, it remains an open question how the length part of the prompt should be optimally configured.
>
> **A**:
> **Answer lengths might bias the evaluation.** Thanks for raising this valuable concern. Following your suggestion, we added an experiment: we let the LLM sequentially output a longer answer by using the prompt template "Please give a long answer for the following question: {request}" so that the answer length is comparable to the one from SoT, and then compare the lengthy sequential answer and the SoT answer. The figures are shown in Figure 26-27 in the revision (Appendix I.3). We can see that SoT does not cause severe quality degradation with roughly the same answer length.
>
> **How the length part of the prompt should be optimally configured.** These words and concrete numbers were crafted on Vicuna-7B V1.1 manually and directly used for other open-source models. That being said, we did observe some differences in how models obey the constraints described by these words (e.g., see the total lengths of answers in Figure 11). How different models can obey these number-of-words or number-of-points commands is indeed an interesting direction to study on its own. We note that a recent paper [1] has studied this topic, and it will be interesting to incorporate their insights to improve SoT.
>
> [1] Sun, Jiao, et al. "Evaluating Large Language Models on Controlled Generation Tasks." arXiv preprint arXiv:2310.14542 (2023).
>
> **Q6**: How would Figure 1b look like for the LLMZoo evaluation? This kind of plot is answering the question for the speed-quality trade-off at one glance, so it would be great to have as well for LLMZoo.
>
> **A**: We have added the speed-quality trade-off plot to Figure 25 in the revision. LLMZoo and FastChat give different net win rate results. Nevertheless, the general message (which categories and models are more suitable for SoT) is consistent.
>
> **Q7, writing suggestion**: I do not think that Figure 3 is very expressive, as it is an accumulated result across all models, which in turn (3.2.2) vary largely. Perhaps this could be replaced with a deeper analysis of the lose cases? This would allow the reader to get a better idea of the potential risks.
>
> **A**: We agree with the reviewer: the fact that different models have large quality variances makes Figure 3 less informative, and qualitative knowledge should be discussed on a per-model basis. Nevertheless, when we wrote the draft, we finally decided to put Figure 3 so that it is easy for readers to grasp the overall answer quality of SoT without the need to read more complicated discussions and figures later on. A full analysis of the failure case is discussed in the main paper with examples attached in the appendix for readers to look up.

---

> ### Author Response · Authors · 2023-11-19
> **Response to Reviewer ieuZ (Part 3)**
>
> **Q8, about method extension**: How much would a final “summarize” decoding step add to the quality and remove from the efficiency? I could imagine this increasing the coherence of the answers.
>
> **A**: Thanks for this valuable idea. We had some considerations and attempts in this direction -- using another aggregation method other than simple concatenation. (1) Post-processing relying on a language model (i.e., a summarization decoding step): Using the autoregressive decoding method in this summarization step will incur significant latency. As this work's main objective is to maximize hardware utilization and accelerate generation, we did not experiment with this method. An optimization to this idea could be to use a smaller model for summarization, which is feasible since the main structure and facts are already there. (2) Prompting the LLM to choose from a few manually designed aggregation choices, for example, whether or not to keep the "1. 2. 3." indexes in the final response. We find that GPT-3.5 can give reasonable judgment according to our human justification, but open-source models cannot give reasonable choices yet. We did not conduct large-scale experiments at this point and defer this to future work.
>
> **Q9, SoT-R statistics**: For 4.3.1 the most relevant question seems to be which ratio of questions the router decides to process with skeleton-of-thought. It would be interesting to report this ratio by category in order to evaluate if it agrees with the categories that were identified to work well for skeleton-of-thought in 3.1.2.
>
> **A**: Thanks for this suggestion. On the Vicuna-80 dataset, on the counterfactual, common-sense, knowledge, and generic categories, the number of questions that are suitable for SoT mode is very high (>7/10) based on our routers. While on the other categories (especially math and coding), the number of suitable questions for SoT is quite small. The results show agreement with the results in 3.1.2. We've added the statistics to Appendix K.1 in the revision.

---

> ### Author Response · Authors · 2023-11-21
> **Follow-up**
>
> Dear Reviewer ieuZ,
>
> Thanks again for your valuable time, insightful questions, and suggestions. We would like to know your thoughts on our responses. If you have any further comments, please do not hesitate to inform us. We are more than willing to provide further discussion.
>
> Many thanks!
>
> Authors

---

> > ### Comment · Reviewer_ieuZ · 2023-11-22
> > **Response**
> >
> > Thank you so much for the thorough response and additional results, they're very insightful.

---

> > > ### Author Response · Authors · 2023-11-23
> > > **Thanks for the reviewer**
> > >
> > > Dear Reviewer ieuZ,
> > >
> > > Thank you for your feedback. We appreciate your time and constructive discussions. They are very helpful!
> > >
> > > Thanks again
> > >
> > > Authors

---

### Official Review · Reviewer_KKRk · 2023-11-03

**Soundness:** 2 fair
**Presentation:** 3 good
**Contribution:** 2 fair
**Rating:** 5
**Confidence:** 4

**Summary:**

This paper focuses on the high generation latency of LLM and explores utilizing COT to generate a skeleton of the answer, and then continue to generate the remaining answer parallelly for inference efficiency.

**Strengths:**

Speed up the process of generation

**Weaknesses:**

1. There are requirements for tasks, and multiple points of tasks must be able to be generated at ordinary times without affecting each other.
2. This method of generation may compromise the quality of the results generated.

**Questions:**

1. In a particular training dataset, how many examples can be used with SoT?
2. How does SoT perform in other general datasets?
3. Is there anything wrong with the content generated this way? Such as repeating generation or generating blank.

---

> ### Author Response · Authors · 2023-11-19
> **Response to Reviewer KKRk**
>
> **Q1&Q2&Q4, about the general applicability**: There are requirements for tasks, and multiple points of tasks must be able to be generated at ordinary times without affecting each other. This method of generation may compromise the quality of results. How does SoT perform in other general datasets?
>
> **A**: Thanks for this question! You are right that the current SoT method is mainly suitable for questions that require long answers whose structure can be planned ahead and contain multiple points that are relatively independent, while not suitable for questions that require step-by-step reasoning.
>
> We want to note that the the three datasets used in this paper (Vicuna, WizardLM, LIMA) *already contain both types of questions that are suitable and not suitable for SoT*. In fact, a large fraction of the questions in these datasets are not suitable for SoT (see Appendix K.1). That is why we see in Figure 8 that SoT compromises the answer quality in some question categories. Therefore, we designed a simple extension SoT-R to trigger SoT only for suitable questions by a router module, and we see that SoT-R does not compromise answer quality in all question categories in these datasets.
>
> Beyond SoT-R, there are many other potential pathways to expand SoT's applicability. For example, we discussed the outlook of extending "skeleton-of-thoughts" to "graph-of-thoughts", and "ahead-of-time planning" to "just-in-time planning (a.k.a. dynamic graph-of-thoughts)". We can not fully explore all these potentials in this single paper, but all of them are worth exploring in the future.
>
> **Q3, SoT-R statistics**: How many examples can be used with SoT?
>
> **A**: Thanks for this question! We have updated this information to Appendix K.1 in our revision. For the datasets used by this paper, there are 37/80, 58/218, 371/1030 questions that are suitable for SoT in the Vicuna, WizardLM, and LIMA datasets (according to human assessment), respectively. For Vicuna-80 dataset, on counterfactual, commen-sense, knowledge, and generic categories, most of the questions (>7/10) can be used with SoT based on both the human assessment and our routers. While on the other categories (especially math and coding), the number of suitable questions for SoT is quite small.
>
> **Q5, failure patterns**: Is there anything wrong with the content generated this way? Such as repeating generation or generating blank.
>
> **A**: We did not observe issues like repeating generation or generating blank. Nevertheless, applying SoT on unsuitable questions indeed results in unreasonable answer skeleton and wrong reasoning (for math and fermi), and empty talk about the strategy without giving the actual code (for code). These cases are discussed in Appendix I.1.

---

> ### Author Response · Authors · 2023-11-21
> **Follow-up**
>
> Dear Reviewer KKRk,
>
> Thanks again for your valuable time and insightful questions. We would like to know your thoughts on our responses. If you have any further comments, please do not hesitate to inform us. We are more than willing to provide further discussion.
>
> Many thanks!
>
> Authors

---

### Official Review · Reviewer_vbj6 · 2023-11-03

**Soundness:** 3 good
**Presentation:** 4 excellent
**Contribution:** 2 fair
**Rating:** 3
**Confidence:** 4

**Summary:**

This paper presents a very simple method for reducing the inference latency of LLM. Given a query, the LLM first generates a list of key points, referred to as the "skeleton" in this study. Subsequently, each point is elaborated in parallel via separate LLM executions. The simultaneous execution of these runs reduces the overall inference latency.

**Strengths:**

- This paper is extremely well-written.

**Weaknesses:**

- The core idea is quite straightforward: first list the key points, then elaborate each point. I am afraid that similar ideas have been extensively studied in the literature of NLP (if the authors search hierarchical text generation or planning for text generation or long text generation). Yet this paper gives no discussion. Below I list several references [1][2] (but clearly not sufficient).
- The proposed method only makes sense when the answer is long, decomposable into several key points, and the interdependence of these key points is low (because the writing of each point only sees the skeleton).
- As a result, the proposed method inevitably hurts the overall coherence of the answer. The experiments in this paper also confirm that the proposed method hurts the immersion and coherence judged by GPT4 (Section 3.2.4).
-  All evaluation is conducted by GPT4 or ChatGPT-3.5. The results could be biased or misleading, human evaluation is required.

[1] A Hierarchical Neural Autoencoder for Paragraphs and Documents (ACL15)

[2] Long and Diverse Text Generation with Planning-based Hierarchical Variational Model (EMNLP19)

**Questions:**

n/a

---

> ### Author Response · Authors · 2023-11-19
> **Response to Reviewer vbj6 (Part 1)**
>
> **Q1**: The core idea is quite straightforward. I am afraid that similar ideas have been extensively studied in the literature of NLP. Yet this paper gives no discussion.
>
> **A**: Thanks for referring to these works on hierarchical text generation. We have added the discussion to Appendix D.3 in the revision. As we position our method as a data-level efficient LLM method, we previously discussed SoT in the efficient LLM literature. However, we agree that previous hierarchical text generation studies indeed share some ideas with SoT (e.g., structured generation), and we should discuss them.
>
> The high-level differences between SoT from this literature are:
> - Objective: Prior studies in hierarchical text generation all focus on enhancing the answer quality, including improving the long-range coherence, relevance to the topic, or reducing redundancy. They still employ sequential word-by-word generation without parallelization between sentences.
> - Methodology: Instead of designing new hierarchical architectures or planning modules, SoT exploits the emerging planning and instruction-following abilities of LLMs to do explicit (which means the plan is described by interpretable language) and free-form planning. This allows SoT to be applied to off-the-shelf LLMs for producing structured answers.
>
> As we summarized in the global response, we think this **"prompting an answer plan for efficiency"** idea has its own merits and is worth future study. We appreciate this suggestion and think the hierarchical text generation literature might be a mine of insights for further extending SoT's idea to enable high-quality structured generation in the LLM era.
>
> **Q2, about the general applicability**: The proposed method only makes sense when the answer is long, decomposable into several key points, and the interdependence of these key points is low (because the writing of each point only sees the skeleton). As a result, the proposed method inevitably hurts the overall coherence of the answer.
>
> **A**: You are right that the SoT method is mainly suitable for questions that require long answers whose structure can be planned ahead and contain points that are relatively independent, while not suitable for questions that require a short answer (no structure to plan and exploit) or step-by-step reasoning (points or steps are dependent). We share our perspective as follows.
>
> First, **in our daily usage of chatbots, quite some questions require a long and structured answer with independent points**. Actually, the unnecessarily long wait before getting the answer structure and certain key points in the interactive chatbot application is what motivates us to investigate this topic in the first place. As for the datasets used by this paper, there are 37/80, 58/218, 371/1030 questions that are suitable for SoT in the Vicuna, WizardLM, and LIMA datasets (according to manual justification), respectively. And we have designed a simple extension SoT-R to trigger SoT only for suitable questions by a router module for practical use.
>
> Second, we **foresee a lot of potential pathways to improve the applicability following the idea of "explicit answer structure planning"**. For example, Section 6 discussed the outlook of extending "skeleton" to "graph" (with parallelizable subgraphs for acceleration, and dependencies for coherence), and extending "ahead-of-time planning" to "just-in-time planning" (a.k.a. dynamic graph-of-thoughts). Another potential solution is to let LLM self-trigger SoT generation mode during the middle of generation.

---

> ### Author Response · Authors · 2023-11-19
> **Response to Reviewer vbj6 (Part 2)**
>
> **Q3**: All evaluation is conducted by GPT-4 or ChatGPT-3.5. The results could be biased or misleading, human evaluation is required.
>
> **A**: When we did the experiments, we carefully considered what the best evaluation strategy was.  As discussed in Section 6, we did not think human evaluation is a good strategy here, as it is easy for a human to tell whether an answer is generated with SoT due to its distinctive pattern, which causes evaluation bias. Recent work [1] also shows that human evaluation could be biased towards good styles (e.g., in a list of points, which SoT forces the answers to have) instead of the correct content. We therefore resort to LLM-based approaches. To make the evaluation more comprehensive, we used two types of judges (GPT-4 and ChatGPT-3.5) and two sets of judge prompts (FastChat and LLMZoo).
>
> In addition, to take the GPT-4's evaluation bias on the answer length into consideration, we follow Reviewer ieuZ's suggestion to add a comparison between a longer sequential answer and the SoT answer. The results in Appendix I.3 in the revision show that, when the overall answer lengths are similar, the quality of the SoT answer is comparable to that of the long normal answer.
>
> Nevertheless, we agree with the reviewer that the current evaluation approach in the paper is far from perfect due to the possible bias of LLM judges (which we discussed in Sec. 6). We have taken the best strategy we could think of. More generally, how to evaluate LLMs is an open and heavily discussed question in the community and is worth further studying in the future.
>
> [1] Gudibande, Arnav, et al. "The false promise of imitating proprietary llms." arXiv preprint arXiv:2305.15717 (2023).

---

> ### Author Response · Authors · 2023-11-21
> **Follow-up**
>
> Dear Reviewer vbj6,
>
> Thanks again for your valuable time and helpful suggestions. We would like to know your thoughts on our responses. If you have any further comments, please do not hesitate to discuss them with us.
>
> Many thanks!
>
> Authors

---

> ### Comment · Reviewer_vbj6 · 2023-11-22
>
> I thank the authors for their detailed response.
>
> Below are my opinions after reading the rebuttal.
>
> (1) Similar ideas in the literature of NLP should be discussed in the main body of the paper rather than the appendix. This is for correctly understanding the contributions of this work. To my understanding, the technical novelty is very limited and the application scenarios are new because the emergence of today's LLMs.
>
> (2) The authors just admitted the inherent limitations of their method.
>
> (3) The authors just ignored my concerns on the experiment results regarding the coherence of generated answers.
>
> (4) The authors argued that human evaluation might not be reliable due to significant differences in output formats across different methods. But how can LLM evaluation avoids the issue? I feel that one potential way to alleviate the problem is to have a baseline that explicitly prompt the LLMs to write answers as a list of points.
>
> (5) after reading other reviewers's comments. I started to question the practicality of the proposed method. (1) The total amount of compute actually increases due to additional generation of a "skeleton". (2) Why would a user like to see multiple points being generated simultaneously.

---

> ### Author Response · Authors · 2023-11-22
> **Response to Reviewer vbj6's follow-up (Part 1)**
>
> We thank Reviewer vbj6 for the follow-up opinions. We would like to clarify a few points, and use this opportunity to share our opinions. We hope the following responses can help us get to some consensus.
>
> **Q1: Similar ideas in the literature of NLP should be discussed in the main body of the paper rather than the appendix. This is for correctly understanding the contributions of this work. To my understanding, the technical novelty is very limited and the application scenarios are new because the emergence of today's LLMs.**
>
> **A**: We'll add a small paragraph to the main body. We respect that opinions might differ. But we can't agree with the opinion "the technical novelty is very limited and the application scenarios are new because the emergence of today's LLMs".
>
> As we have discussed in our response and the newly added Appendix D.3:
> 1. Existing hierarchical text generation methods design and train hierarchical architecture to model paragraphs, sentences, and words using different latent features. This means that they need training and **cannot be applied to existing LLMs**. And note that there is *no explicit language-form planning*.
> 2. As for explicit planning, existing methods rely on a specially trained module to do *close-form planning*. This means that they only reorder and group the input keywords, rather than producing free-form plans on "what to say" and "how to say".
>
> These characteristics are already vastly different.
>
> Actually, the main similarity between this work and the 2nd type of literature lies in the argument about "what high-level planning can bring to the answer quality": We all found that high-level answer planning helps with topic relevance and redundancy reduction.
>
> Differently:
> - The main **motivation** and **objective** of this work is to enable higher hardware utilization and acceleration.
> - The **target** is to question one major causes -- the fully sequential decoding.
> - The **idea** is to lay out the answer planning to make the overall generation workload more *parallelizable*.
> - The **method design** is a data-level prompting method instead of an architectural design, which can be directly applied to existing LLMs. Also, the **method design** utilizes the emerging ability of LLMs to let the LLM give and follow free-form answer planning.
> - As for the **application scenario**,  the important thing is that more and more long and structured texts are generated with the LLM. Among all application scenarios, general assistant-style chatbots and agent-agent interactions are indeed new scenarios enabled by LLMs, rather than previous news or story generation. But we don't regard this as the major methodological difference between SoT and previous literature.
>
> The above is how we define "motivation or objective", "target", "idea", "method design", and "application scenario". The **motivation and objective**, **target and idea**, and the **method design** are all very different. We're open to any further discussions.
>
> To summarize, SoT has limitations to be further improved. But **in terms of "novelty", actually, we think the most valuable point of SoT lies in its ideological novelty**. Our personal outlook is that instead of conducting fully sequential writing until the end of time, future AI models will use, or at least explore the direction of "explicit data organization for efficient and structured generation".
>
> **Q2: The authors just admitted the inherent limitations of their method.**
>
> **A**: We would like to humbly point out two things.
> 1. We didn't *just* admit the limitations. Our original manuscript disclosed all limitations and some possible pathways in the main text and analyzed failure patterns in the appendix. The only exception is the speed-up controllability issue compared with model- and system-level techniques (thanks to Reviewer 4Trv and ieuZ for pointing that out).
> 2. We think the limitations are not "inherent", but are solvable with many possible pathways. Even if we only go with purely prompting, there are many possible extensions (aggregation model, prompt an answer structure with dependencies, etc.) that can mitigate the current limitations. Not to mention that we can tune the LLM itself to "be better at" outputting a plan explicitly first and then writing the answer.

---

> ### Author Response · Authors · 2023-11-22
> **Response to Reviewer vbj6's follow-up (Part 2)**
>
> **Q3: The authors just ignored my concerns on the experiment results regarding the coherence of generated answers.**
>
> **A**: We would like to humbly point out that we **didn't** ignore this question. Just this question is saying the same thing as the second question, so we merge them into one question. As one can see, we have already merged the "coherence" question into Q2:
>
> > **Q2, about the general applicability**: The proposed method only makes sense when the answer is long, decomposable into several key points, and the interdependence of these key points is low (because the writing of each point only sees the skeleton). **As a result, the proposed method inevitably hurts the overall coherence of the answer.**
>
> We've already answered this question in the response to Q2. Current SoT is mainly suitable for questions that require answers whose structure contains points that are relatively independent. If it is not the case, SoT hurt the immersion and coherence, as we said in Section 3.2.4. Also, qualitative failure pattern analyses have been shown in the appendix of our original manuscript.
>
> **Q4: The authors argued that human evaluation might not be reliable due to significant differences i output formats across different methods. But how can LLM evaluation avoids the issue? I feel that one potential way to alleviate the problem is to have a baseline that explicitly prompt the LLMs write answers as a list of points.**
>
> **A**: Unlike humans who can easily judge whether the output is from the SoT generation or a normal generation explicitly, the LLM evaluation won't judge this. In other words, the LLM judge can have potential bias, but at least, it does the evaluation just based on the content, without the influences from personal value judgment (e.g., I like this work, given that I can tell which of the two answers is from SoT, I'll rate it higher and find evidence to explain my rating; or vice versa).
>
> > I feel that one potential way to alleviate the problem is to have a baseline that explicitly prompt the LLMs write answers as a list of points.
>
> This is a great suggestion! Thank you for this. We just started working on this.
>
> **Q5: After reading other reviewers's comments. I started to question the practicality of the proposed method. (1) The total amount of compute actually increases due to additional generation of a "skeleton". (2) Why would a user like to see multiple points being generated simultaneously.**
>
> **A**: Thanks for the questions. We have already answered these questions (1) to Reviewer JQY3, 4Trv, N4tW, and (2) to Reviewer 4Trv.
>
> For (1), we'd like to clarify that the additional computation overhead is *not* due to the generation of a skeleton but due to the prefilling of SoT prompts. And we'd like to emphasize that this additional prefilling computation has already been taken into consideration in all the reported results. Note that in memory-bounded scenarios (with a moderate batch size), even if the overall computation increases, the overall latency decreases, as the batched inference in SoT improves the computation utilization of hardware.
> And in computation-bounded scenarios (batch size >>1), the additional prefilling computation might cause a throughput decrease. This concern is already discussed in Section 6 and Appendix H in the original manuscript.
>
> As long as the SoT prompts can be compressed, the overhead can be made very small. Actually, in our experiments, we found SoT can be triggered with a much shorter prompt without any demonstrations on stronger models (e.g., LLaMA-2 7B and 13B). We keep the same prompt for all open-source models for the evaluation consistency of multiple models. There are other pathways (compress the prompt, share the prefilling computation, etc.) to reduce this overhead, as other reviewers pointed out and we discussed.
>
> For (2), please refer to the response to Reviewer 4Trv's Q3 and the newly added Appendix L.

---

> ### Author Response · Authors · 2023-11-23
> **Response to Reviewer vbj6's follow-up (Part 3)**
>
> Dear Reviewer vbj6,
>
> Following your suggestion in Q4, we conduct a quick human evaluation. Due to the time limit, we can only conduct the evaluation for Vicuna-7b-v1.3 model on Vicuna-80. We follow your suggestion to explicitly prompt the LLMs to write answers as a list of points to alleviate the differences of output formats. On Vicuna-80, we choose 16 out of 37 questions suitable for SoT. And we adopt four metrics (i.e., relevance, diversity, coherence and immersion) from LLMZoo to evaluate the answer quality.
>
> For the human evaluation, due to the time limit,we can't set up a more scientific human evaluation on Amazon Turk during the discussion period. Instead, we ask 11 undergraduate and graduate students in a lab to compare the responses from SoT and normal generation from the four aspects corresponding to the above four metrics. These students are all good at English, and fill the form independently. They are unaware of the implementation of SoT.
>
> For each question, we aggregate all 11 votes from the students. We regard SoT as "winning" or "losing" on this question when |vote_win-vote_lose|>=2, and regard it as "tie" when |vote_win-vote_lose|<2. As a result, the number of questions on which SoT achieves a win, tie, or lose is listed as follows:
>
> ```
> Relevance    9:5:2
> Diversity    6:2:8
> Coherence    9:3:4
> Immersion    9:5:2
> ```
>
> From the results, at least we can say, SoT is not worse on these questions. SoT outperforms the normal generation in relevance, coherence and immersion metrics, and achieves the competitive performance in diversity metric. We note  the higher diversity of the normal generation. We think it might be because that we force the LLMs to output the responses as a list of points, which is different from the normal generation process in the main paper.

---

### Author Response · Authors · 2023-11-19
**Global Response to All**

Dear All,

We appreciate all reviewers' valuable time invested in reviewing our paper. We are encouraged that the reviewers recognize the neat idea (ieuZ, 4Trv, JQY3, N4tW), wide evaluation and insights (ieuZ, 4Trv, N4tW), effectiveness in acceleration (KKRr, ieuZ, 4Trv, N4tW) and thus attractive for practical adoption (ieuZ), inspirations for future work (ieuZ), and good presentation (vbj6, 4Trv). We are also thankful for all the concerns and suggestions. The concerns and suggestions are helpful, inspiring, and worth further discussion.

According to the suggestions, we have revised the paper (including adjusting the introduction, limitation, and related work; adding application scenario discussion; adding a table-of-contents for the appendix) and added two new experiments (comparing with a longer normal generation, combining with quantization) in Appendix I.3 and Appendix J. We also added some new analysis plots of existing experiments (Figure 22, 25-27, 37-40 in the revision).

As a high-level summary, SoT's core idea is to **let LLM lay out the answer planning explicitly and then do the not-fully-sequential generation**, to enable (1) *parallel generation for better hardware utilization and acceleration*, as well as (2) *quality improvements in certain aspects*. As the LLMs get more widely applied and stronger, "data-level techniques for efficiency" could potentially become a standard component in the efficiency toolbox in the future. If this is the case, we hope SoT will be the first of them to inspire future work.

The current SoT solution indeed has limitations. We thank the reviewers for the insightful discussions and ideas. We **summarize the limitations here and describe possible remedies to call for future research**:
- *The current skeleton structure constraint (as the paper discussed and all reviewers pointed out) and the simple aggregation method hinder the general applicability to all question categories.* There are many possible pathways to address it. Our paper discussed some possible extensions, including more generalized structures like graph-of-thoughts and dynamic graph-of-thoughts. Reviewer ieuZ Q8 points out, and JQY3 Q5 implies that designing a better aggregation method might be helpful. Reviewer N4tW Q8 mentions that generating partial segments and prompting the LLM to complete might help with coherence.
- *The current long SoT prompt on open-source models induces prefilling computation overhead, which would cause a throughput decrease in computation-bounded scenarios (as the paper discussed and noted by Reviewer JQY3, 4Trv, N4tW).* Fortunately, we can foresee this issue caused by long SoT prompts to be solvable. Our paper discussed some pathways, such as system workload-aware SoT triggering, prompt tuning to compress the length, enabling LLM to self-trigger parallel generation, etc.
- *The acceleration ratio of the data-level techniques is not as controllable as model-level and system-level acceleration techniques (thanks to Reviewer 4Trv W2 for explicitly raising this, and Reviewer ieuZ Q3&Q5 for implying this), which might be undesirable for some users and scenarios.* It would be interesting to explore how to make speed-up more controllable, potentially by more fine-tuned content planning than the current skeleton.

---

### Meta-Review · Area_Chair_dwh5 · 2023-12-05

**Metareview:**

This work proposes a new decoding strategy where the model first outputs a skeleton of its generation and then fills in the skeleton in parallel.  Reviewers praised the quality of writing, the motivation, the extensive evaluations on multiple benchmarks and including multiple LLMs, and that SoT improves latency and also quality in some cases.  Some reviewers were dissatisfied with the LLM-evaluation, but the authors added a brief human eval during the discussion period.  Reviewers also pointed out that SoT is only appropriate for certain settings (both in terms of speed and performance), but I think SoT is a valuable contribution and does not need to make improvements for every domain to be worthwhile.  Therefore, I’m inclined to accept this paper.

**Justification For Why Not Higher Score:**

The paper is missing human evaluation (although the authors ran a tiny study during rebuttal) which is important for evaluating LLM quality, and it is unclear exactly where this method will or won't improve/damage performance or save compute.

**Justification For Why Not Lower Score:**

The paper presents a clever and intuitive idea that can improve LLM decoding speed.  This improvement will be useful to a broad audience at ICLR 2024.

---

### Decision · Program_Chairs · 2024-01-16

Accept (poster)